# A sensor-agnostic albedo retrieval method for realistic sea ice surfaces - Model and validation

Yingzhen Zhou[1], Wei Li[1], Nan Chen[1], Yongzhen Fan[2], and Knut Stamnes[1]

[1]Light and Life Laboratory, Department of Physics, Stevens Institute of Technology, Hoboken, NJ 07307, USA
[2]Cooperative Institute for Satellite Earth System Studies (CISESS) , Earth System Science Interdisciplinary Center (ESSIC), University of Maryland, College Park, MD 20740, USA,

**Correspondence:** Yingzhen Zhou (yzhou64@stevens.edu)

**Abstract.** A framework was established for remote sensing of sea ice albedo that integrates sea-ice physics with high computational efficiency, and can be applied to optical sensors that measure appropriate radiance data. A scientific machine learning (SciML) approach was developed and trained on a large synthetic dataset (SD) constructed using a coupled atmosphere-surface radiative transfer model (RTM). The resulting RTM/SciML framework combines the RTM with a multi-layer artificial neural network SciML model. In contrast to the Moderate Resolution Imaging Spectroradiometer (MODIS) MCD43 albedo product, this framework does not depend on observations from multiple days, and can be applied to single angular observations obtained under clear-sky conditions. Compared to the existing melt pond detection (MPD)-based approach for albedo retrieval, the RTM/SciML framework has the advantage of being applicable to a wide variety of cryosphere surfaces, both heterogeneous and homogeneous. Excellent agreement was found between the RTM/SciML albedo-retrieval results and measurements collected from airplane campaigns. Assessment against pyranometer data (N = 4144) yields RMSE = 0.094 for the shortwave albedo retrieval, while evaluation against albedometer data (N = 1225) yields RMSE = 0.069, 0.143, 0.085 for the broadband albedo in the visible, near infrared, and shortwave spectral ranges, respectively.

## 1 Introduction

Sea ice regulates global climate through several feedback mechanisms[1]. Broadband albedo is a critical parameter determining the radiative energy balance of the complex atmosphere-cryosphere system.

For decades, optical sensors deployed on geostationary and polar-orbiting satellites have been used to derive the global-scale surface broadband albedo. However, the majority of albedo products are land-surface products, while ocean surface albedo data (including sea-ice) are left blank (Qu et al., 2015). Table 1 compares the currently operational products and algorithms capable of retrieving albedo at the sea ice surface.

The broadband albedo estimated by APP-x is based on a narrow-to-broadband conversion (NTBC) of reflectance under a Lambertian-surface assumption (Key et al., 2016), which implies that the radiance reflected from the surface is isotropic and the value of albedo equals $\pi$ times the reflected radiance. However, fresh snow and white ice surfaces cannot be considered

---

[1]In this paper, 'sea ice' refers to the surface conditions of sea-ice's entire lifecycle: open-water, bare sea-ice, melt-pond, snow-covered sea ice and their mixtures. The phrase 'bare sea-ice' is used to refer to the sea-ice that is not covered by melt-pond or snow.

**Table 1.** Summary of the currently operational products and algorithms capable of retrieving albedo at the cryosphere's surface.

| Retrieval Method | Albedo Product (sensor) | Spatial resolution | Temporal resolution | Temporal Coverage | Reference |
|---|---|---|---|---|---|
| Narrow-to-broadband conversion (NTBC) | APP-x (AVHRR) | 25km | 1 d | 1982 - now | Key et al. (2016) |
| | CLARA-SAL (AVHRR) | 25km | 5 day or 1 month | 1981 - 2015, 2019-now | Riihelä et al. (2013); Karlsson et al. (2017) |
| Spectral-to-broadband conversion (STBC) | QA4ECV (MISR) | 1km | 1, 5, 15 and 31 days | 2000 - 2016 | Kharbouche and Muller (2018) |
| Kernel method (BRDF angular modeling) | MCD3D (MODIS) | 1km | 1 d | 2000 - now | Lucht et al. (2000); Schaaf et al. (2002) |
| Melt-pond-detection (MPD) algorithm | MERIS MPF V1.5 (MERIS) | 12.5km | 1 d | 2002 - 2012 | Zege et al. (2015); Istomina et al. (2015) |
| | OLCI MPF V1.5 (OLCI) | 12.5km | 1 d | 2016 - now | Istomina (2020) |
| Direct-estimation method | GLASS (MODIS, AVHRR) | 1km or 0.05 ° | 8 d | 1981 - 2019 | Qu et al. (2016) |
| | SURFALB L2, L3 (VIIRS) | 1km | 1 d | 2020 - now | Peng et al. (2018) |

Lambertian; dry snow and ice surfaces exhibit strong forward-scattering, and the impact of the bidirectional distribution of radiance reflected must be rectified in a post-processing step as discussed by Li et al. (2007).

Taking into account the anisotropic properties of the sea ice surface, the (broadband) albedo retrieval procedure requires three steps: (1) atmospheric correction; (2) anisotropy correction to obtain narrow-band or spectral albedo; and (3) use of spectral to broadband conversion (STBC) to integrate over the spectral range to obtain, or alternatively derive coefficients to estimate, broadband albedo (Knap et al., 1999; Liang, 2000; Xiong et al., 2002; Stroeve et al., 2005). The STBC coefficients derived by Liang et al. (1999); Liang (2000) were applied to retrieve sea ice albedo using MEdium Resolution Imaging Spectrom-
eter (MERIS) and Multi-angle Imaging SpectroRadiometer (MISR) instruments, respectively (Gao et al., 2004; Kharbouche and Muller, 2018). The retrievals are based on atmospherically corrected Level-2 (reflectance) data from the instruments, as opposed to Level-1 (radiance) data measured at the top of the atmosphere (TOA).

The Moderate Resolution Imaging Spectroradiometer (MODIS) MCD43A and MCD43D products describe the effect of reflectance anisotropy on land/ocean surfaces using the RossThick-LiSparse (RTLSR) model provided by Lucht et al. (2000),
which is a semi-empirical linear kernel-driven model that requires a sufficient amount of cloud-free observations within a 16-day window. Because MCD43 is a land albedo product, it only delivers very limited shortwave albedo values near the coast due to the lack of a spectral bi-directional reflectance distribution function (BRDF) for sea ice surfaces. In fact, there are only a few BRDF measurements that can be used to assist in correcting the anisotropy of snow and sea ice surfaces (e.g. Gatebe et al. (2005); Dumont et al. (2010); Gatebe and King (2016)), and they are far from conclusive in covering the complicated sea ice
surface or encompassing a sufficient angular and spectral range.

Due to the scarcity of observations, in more recent efforts, the BRDF of the cryosphere surface are approximated using radiative transfer models (RTMs). Examples of such efforts are the sea ice albedo retrieval based on the melt-pond-detection (MPD) algorithm (Zege et al., 2015) and the direct-estimation algorithm (Qu et al., 2016). Both algorithms try to establish a relation between TOA-measured radiance and surface albedo in two steps; the radiative processes in the atmosphere and
on the cryosphere surface were considered independently (i.e. 'uncoupled'). The atmospheric reflectance and transmittance are calculated with RTMs (Tynes et al., 2001; Vermote et al., 1997). Following this step, the calculated values are used to determine the TOA radiance/reflectance that corresponds to some specific surface condition, and the surface is modeled as the 'linear-blend' of sea ice/snow/melt-pond/water components.

In contrast, we present a framework that integrates a coupled atmosphere-surface RTM (Stamnes et al., 2018), with scientific machine learning (SciML) models. The coupled RTM model considers all radiative processes occurring in the coupled atmosphere, snow/ice system. Multiple reflections between the surface and the atmosphere as well as the atmospheric molecular and aerosol-induced modifications to the incident spectral distribution of the solar radiation are both taken into account. At the atmosphere-surface interface, Fresnel's equations and Snells' law appropriately describe light interactions as required. Additionally, this RTM is combined with a snow/ice/water model that simulates the snow/ice crystals and their inherent optical properties (IOPs), the snow/ice/melt-pond layering, and the impurities included within the snow/ice/water layers (Stamnes et al., 2011).

In the direct-estimation method, the same IOPs were deployed to derive the BRDFs of the snow surface (Qu et al., 2016). However, because the direct-estimation algorithm decouples the atmosphere from the ocean layer, it is unable to accurately simulate the 'snow-covered sea ice' situation; the 'snow surface' scenario refers to snow that has been placed on land. In a coupled RTM, snow is correctly simulated as a layer of snow on the surface above the air-sea ice interface. Similarly, the MPD algorithm uses an uncoupled RTM. Based on the absorption of yellow pigments in ice, it models sea ice's BRDF exclusively for dry and white ice, ignoring the effects of air bubbles and brine pockets (Zege et al., 2015). In a coupled RTM, sea ice is simulated as a layer of ice with brine pockets and air bubble inclusions floating on deep ocean water.

This paper is structured as follows: Section 2 provides a summary of the RTM/SciML framework for albedo retrieval, describes the cloud screening and surface classification model, and introduces the validation and comparison datasets used in this study. Section 3 is devoted to validation of the albedo-retrieval products, and Section 4 addresses the possible sources of uncertainty in the validation data. In Section 5, the albedo product is compared to the MCD43 product (Section 5.1), two MPD-based products (MERIS and OLCI, respectively in Sections 5.2 and 5.3) and two direct-estimation products (GLASS and VIIRS, in Section 5.4). A conclusion and summary is provided in Section 6.

## 2 Methodology

### 2.1 Overall framework

In this work, we present a new method for albedo retrieval: scientific machine learning based on a coupled RTM (hereafter RTM/SciML-albedo algorithm). Figure 1 shows a flowchart of the proposed RTM/SciML framework for albedo retrieval, while Sections 2.2∼2.6 discuss the steps in more detail.

First, a synthetic dataset (SD) is constructed using a coupled-RTM, AccuRT (Stamnes et al., 2018). It consists of TOA radiances simulated in suitable satellite channels as a function of observational and solar angles, as well as the associated broadband albedo at the surface. This SD encompasses a range of different surface types, including snow-covered ice, melt ponds, and bare sea ice, as well as their mixtures. All optical properties of surface and atmospheric constituents, as well as radiative processes within the coupled atmosphere-snow-sea ice-water system, are deduced from first principles; information of both the surface BRDF and the IOPs of the atmosphere is implicitly taken into account. We are thus saved from the procedure to perform atmospheric corrections.

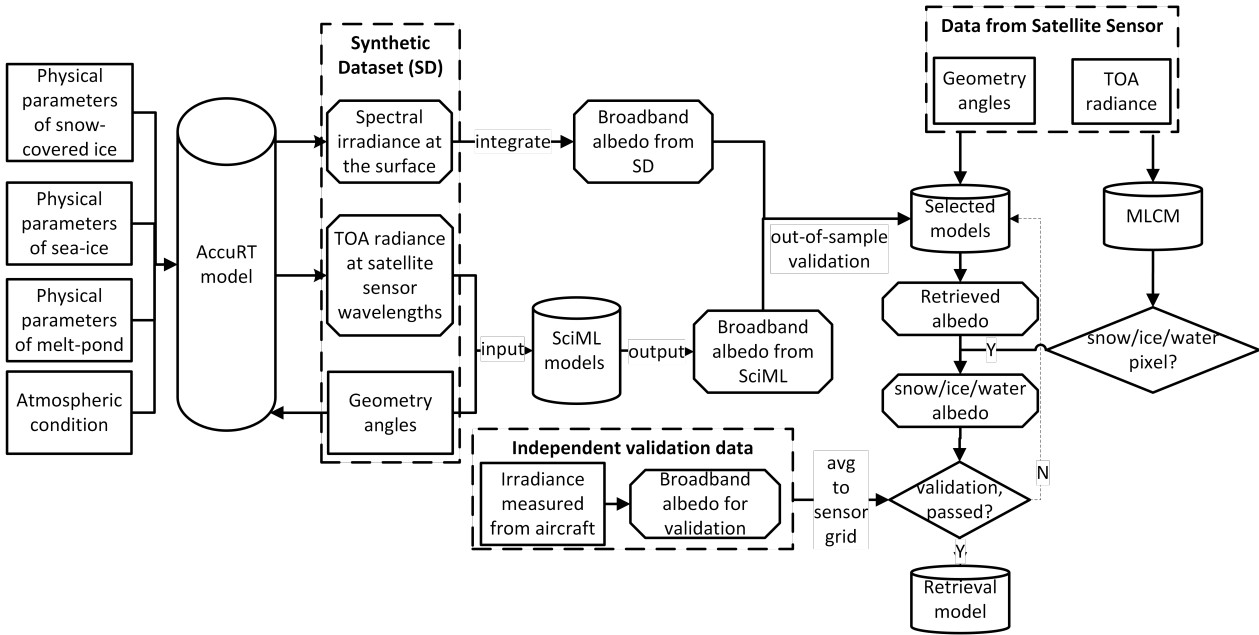

**Figure 1.** Flowchart of the proposed RTM/SciML framework for albedo retrieval.

The coupled RTM ensures that the 'forward problem' is solved correctly, yielding reliable total radiance and irradiance values at any location of the coupled system given the IOPs of the system constituents. Following the physically consistent SD, scientific machine learning (SciML) models can be used to approximate solutions to the 'inverse problem', which answers

the question, 'given the observed TOA radiance and the sun-satellite geometry angles, what is the most likely albedo of the sea ice surface?' through an implicit iterative process (i.e. by minimizing a loss function repeatedly during the training of SciML models). The SciML models perform without reliance on predefined spectral reflectance threshold values for individual types of surface, which eliminates errors caused by incorrect surface condition assumptions. When deployed in practice, the surface classification and albedo retrieval are separate processes; an independent machine learning classification mask (MLCM,

described briefly in Section 2.4) performs both cloud screening and surface pixel classification.

## 2.2 Synthetic Dataset (SD) generated by coupled-RTM

The AccuRT Radiative Transfer model (Stamnes et al., 2018) is able to simulate a coupled system with changes of refractive index across the atmosphere-water (in solid or liquid form) interface. Inputs to AccuRT are the IOPs of the two adjacent coupled slabs (upper slab is atmosphere/snow and lower slab is ocean/ice). Each slab can be partitioned into many adjacent

layers, with the IOPs being constant within each layer but varying between them. The IOPs depend on the medium's absorption and scattering coefficients, as well as the scattering phase functions. Within the AccuRT computer code, Mie theory and the particle size distributions are used to generate IOPs based on user-defined physical properties of the medium (Stamnes et al.,

2011). Table 2 summarizes the parameters used to calculate the IOPs of bare ice, snow-covered ice, open water, melt-ponds, and aerosols. Appendix A discusses the value ranges of the physical parameters in more detail.

Among the sea ice inclusions, air bubbles are modeled as pure scatterers, brine pockets scatter and absorb light, and black carbon impurities mainly absorb light. The effective grain size $r_e$ of snow particles closely resembles the effective light path traveled by a photon, and hence a larger $r_e$ suggests a lower reflectance (and albedo). Additionally, bulk properties such as ice/pond/snow thickness ($h$, $h_{\mathrm{m}}$, $h_{\mathrm{s}}$,) and snow density ($\rho_s$) also impact the optical depth of the medium. Notably, the parameterizations employed here are consistent with the physics of sea ice/snow/water (Grenfell and Maykut, 1977; Grenfell

and Perovich, 1984; Warren, 2019).

**Table 2.** Properties used to compute the inherent optical properties (IOPs) of bare ice, snow-covered ice, open water, melt-pond, and aerosol, which are utilized as inputs to the radiative transfer model (Stamnes et al., 2011).

| Type | Parameter | Properties | Description |
|---|---|---|---|
| bare ice | $X_{ice}$ | $h$ | sea ice thickness |
| | | $V_{\mathrm{br}}$ | brine pocket volume fraction |
| | | $r_{\mathrm{br}}$ | brine pocket radius |
| | | $V_{\mathrm{bu}}$ | air bubble volume fraction |
| | | $r_{\mathrm{br}}$ | air bubble radius |
| | | $f_{\mathrm{bc,i}}$ | black carbon impurity fractions in ice |
| | | $X_{water}$ | physical properties of the ocean water beneath the sea ice layer |
| snow-covered ice | $X_{snow}$ | $r_{\mathrm{e}}$ | effective grain size of snow particles |
| | | $\rho_{\mathrm{s}}$ | snow density |
| | | $h_{\mathrm{s}}$ | snow depth |
| | | $f_{\mathrm{bc,s}}$ | black carbon impurity fractions in snow |
| | | $X_{\mathrm{ice}}$ | physical properties of the sea ice below the snow-cover layer |
| | | $X_{water}$ | physical properties of the ocean water beneath the sea ice layer |
| ocean water | $X_{water}$ | $h_{\mathrm{w}}$ | open-water depth |
| | | $f_{\mathrm{chl}}$ | chlorophyll-a concentration |
| | | $f_{\mathrm{CDOM}}$ | colored dissolved organic matter (CDOM) concentration |
| melt-pond | $X_{melt}$ | $h_m$ | melt pond thickness |
| | | $X_{\mathrm{ice}}$ | physical properties of the sea ice below the melt pond layer |
| | | $X_{\mathrm{wat}}$ | physical properties of the ocean water below sea ice layer |
| aerosol | $X_{aerosol}$ | $\tau_{\mathrm{aero}}$ | aerosol optical depth in the atmospheric layer |

Pseudo-random values of the physical parameters of ice, water, and snow within realistic ranges were applied to generate 158,000 cases each for (i) bare-ice, (ii) ice with melt water, and (iii) ice with snow cover (the value ranges are shown in Tables A1-A3). These surface and atmospheric parameters, together with randomly distributed solar/viewing geometries (Table A4) form a dataset suitable for generating a SD of TOA radiances and corresponding albedo values that can be used to train a

SciML algorithm.

Altogether, the $158{,}000 \times 3 = 474{,}000$ configurations cover most expected combinations of surface types and atmospheric conditions encountered during the sea ice life cycle. For each case, TOA radiances at appropriate sensor channels as well as the three broadband albedo values ($\alpha_{\mathrm{VIS}}$, $\alpha_{\mathrm{NIR}}$, $\alpha_{\mathrm{SW}}$) were computed based on downward and upward irradiances simulated using AccuRT (Stamnes et al., 2018).

To summarize, AccuRT's nature as a coupled RTM and as a model that incorporates the physical properties of ice/snow/water to calculate their IOPs provides the following benefits in order to tackle the 'forward problem':

1. There is no need to (i) perform atmospheric correction, (ii) build a BRDF dataset, or (iii) employ angular modeling or anisotropy correction. All these effects are implicitly included in the coupled RTM.

2. Because each simulation performed by the coupled RTM represents a combination of atmosphere/surface conditions, sun-sensor zenith and azimuth angles, a SD can be constructed that is designed to include (i) the complicated surface and atmosphere conditions by varying the optical properties in Table 2, and (ii) the possible combinations of illumination/viewing geometries (Table A4).

With a comprehensive SD, we are not restricted to a linear regression model (as in the direct-estimation method) to derive the relationship between the spectral radiance at TOA and the blue-sky albedo at the surface; any reliable SciML model may be evaluated and compared as long as it is capable of solving the 'inverse problem' (i.e. a regression model, with albedo being the target and the radiances as well as sun-satellite geometry angles being the features).

## 2.3 Selection of radiance channels for albedo retrieval

With our knowledge of radiative transfer theory and the differences in the radiative properties of the constituents in the coupled atmosphere-surface system, we first chose the input channels based on the following criteria:

– Avoid wavelengths with significant absorption by water vapor and/or other atmospheric constituents.

– Avoid sensor channels that have been found to be saturated in previous sensitivity investigations.

– Select wavelengths that, based on their albedo spectra, can best identify snow cover, bare ice, open water, and melt-pond surface conditions.

With the assistance of the auto-associative neural network (AANN) technique, channels with a significant reconstruction error are deemed unsuitable for use as input to the retrieval model. More specifically, an AANN is trained using the synthetic data generated by the RTM, which takes as input the three sun-satellite geometry angles as well as all radiance data that meet the aforementioned requirements and outputs all radiances. The trained AANN is believed to have picked up on the patterns in the RTM-generated dataset. Following that, the AANN is fed the same input features derived from the satellite sensor. We calculate the absolute percentage error of the reconstruction output and prune channels with an error greater than 5%.

This method is intended to avoid 'covariate shift' — a phrase used in machine learning to refer to the difference between independent variables in training and real-world data. Covariate shift is due to either (a) the saturation of certain satellite

channels, which results in a much narrower dynamic range of radiance data from the satellite sensor (real world) than that calculated using the RTM (training data), or (b) the response function and wide wavelength range results in a non-negligible difference between the radiance derived from the central wavelength and that obtained from the sensor. It has been demonstrated that the AANN technique is effective in detecting mismatches between data acquired for the retrieval task and data utilized for training; A recent paper (Fan et al., 2021) discusses how the AANN approach was used to identify both optimal channels for retrieving ocean color products using a variety of sensors and the pixels that are outside the scope of training dataset.

Similar approaches have been used to identify acceptable channels for albedo retrieval. Table 3 lists the MODIS channels that were utilized to retrieve albedo, as well as the Global Change Observation Mission - Climate (GCOM-C)/SGLI channels that were evaluated and eventually employed [2].

## 2.4 Surface classification and cloud filtering

Imperfect cloud screening brings considerable uncertainty to the retrieved sea ice albedo. To mitigate this uncertainty, pixels covered by clouds are detected and removed by a neural network based cloud screening and surface classification algorithm (MLCM) developed by Chen et al. (2018). The MLCM (short for Machine Learning Classification Mask) is a threshold-free algorithm trained by extensive radiative transfer simulations. It can be applied to a great variety of surface types to provide reliable cloud mask and surface classification. A comparison between the MLCM and other standard cloud mask algorithms showed that the MLCM is better able to detect cloud edges and deal with high solar zenith angles (Chen et al., 2018). Section 3 indicates that the MLCM can assist in filtering cloud, fog (sea smoke) and hazy atmospheric conditions.

The MLCM also has the capability to distinguish snow-covered sea ice pixels (with a minimum snow depth of 1 cm) from bare sea ice pixels. Independent treatment of classification and albedo retrieval ensures that even on highly heterogeneous surfaces or in conditions where classification of certain pixels is difficult (e.g. the long-standing difficulty in distinguishing between slushy mixtures of ice and water, very thin ice, and melt ponds), the retrieval of their surface albedo values is unaffected.

## 2.5 Data

### 2.5.1 Radiance data from the MODIS and SGLI sensors

In this study, the Level 1B calibrated radiance data (MOD021) from the MODIS sensor and the infrared scanner (IRS) as well as the near-infrared radiometer (VNR) on the Second-generation GLobal Imager (SGLI, data available at JAXA page) were employed for albedo retrieval and surface classification. While only two sensors are discussed, the retrieval method described in Section 2 is generic and applicable to any satellite sensor having suitable radiance data (e.g. the Visible Infrared Imaging Radiometer Suite, abbreviated as VIIRS).

Central wavelengths used by MODIS and SGLI sensors to retrieve albedo and obtain cloud / surface-classification mask are provided in Table 3. Data in all channels have been aggregated at a common spatial resolution of 1 km.

---

[2]Our team initially discovered the saturation issue in the 673.5 nm channel using AANN, submitted the finding to the GCOM-C/SGLI team, and obtained confirmation of the issue.

**Table 3.** Central wavelengths used by SGLI and MODIS to retrieve albedo and obtain cloud and surface classification mask. The 673.5 nm wavelength channel from SGLI was proven to be saturated and hence was not used to derive albedo.

| | SGLI channels | | | MODIS channels | |
|---|---|---|---|---|---|
| $\lambda$ (nm) | albedo | cloud & classification | $\lambda$ (nm) | albedo | cloud & classification |
| 380 | | x | | | |
| 443 | x | x | 469 | x | x |
| 530 | x | x | 555 | x | x |
| 673.5 | | x | 645 | x | x |
| 868.5 | x | x | 858.5 | x | x |
| 1050 | x | x | 1240 | x | x |
| 1630 | x | x | 1640 | x | x |
| 2210 | x | x | 2130 | x | x |

### 2.5.2 Independent validation data

Three campaigns conducted in the Arctic provided broadband irradiance data across the Arctic sea ice surface. They are: the ACLOUD (Arctic Cloud Observations Using Airborne Measurements During Polar Day) campaign conducted during the 2017 spring-summer transition, the AFLUX (airborne measurements of radiative and turbulent FLUXes of energy and momentum in the Arctic boundary layer) campaign, which took place in April 2019 north of Svalbard, and the MOSAiC (Multidisciplinary drifting Observatory for the Study of Arctic Climate) campaign, which was conducted in late August to September 2020 (Wendisch and Brenguier, 2013; Lüpkes, 2017; 201, 2016).

With the MLCM, we performed cloud filtering and compared the flight transits to the cloud-free area. By calculating the percentage of cloud coverage ($f_c$) for the matching dates in the latitude-longitude range of flight operations, only seven days were found to include clear-sky sea ice observations ($f_c \leq 90\%$), and the flight transits were retained (Table 5). Measurements from two days during the AFLUX campaign were partly from clear-sky, while the remainder were entirely from broken clouds (for a description of the AFLUX data, see Stapf et al. (2021b)). The MOSAiC campaign included fewer than 50 valid data points (Jäkel et al., 2021b), and all obtained for broken cloud conditions. To eliminate errors caused by dense cloud cover, MOSAiC data were omitted from validation.

During the ACLOUD and AFLUX campaigns, the upward and downward irradiance data ($0.2 \sim 3.6~\mu$m) were collected by two pairs of CMP-22 pyranometers (Stapf et al., 2019, 2021a). The pyranometers operated at a frequency of 20 Hz and were mounted on the Polar 5 (the ACLOUD campaign used an additional aircraft, Polar 6) research aircraft. Pre-processing was used to avoid data received during high aircraft pitch/roll angles and suspicious equipment frost conditions. Technical details of the pyranometer are provided by Wendisch and Brenguier (2013). Along with the pyranometers on Polar 5 and 6, a Spectral Modular Airborne Radiation measurement system ("SMART Albedometer") on Polar 5 measured spectral irradiances between 0.4 and 2.155 $\mu$m at a frequency of 2 Hz during the ACLOUD campaign (Jäkel et al., 2018).

Gröbner et al. (2014); Ehrlich et al. (2019) reported that the pyranometer's uncertainty was less than 3%, whereas the uncertainty of the "SMART Albedometer" was 7%.

### 2.5.3 Pre-processing of validation data

The albedo obtained from SciML has the same spatial resolution as the geolocation data (MOD03), which is 1 km, whereas the aircraft's pyranometers/albedometer have a significantly smaller footprint. As a result, when evaluating SciML models, the estimated albedo is collocated with the MODIS grid and the average value of about 170 measurements from each flight is mapped to a single MODIS pixel. With the MLCM, the pixels with cloud contamination are filtered, and only three surface types were kept in the validation dataset (i.e. sea ice, snow-covered sea ice, and water).

Finally, while the SciML-derived albedo is valid for surface observations, there may be a bias between it and the albedo measured at aircraft height. The Polar 5 and Polar 6 aircraft reached 3000 m height during the mission, and therefore, the albedo measurements were influenced by the entire atmospheric layer below the aircraft, resulting in a significant variance in albedo results. To account for the difference caused by the atmospheric constituents, we used the coupled RTM (AccuRT) to simulate the albedo of snow, bare ice, melt pond, and open water surface at three different levels: surface ($\alpha_s$), low flight level ($\alpha_l$, $l = 350$ m), and high flight level ($\alpha_h$, $h = 3000$ m). Albedo ratios, $r_l = \frac{\alpha_l}{\alpha_s}$ and $r_h = \frac{\alpha_h}{\alpha_s}$, were calculated to determine the difference in albedo induced by the atmospheric layer below the flight height.

Figure 2 illustrates the aforementioned spectral and broadband albedos, as well as the ratios. There is significant influence of multiple scattering above the open water surface due to atmospheric components, particularly in the visible (VIS) range; the albedo at $h = 3000$ m is double that on the surface, whereas the presence of an atmosphere with aerosols results in a decrease in albedo over a bright surface. Among these, the reduction is particularly noticeable in the near-infrared (NIR) band, where air absorption results in a 4% and a 22% decrease in albedo at low and high levels, respectively. Similar simulation results were found by Jäkel et al. (2021a), using the Two-streAm Radiative TransfEr in Snow (TARTES) model (Libois et al., 2013). As can be observed, the difference between low-level and surface albedo is less than 5%, whereas the difference between high-level and low-level albedo is significantly greater. As a result, we did not use aircraft observations taken above 350 m in order to improve the validation of the 'surface' albedo retrieval.

In all, the selected flight sections are depicted in Fig. 3. Up to four satellite images per day could be employed for retrieval and evaluation in the polar regions. The MODIS transits that correspond to the aircraft operation time and $f_c$ are listed in Table 5.

### 2.5.4 Comparison data

The retrieval results from RTM/SciML models are compared to the currently operational albedo retrieval products listed in Table 1.

**MODIS MCD43D:** The 1-km spatial resolution MODIS MCD43D products are the successors to the MCD43B products. The retrievals are performed daily using kernel weights that best represent the majority of situations across the 16-day period, with the day of interest emphasized. MCD43D49∼51 correspond to the Black Sky Albedo (BSA) for the visible (0.3∼0.7 $\mu$m,

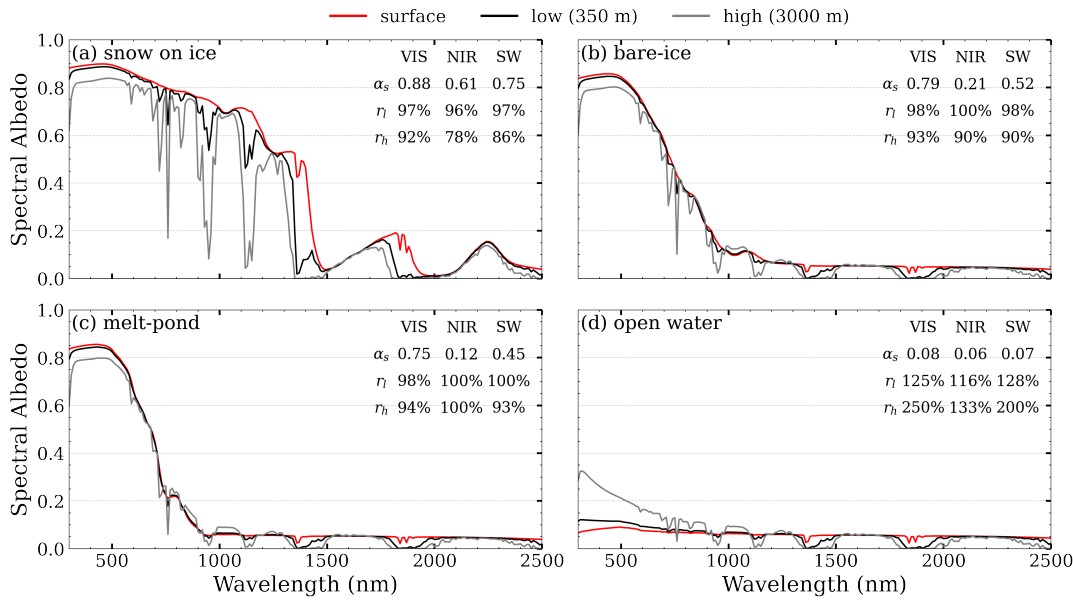

**Figure 2.** Spectral albedo computed by the RTM at various altitudes (350-m in black line, and 3000-m in gray line) and the surface (red line).

VIS), near infrared (0.7∼5.0 $\mu$m, NIR), and shortwave (0.3∼5.0 $\mu$m, SW) bands, while MCD43D59∼61 correspond to the White Sky Albedo (WSA) for the three broadband ranges (Schaaf and Wang, 2015b).

**MERIS MPF V1.5 and OLCI MPF V1.5:** Istomina et al.(2015; 2020) developed two albedo retrieval products, utilizing MERIS data from Envisat-1 and Ocean and Land Colour Instrument (OLCI) data from Sentinel-3, respectively. The MERIS sensor was only operational from 2002 to 2012, and the OLCI-product is the successor to the MERIS-albedo product (Istomina, 2020); both products are based on the MPD algorithm proposed by Zege et al. (2015). Daily retrieval data are gridded onto a polar stereographic grid with a 12.5 km resolution (the NSIDC grid). In Istomina et al. (2015), the MERIS-albedo product was evaluated by comparing the estimated albedo values to aerial measurements from the MELTEX campaign. Land-fast ice showed the best agreement, with an R-squared value of 0.84 and an RMSE of 0.068 for 169 matched pixels.

**GLASS (AVHRR) and GLASS (MODIS):** Global Land Surface Satellite (GLASS) products are primarily derived from NASA's Advanced Very High Resolution Radiometer (AVHRR) long-term data record and MODIS data, together with other satellite data and auxiliary information (Liang et al., 2021). The sea ice surface albedo in GLASS is derived using the direct-estimation algorithm (Qu et al., 2016); a spatiotemporal filtering algorithm Liu et al. (2013) is utilized to provide gap-free albedo with an 8-day temporal resolution. The spatial resolution of GLASS-albedo retrieved from the AVHRR sensor is 0.05° and from MODIS sensor is 1 km. In Qu et al. (2016), the GLASS (MODIS) product was compared to Tara polar ocean expedition measurements. During the 90-day expedition, the daily average albedo from cruise measurements with around 50 matched retrieval-measurement data points yielded an R-squared value of 0.67.

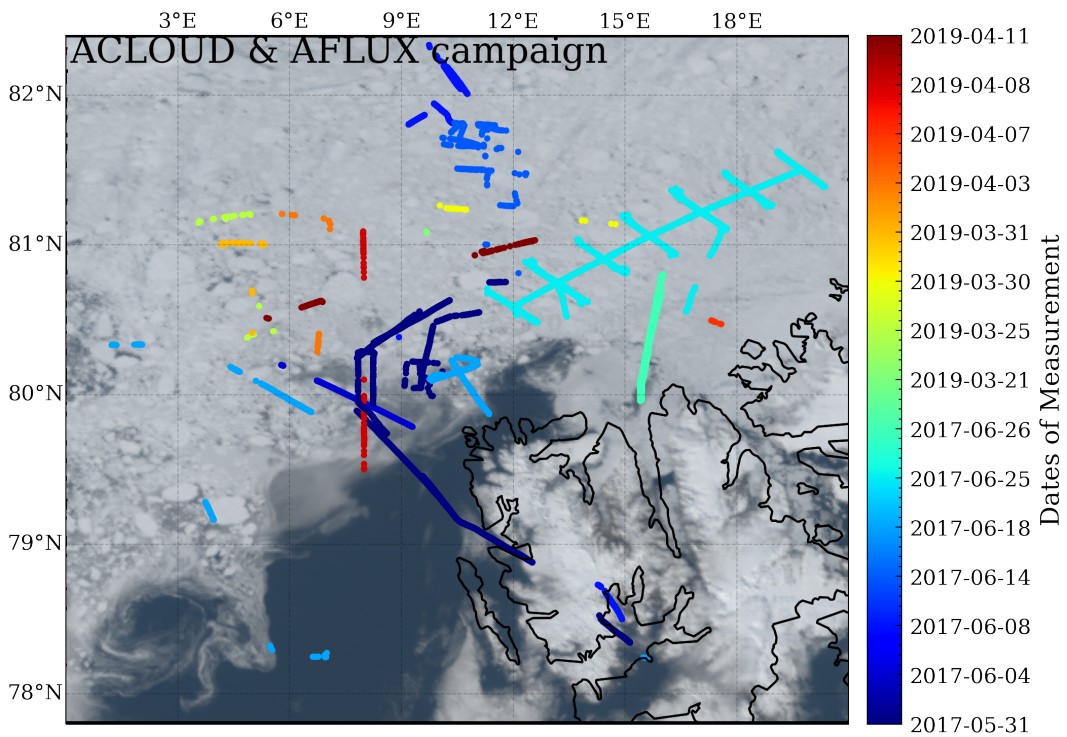

**Figure 3.** ACLOUD and AFLUX transits of selected flight segments from Polar 5 and Polar 6. In the background, the true color composite from June 25th, 2017 is shown; MODIS channels 645 nm, 555 nm, and 469 nm are used as (R,G,B) bands for the true color RGB. The dates of flight operations are denoted by hues defined in a colorbar placed on the right.

**VIIRS L3-SURFALB:** Peng et al. (2018) improved the direct-estimation method for sea-ice retrieval proposed by Qu et al. (2016) and made it applicable to the VIIRS. The BRDF in Qu et al. (2016) is a weighted average of sea ice, snow-covered ice, and open water, with melt-pond conditions omitted. In Peng et al. (2018), all four surface conditions are evaluated and the MPD algorithm is used to derive the BRDF of melt-pond. The VIIRS surface albedo (SURFALB) product has been validated using ground truth from the Greenland Ice Sheet and snow-covered land (Peng et al., 2018). Note that the validation data are not applicable to the highly fluctuating sea ice surface; sea ice refers to a floating sheet of ice formed from ocean water, which might be covered by melt-ponds or snow during its lifecycle. The spatial resolution of both the SURFALB Level-2 (L2) Granule product and the Level-3 (L3) Gridded product is 1 km, and the L3 data is used for comparison in this study.

When computing the statistical values (albedo, Pearson-r, mean, etc.), the RTM/SciML-retrievals are regridded to the same spatial and temporal resolution as their counterparts (e.g. 12.5km for OLCI/MERIS and 8-day average for GLASS), and if matching validation data are available, the flight measurements are likewise rescaled to the same grids. In retrieval-map figures, data are represented with their original spatial resolution. The CLARA-SAL (Karlsson, Karl-Göran et al., 2012; Karlsson et al., 2017) and APP-x (Key et al., 2016) albedo products produced from the AVHRR sensor are also provided for reference.

## 2.6 Scientific Machine Learning (SciML) model

Neural network models, specifically models with the multi-layer artificial neural network (MLANN) structure, have been demonstrated to be useful for retrieving and estimating snow and sea ice parameters. Successful implementations include the sea ice and snow thickness retrievals using microwave data (Herbert et al., 2021; Hu et al., 2021; Zhu et al., 2021; Wang et al., 2020; Braakmann-Folgmann and Donlon, 2019; Lee et al., 2019; Tedesco and Jeyaratnam, 2016; Cao et al., 2008; Tedesco et al., 2004), and sea ice concentration (SIC) or melt pond fraction (MPF) retrievals using radiance or reflectance data (Chi et al., 2019; Karvonen, 2017; Rösel and Kaleschke, 2012).

A MLANN's computational unit is a neuron (alternatively called a perceptron), which is organized in a network topology. A set of functions $f_j(x; w_j, b_j)$ connected in a chain can be used to describe the architecture of a fully connected neural network with $n$ layers in total (including the input and output layers):

$$F(y) = f_n \left( \ldots f_3 \left( f_2 \left( f_1(x) \times w_1 + b_1 \right) \times w_2 + b_2 \right) \ldots \right). \tag{1}$$

The previous layer's output is transformed by $f_j(x) \times w_j + b_j$, and then an activation function $f_{j+1}$ (which is typically non-linear) is applied to perform an element-wise operation.

Generally, as the network grows deeper, the features become more abstract and complicated, as each succeeding layer is constructed from the already transformed features of the prior layer (LeCun et al., 2015). The network's $n - 2$ hidden layers enable it to describe non-linear relations between the independent and dependent variables $(x, y)$. The matrices $w_j$ define the weights for linear transformations between layers, whereas the vectors $b_j$ define the biases. The final layer (output layer) has three neurons encoding the predicted albedo values ($\hat{\alpha}_{\text{VIS}}$, $\hat{\alpha}_{\text{NIR}}$, $\hat{\alpha}_{\text{SW}}$).

Neural network models with different configurations are trained using 80% of the SD, and the portion of the SD that was not included in the network training process was held out for validation (the 'out-of-sample validation' step shown in Fig. 1). To obtain an efficient MLANN, both the network structure's topology and hyperparameters should be tweaked properly.

A stochastic gradient descent (SGD) optimization is used to update the weights and biases in an MLANN during back-propagation. To perform the SGD, the adaptive moment estimation (Adam, Kingma and Ba (2014)) was chosen, which trains the models in 200 epochs with a batch size of 64. A MLANN's hyperparameters include the learning rate and the activation function. To determine the optimal learning rate, Bayesian optimization was employed (Brochu et al., 2010), and the Rectified Linear Units (ReLU, (Nair and Hinton, 2010)) were used as the activation function in the hidden layers. Batch normalization (Ioffe and Szegedy, 2015) is performed to enhance the MLANN's generalization capabilities and make the network less sensitive to random initialization of the weights and biases (Santurkar et al., 2019). To avoid overfitting, dropout layers (Srivastava et al., 2014) were included as a regularization for networks with more than two hidden layers. In our evaluation, dropout layers with a rate of 0.2 were found to be optimal, implying that one in every five inputs is randomly eliminated from each update cycle.

The network structure with three hidden layers was shown to perform well when evaluated using the hold-out dataset (the 20% of SD) [3]. Furthermore, when these MLANNs were applied to the MODIS sensor, the distributions of their retrieval results were very comparable (see Fig. B1 (b)). Consequently, the "winning model" was determined by comparing the retrieval results of each candidate against independent validation data. The independent validation data for the MODIS sensor were derived from pyranometer measurements described in Sec. 2.5.2, and 4000 entries were sampled from the total dataset (N=7964). The $16 \times 10 \times 8$ network topology demonstrated a slight overall performance advantage (Table B1) [4]. The final retrieval model for the SGLI sensor was determined by comparing the R-squared of SGLI and MODIS retrieval results from the same day [5].

As a final note, the "SMART Albedometer" measures within a narrower irradiance range (as compared to pyranometer). Therefore, a separate MLANN model was trained utilizing broadband albedo from different ranges (i.e., modifying the ir-radiance ranges of the SD in the 'integrate' stage of Fig. 1). The difference in wavelength-range is summarized in Table 4. Throughout the text, the two models are discussed separately: (a)$\sim$(c) of Fig.4 and Fig.9 as well as Fig.7 (a) are broadband albedo values with adjusted ranges, while the remainder of the discussion pertains to the $300 \sim 2500nm$ broadband [6].

**Table 4.** Difference between the wavelength ranges of the retrievals from the two MLANN models. The validation of retrieval results are shown in Figures 3, 6 and Table A2.

|  | Model 1 | | Model 2 | |
| --- | --- | --- | --- | --- |
|  | λ range (nm) | validation data | λ range (nm) | validation data |
| Visible | 300-700 | / | 400-700 | albedometer |
| Near Infrared | 700-2500 | / | 700-2100 | albedometer |
| Shortwave | 300-2500 | pyranometer | 400-2100 | albedometer |

---

[3]A deeper structure was also tested, but the error from a network with more than three hidden layers doubles. Liu et al. and Rösel et al. previously employed fewer neurons in each layer (no more than 15) and assigned the first hidden layer the same number of neurons as the feature input, in the hope that after the model training, each neuron represents exactly one satellite channel or one of the geometry angles (i.e. the lingo 'semantics' in deep learning). However, there is no guarantee that these input-neuron relations will remain one-to-one during actual neural network training. In fact, the back-propagation process combines all of the inputs in the '$\times$' operation (the so-called 'non-locality' effect).

[4]This N=4000 is a subset of the data shown in Fig. 4 (d). However, note that the MLANNs with three hidden layers showed highly comparable performance on this independent validation data (see Table B1 and Figure B1 (a)). This stage is performed solely for the purpose of selecting an MLANN model that supports discussions throughout the paper (as it is infeasible to include all the enlisted MLANNs for comprehensive comparisons with the other retrieval methods). The authors also evaluated various machine learning (ML) models; nonetheless, the loss on hold-out dataset was larger for all ML models than for the MLANN models. Figure B1 (c)-(d) illustrates how this 'under-performance' impacts final retrievals when these models are applied to MODIS images.

[5]The SGLI on GCOM-C was launched in late 2017. The data included in Fig 5 are, to our knowledge, the only cloud-free validation data during its operational periods. Therefore, retrieval results from MODIS of the same date were utilized as a benchmark to select a retrieval model for the SGLI sensor, taking advantage of the fact that the observation by MODIS and by SGLI are only 15 minutes apart.

[6]Although Model 1 was trained using irradiance in the wavelength range of 0.3 to 2.5 $\mu$m rather than the pyranometer's 0.2 to 3.6 $\mu$m range, we would expect a small deviation due to the wavelength range difference, because there is virtually no radiation reaching the surface of the Earth for wavelengths shorter than 0.3 $\mu$m due to ozone absorption in the stratosphere, and because the contribution to the albedo for wavelengths longer than 2.5 $\mu$m is negligible.

## 3 Validation

### 3.1 Validation of MODIS-retrievals

Polar-orbiting satellites transit the two polar areas up to four times per day, and cloud-free retrievals from all transits (Table 5) are used to compare with the independent validation data measured on Polar 5 and Polar 6 aircraft. Due to the absence of a time constraint on the retrieval data, a single gridded airplane measurement can match many satellite retrievals.

**Table 5.** Time stamps for airborne observations, satellite overflights in UTC, and cloud pixel percentages in the latitude-longitude ranges of aircraft during the ACLOUD and AFLUX campaigns (77.8~82.4°N, and -0.25~20.5°E.). Note that days with just broken-cloud observations (cloud coverage greater than 90%) are not included in the table.

| Date | Polar 5 | Polar 6 | MODIS | $f_{cloud}$ (%) |
|------|---------|---------|-------|-----------------|
| 31 May 2017 | 15:05-18:57 | 14:59-19:03 | 14:30, 16:05, 17:45 | 31.35% |
| 8 June 2017 | 7:36-12:51 | 7:30-12:20 | 12:00, 13:40, 15:15 | 74.37% |
| 18 June 2017 | 12:03-17:55 | 12:25-17:50 | 11:00, 12:40 14:15, 15:55 | 82.69% |
| 25 June 2017 | 11:09-17:11 | 11:03-16:56 | 11:05, 12:45 14:20, 16:00 | 28.06% |
| 26 June 2017 | 12:34-15:17 | 12:32-14:48 | 11:50, 13:25 15:05, 16:40 | 33.74% |
| April 08 2019 | 9:11-13:50 | / | 11:30, 13:10 | 46.52% |
| April 11 2019 | 9:42-12:46 | / | 12:00, 13:40 | 81.12% |

Figure 4 indicates that the RTM/SciML retrieval method is capable of producing satisfactory albedo outputs with low errors and high Pearson-r coefficients. The agreement between the retrieved shortwave albedo and the pyranometer-measured albedo is better than with the albedometer; the regression line (marked in red) connecting the pyranometer-measured albedo and the retrieval almost completely overlaps the (0,1) line, indicating a perfect one-to-one correspondence. Note that due to the addition of data from two aircraft during the ACLOUD campaign and two flights from the AFLUX campaign, the acquired pyranometer-data is twice as extensive as the albedometer measurements (7964 versus 3936).

The highest inaccuracy of retrieval occurs in the near-infrared band: around 15 percent over-estimation for bare sea ice and snow-covered ice pixels. Apart from the larger positive bias inherent in the SciML algorithm (as compared to the visible band), the disparity in NIR values could be partly caused by the measurement uncertainty; the albedometer's overall uncertainty in the near-infrared wavelength band is reported to be the greatest (Jäkel et al., 2021a). It is also worth noting that even though the comparison is confined to low-level ($\leq$ 350 m) flight measurements, there still is a non-negligible height-induced error in the NIR values (refer to the RTM simulation findings in Fig. 2): On snow-covered sea ice, the error is - 4% (lower than the surface), which increases the gap between the retrieval (of ground) and the measurements (of low-altitudes). For open water, the near-infrared albedo at 350 m is + 16%.

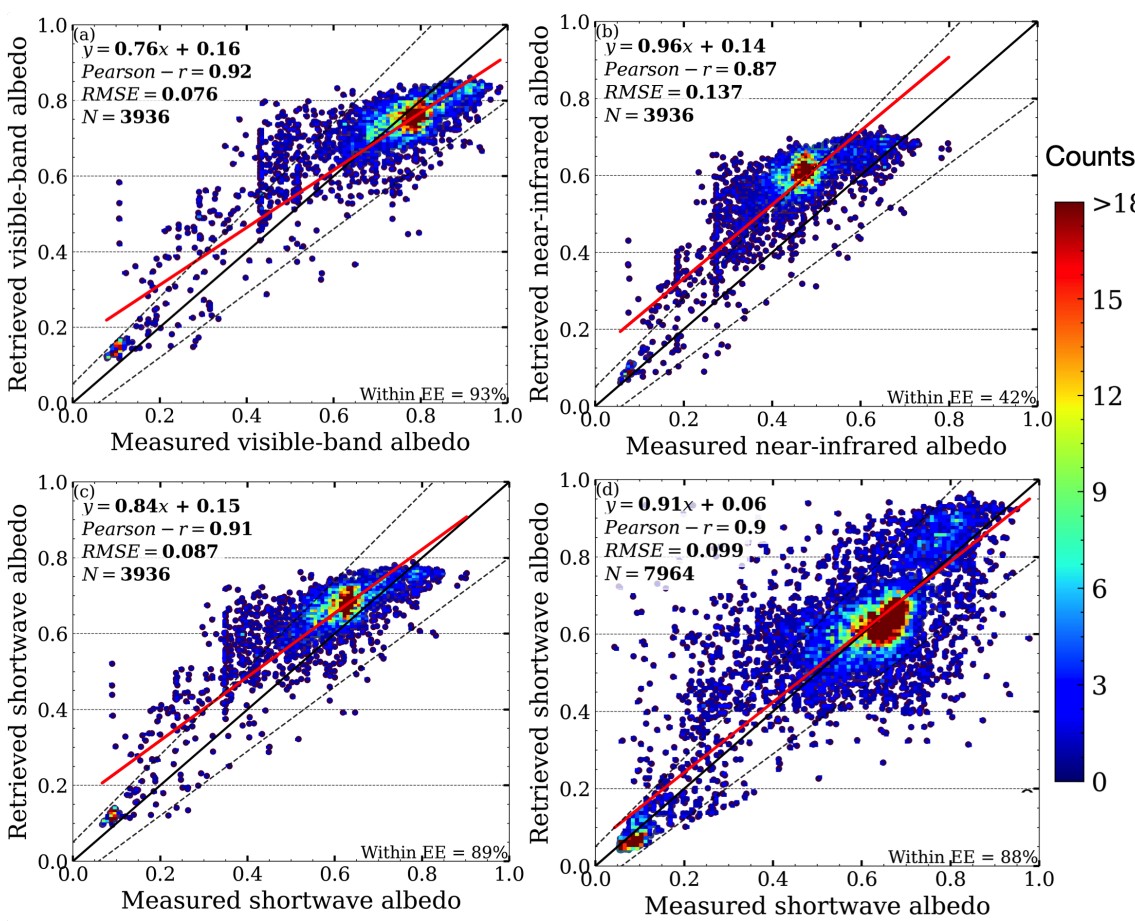

**Figure 4.** Correlation between ACLOUD-AFLUX albedo measurements and MODIS overflight. The comparisons with the albedometer in the VIS, NIR, and SW ranges are shown in panels (a)-(c), whereas panel (d) shows the comparison with the pyranometer. The wavelength-range difference between the two instruments and the corresponding retrieval models can be found in Table 4. The color scheme indicates the density of the numbers. On the figure panels, the correlation equation ($\hat{y} = a \cdot y + b$), Pearson $r$ coefficient ($r$), root mean square error (RMSE), the percentage of data within a 15% expected error ($f_{EE \leq 15}$), and number of pixels ($N$) used to compute the statistics are provided. The red lines represent the linear correlation between $\hat{y}$ and $y$; the solid black lines represent $(0,0) - (1,1)$, and the dashed black lines represent the 15% error range.

Overall, as illustrated in Fig. B2, the correlation coefficients for snow-covered ice and open-water are relatively high across the wavelength ranges ($r = 0.75$ and $r = 0.81$, respectively), whereas the correlation coefficients for bare sea ice and ice with melt-pond coverage are lower ($r = 0.54$). The reason is that in contrast to the more stable glacier, sea ice is more variable. For the days covered, the mean time discrepancy between MODIS transit and flight measurements is about two hours, with a maximum time difference of three to five hours (4 h, 4.85 h, and 3 h for May 31, June 8, and June 25, respectively).

When a time window of 1.5 hours was used for time restriction, a greater degree of agreement and smaller error was found; evaluating against pyranometer data (N = 4144) reveals $r = 0.92$ and RMSE = 0.094 for the shortwave albedo retrieval (Model 1), and evaluating with albedometer data (N = 1225) shows $r = 0.93, 0.89, 0.92$, RMSE = 0.069, 0.143, 0.085, in the visible, near infrared and shortwave bands, respectively (Model 2).

In our most recent work, the RTM/SciML-retrievals from the MODIS sensor in the Okhotsk region between 2002 and 2014 are validated using Soya icebreaker data. With a 1-hour time window (N=359), the RMSE is 0.097, and with a 3-hour time window (N = 911), the RMSE is 0.11.

## 3.2 Validation of SGLI retrievals

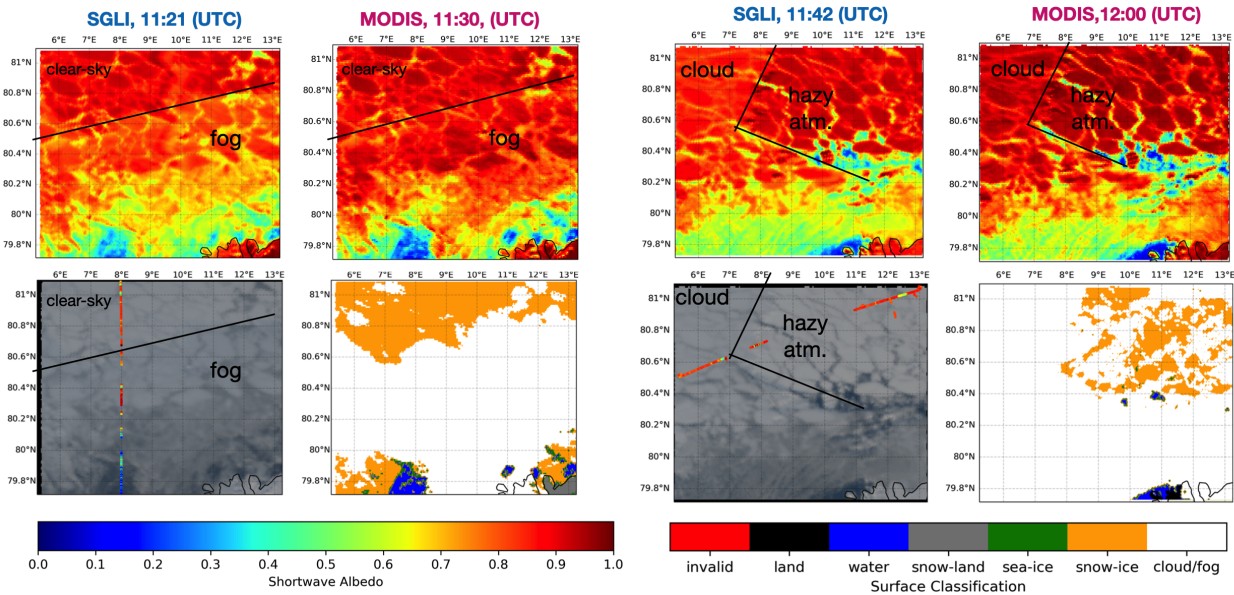

**Figure 5.** Albedo-retrieval maps, RGB maps, and surface classification maps from April 08, 2019 (left) and April 11, 2019 (right).The top-left column represents the albedo values retrieved with the SGLI sensor, the top-right column represents the albedo values retrieved with the MODIS sensor, the bottom-left column represents the spatial overlay of surface albedo measurements made on low-level ($\leq$ 350 m) flights (dots) on true color composite images, and the bottom-right column represents the surface classification results. The black lines represent the distinctions between various atmospheric conditions.

Frequently, fog (sea smoke) forms above sea ice and polynyas (Vihma et al., 2014). On April-08 2019, data taken south of 80.6°N were in such a condition. On April-11, a thick cloud with a sharp edge moved from 15°E at 5:50 UTC to 7°E at 12:25 UTC (Stapf et al., 2021b). As seen in the RGB and surface classification images (Fig. 5), the fog from April-08 and the cloud from April-11 was correctly identified by the MLCM for both sensors.

On April 11, the cloud-free area was seen to have a hazy atmosphere (Stapf et al., 2021b), and RTM models showed a thick aerosol optical depth of 0.065 (wavelength unspecified). The MLCM also detected the haze, which was classified in Fig. 5 as 'cloud/fog'. Figure 5 indicates that, even with the impact of cloud/fog/hazy atmosphere, the SGLI- and MODIS-retrieved albedo values ($\alpha$) for difference surface types are within reasonable ranges: melt pond ($\alpha \leq 0.3$), bare sea ice ($\alpha \approx 0.6$) and sea ice with snow coverage ($\alpha \geq 0.7$). When compared to data from low-level ($\leq 350$ m) aircraft at the same location and to

cloud-free MODIS retrievals (Fig. 5), the values are largely consistent.

     The scatter plot in Fig. 6 illustrates the correlation between the measured and retrieved albedo (under clear-sky conditions on April 8) using SGLI-channel and MODIS-channel radiances. Both results were derived with the same retrieval methods as those outlined in Section 2. For retrieval using SGLI data, 82% of the data were under the 15% expected error (EE), demonstrating a higher degree of agreement than the results produced from MODIS radiance data for this date. Correlation coefficients of

0.984 and 0.892, as well as RMSE of 0.082 and 0.136, indicating that RTM/SciML models can produce satisfactory results for cryosphere surface albedo retrieval.

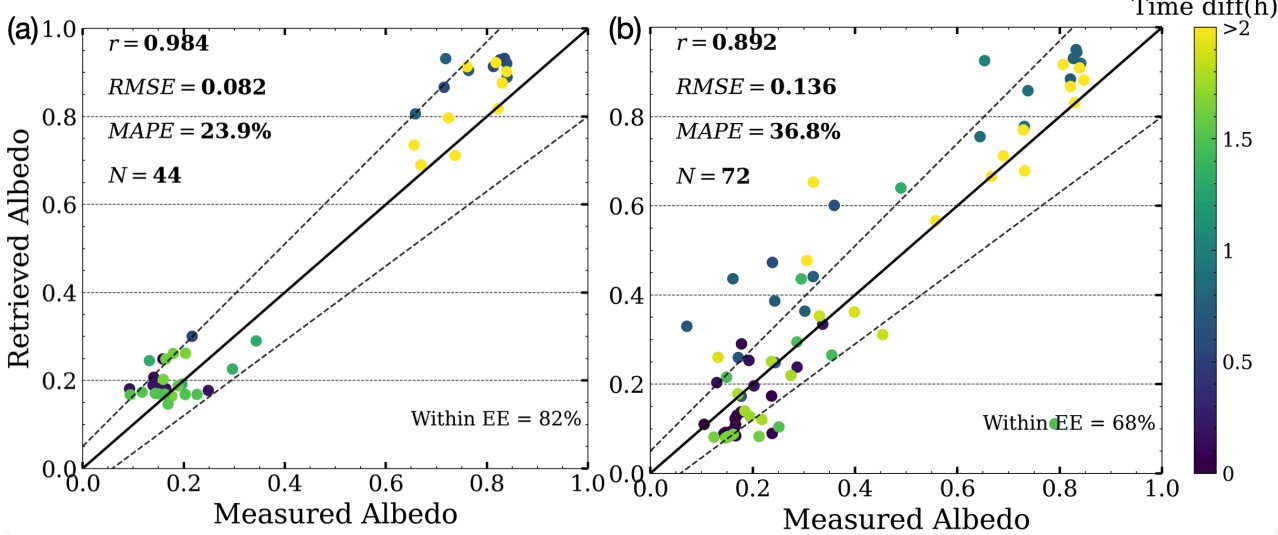

**Figure 6.** Correlation between shortwave albedo measurements from the AFLUX campaign and satellite overflight (SGLI (a), and MODIS (b)), April-08 2019. The color indicates the time interval between campaign measurements and satellite overpass. On the figure panels, the Pearson $r$ coefficient ($r$), root mean square error (RMSE), mean absolute percentage error (MAPE) and the number of pixels ($N$) used to calculate the statistics are provided.

## 4 Discussion on the source of validation error

Independent of the surface type or spectral range, all of the evaluations outlined above are prone to uncertainty due to equipment error, cloud contamination, surface metamorphism (drift and melt/refreeze), as well as differences in observation height, footprint, and solar zenith angle (SZA).

In the following discussion, we will break down and analyze the errors by their sources. The data were subjected to a more stringent temporal constraint of $\delta_t = 1.5$-h in order to allow for a more precise characterization of specific sources of inaccuracy. Figure 7 depicts the % difference between the observed and retrieved shortwave albedo when a maximum time difference between aircraft and satellite of 1.5 hours is allowed. Still, the percentage difference between the RTM/SciML-estimated values and pyranometer measurements are smaller than those with albedometer measurements. Meanwhile, the error for the open-water surface (flight segments within the dashed red line) is significantly larger when compared to the albedometer data than when compared to the pyranometer data.

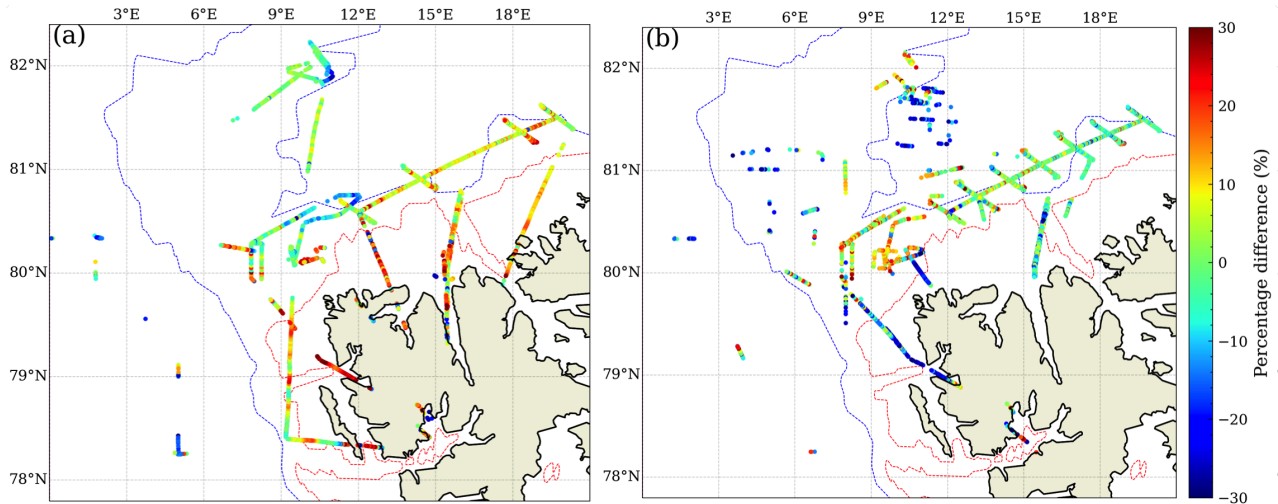

**Figure 7.** A map illustrating the percentage difference between the RTM/SciML-estimated broadband albedo and the albedo measured in-situ using (a) the "SMART" albedometer, and (b) the pyranometer. To choose the matching MODIS retrieval, a tight time constraint of 1.5 hours ($\delta_t \leq 1.5$-h) was applied. On the right, the colorbar indicates the scale of the % difference value for each pixel ($\frac{\hat{y}-y}{y} \cdot 100\%$). Additionally, red and blue dashed lines represent the 200-meter and 1000-meter isobaths. Each flight segment's dates are listed in Fig. 3.

### 4.1 Distinction in the footprint of observation

When aircraft measurements are up-scaled to MODIS footprints, around 170 aircraft measurements were taken to match one satellite pixel. According to prior research, an albedo line 100 meters long surrounding melt ponds would have a standard deviation of roughly 0.4 (Perovich, 2002), while the albedo of thin ice inside a grid cell would have a standard deviation of up to 0.29 (Lindsay, 2001). Due to the fact that measurements along the aircraft transit do not accurately reflect the area's average

albedo (as determined by satellite sensors), the uncertainty introduced by different observation footprints (i.e. subpixel effects) is not negligible but difficult to quantify. On a homogeneous surface such as fresh snow, subpixel effects could be small, but for the heterogeneous surface consisting of a snow-ice-water mixture, the influence is large.

## 4.2   Cloud contamination effects

While cloud screening eliminates cloudy pixels in satellite data, data acquired during airplane flights are not corrected for cloud cover. Given that more than one-third of the operational spatial range is obscured by clouds, it is possible that some pixels labeled as 'clear sky' by the MLCM are cloud-contaminated as a result of (i) a time mismatch between the MODIS overpass and aircraft data collection, or (ii) simply imperfect cloud screening. Multiple reflections between the surface and cloud base would introduce uncertainty into the albedo measured by the airplane.

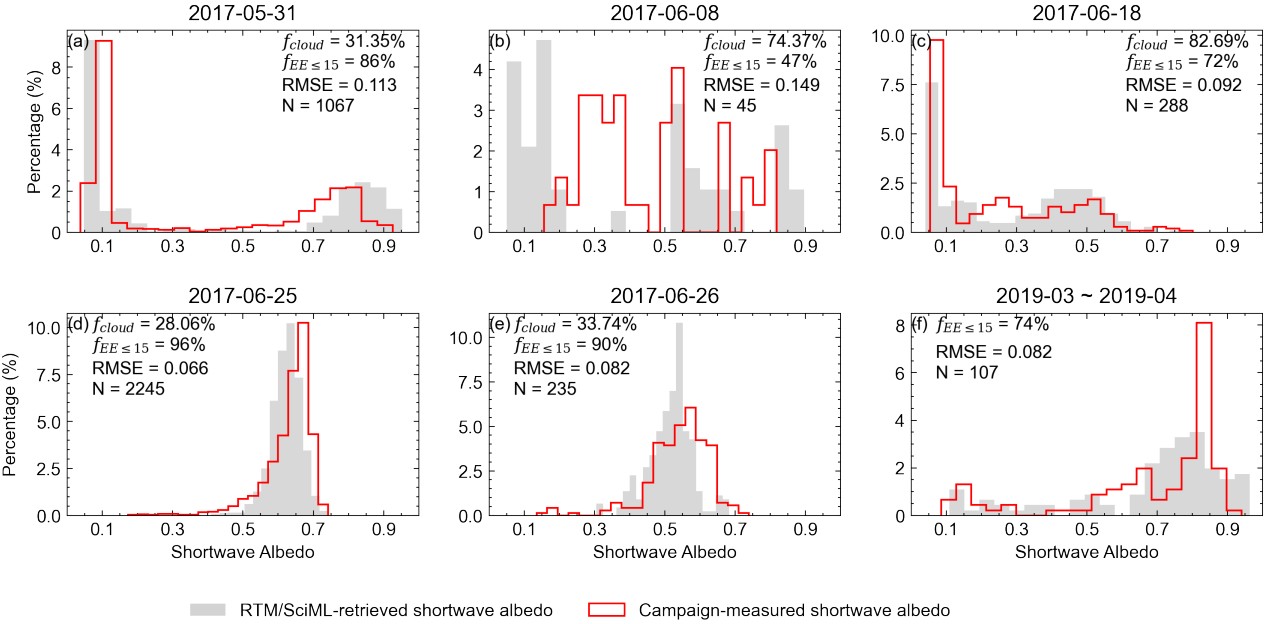

**Figure 8.** Histograms of the shortwave albedo measured during the cloud-free ACLOUD-AFLUX flights (red columns) and the RTM/SciML's corresponding retrieval (gray columns). To choose the matching MODIS retrieval, a tight time constraint of 1.5 hours ($\delta_t \leq 1.5$-h) was applied. The dates are displayed at the top of each panel. Additionally, the upper right corner displays the cloud fraction ($f_{cloud}$), the percentage of data within 15% expected error ($f_{EE \leq 15}$), the root mean squared error (RMSE), and the number of pixels for each evaluation (N). Due to a lack of appropriate data, all AFLUX measurements were combined together for examination, as illustrated in panel (f).

As illustrated in Fig. 8, the agreement is negatively correlated with cloud coverage, with the exception of panel (b). On June 8, 2017, the date depicted in panel (b), a considerable decrease in temperature and an increase in liquid water vapor (IWV) were recorded (Wendisch et al., 2019). Following a day of dense cloud cover, the surface temperature measured on ice floe

camps increased by 2 degrees on June 10, 2017 (Barrientos Velasco et al., 2018). A Magna probe (Sturm and Holmgren, 2018) recorded that snow depths on the ice floe on June 14 were down to 22±18 cm (with 32% of data below 10 cm), while on June 5 the snow depth was 37±24 cm (with 9% of data below 10 cm) (Jäkel et al., 2019). These data indicate that the surface conditions changed radically around June 08, which possibly led to the larger root mean squared error (RMSE of 0.149). The RMSE is all minor on the three days with significantly less cloud cover (panels a, d, and e). The best agreement occurs on June

25, when an analysis of 2245 accessible MODIS pixels reveals an error of only 0.066; 96 percent of the data are within an error of 15%.

  The spatial overlay of observations from June 25, 2017 on a retrieval image utilizing MODIS's 12:45 UTC transit is shown in Fig. 9. Snow-covered ice, open water, and bare ice albedo values are all within acceptable limits and are in good agreement with albedometer/pyranometer determinations.

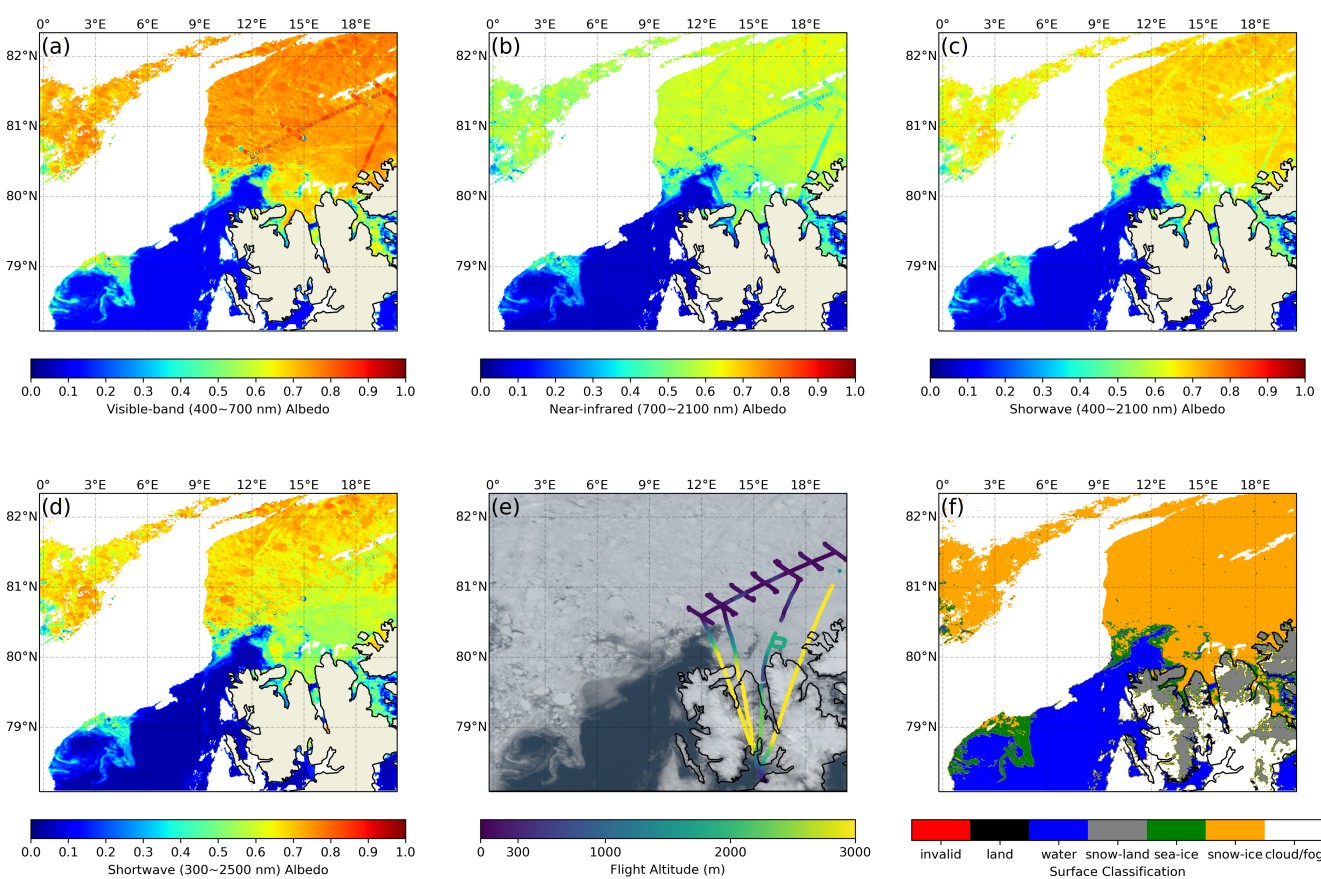

**Figure 9.** (a)-(c): spatial overlay of albedometer measurements on top of the RTM/SciML-derived albedo in the VIS, NIR, and SW ranges from 12:45 UTC, June 25 2017; (d): spatial overlay of pyranometer measurements on the RTM/SciML-derived SW albedo; (e): spatial overlay of aircraft altitudes on the true color composite; and (f): surface classification results. Cloud-contaminated pixels (shown in white in (f)) were removed during the retrieval.

## 4.3 Albedo variations caused by changes in the solar zenith angle

Fortunately, three consecutive days during the campaign, the latitude-longitude range experienced of clear skies, from June 24 to June 26, 2017. The data from four MODIS overflights per day (see Table 5) can be used to determine changes in surface albedo caused by changes in solar zenith angle and surface metamorphism.

Visual examination of the RGB images of the eight MODIS transits from June 24 to June 25 revealed no apparent ice drift during the two days (refer to the accompanied gif image in the supplementary files). Following this, we employed the MLCM to filter out regions that were cloud-contaminated during any of the four MODIS transits on June 25. In other words, assuming there is no ice drift, the changes in albedo value over the period of four satellite transits are due to either surface metamorphism or a shifting solar zenith angle.

The probability density functions (PDFs) of the surface albedo in the region shown in Fig. 9 were produced using the selected data (Fig. 10(a)). A Gaussian kernel probability density (Turlach, 1993) was used to estimate the PDF. The probability of albedo values falling within the interval $P(a < x \leq b)$ is measured by subtracting the two integral values $P(a < x \leq b) = F_x(b) - F_x(a) = \int_a^b f(x)dx$, where $F(x)$ is the cumulative density function (CDF) and $f(x)$ is the PDF. The area between the PDF-curve and the $x$-axis is equal to one (Scott, 2015). The bimodal distribution with one mode at 0.1 and the other at 0.65 indicates that the most prevalent surface types are water and snow-covered sea-ice, with bare ice accounting for a small portion of the data.

We noted the difference in SZA over the four MODIS transits, from 55~59 at 11:05 UTC to a higher value of 61~65 at 16:00 UTC, and the difference increased the albedo for snow and water by 6% and 30%, respectively, while the values on the bare-ice surface remained relatively stable (5% fluctuations).

The variations in albedo caused by increasing SZA were investigated in RTM simulations (see insert table in Fig. 10(b)). The relevant parameters for the simulation are: snow thickness $h_s = 0.2$ m, effective grain size of snow $r_e$=100 $\mu$m, pond depth $h_w = 0.1$ m, and black carbon impurity fractions for snow and ice $f_{bc} = 1.0 \times 10^{-7}$. Although the difference in broadband albedo values indicates that the snow, ice, and water conditions in the RTM simulations do not match the surface conditions at this location, they may serve as a reference for the portion of the albedo difference caused by SZA, which is 2.7%, 50% and 4% for snow-covered ice, water, and bare-ice, respectively. With these numbers in mind, we can conclude that the lower variance in snow/ice surface albedo (despite increasing SZA) and the less noticeable increase in water albedo are both related to surface melting.

## 4.4 Changes in albedo as a result of surface metamorphism

To analyze the fluctuation in albedo caused by surface metamorphism (intra-daily variation due to snow melting, pond development, or pond drainage), we picked the 'consistently clear' latitude-longitude coordinates throughout the eight MODIS overflights over a two-day period. The pixels that have been filtered out due to cloud cover happen to be snow cases. As a result, the following analysis focuses on the phase transitions at the marginal sea ice zone, which is a system composed of bare ice thicker than 30 cm (typical albedo values between 0.4 and 0.5, Brandt et al. (2005); Petrich and Eicken (2009)), melt ponds

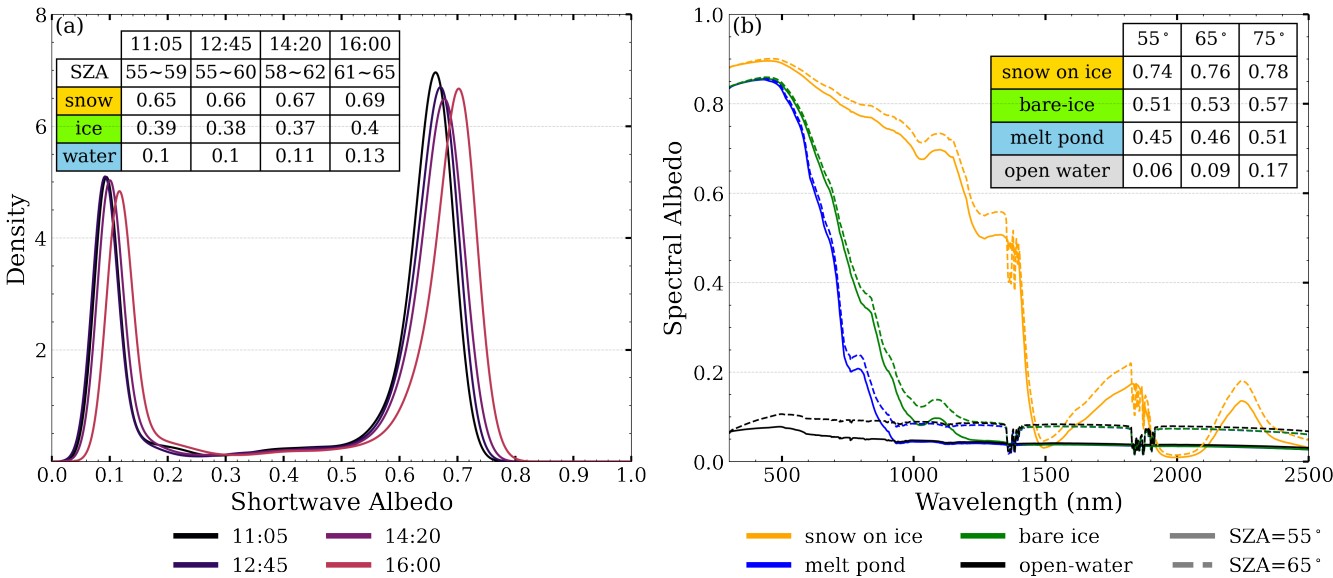

**Figure 10.** (a) Probability density curves of albedo on June 25. The colors denote the retrievals from several MODIS overflights. The average albedo estimated for each surface is provided in the upper left table, along with the solar zenith angle (SZA) ranges. (b) RTM-simulated spectral albedo of snow, melt-pond, bare-ice and open-water surfaces; dashed and solid lines indicate spectral albedo with varying SZA. The table to the right shows the calculated shortwave albedo of each (surface, SZA) pair.

(typical albedo values between 0.2~0.4, Grenfell and Maykut (1977); Grenfell and Perovich (1984); Grenfell (2004); Petrich and Eicken (2009)), and open water (typical albedo below 0.2, Toyota et al. (1999); Petrich and Eicken (2009) ).

Figure 11 depicts the distribution of albedo values at the eight MODIS overflights. Comparing the distributions of $\alpha_{24}$ and $\alpha_{25}$ at the same SZA (i.e. the first two columns of Fig. 11), the histograms appear to have a similar shape. However, the change in albedo at a fixed location over about 24 hours ($\Delta_\alpha$, the third column of Fig. 11) can be as significant as 0.4. As no apparent ice drift was observed and the location was at the intersection between open water and sea-ice regions, this circumstance indicates that the surface was undergoing intermittent melting and refreezing, akin to the situation of a polynya.

To summarize, in Sections 3-4, we evaluated the RTM/SciML retrieval results using *in-situ* measurements. The uncertainties associated with the evaluation were explained in detail, per source of error, in order to evaluate their impact. Despite these uncertainties, the current technique for albedo retrieval, which is based on (1) an AccuRT-generated SD and (2) a SciML model trained using the SD as prior knowledge, can indeed produce reasonable albedo outputs, with a RMSE = 0.094 evaluated on over 4000 pyranometer measurements under clear-sky conditions.

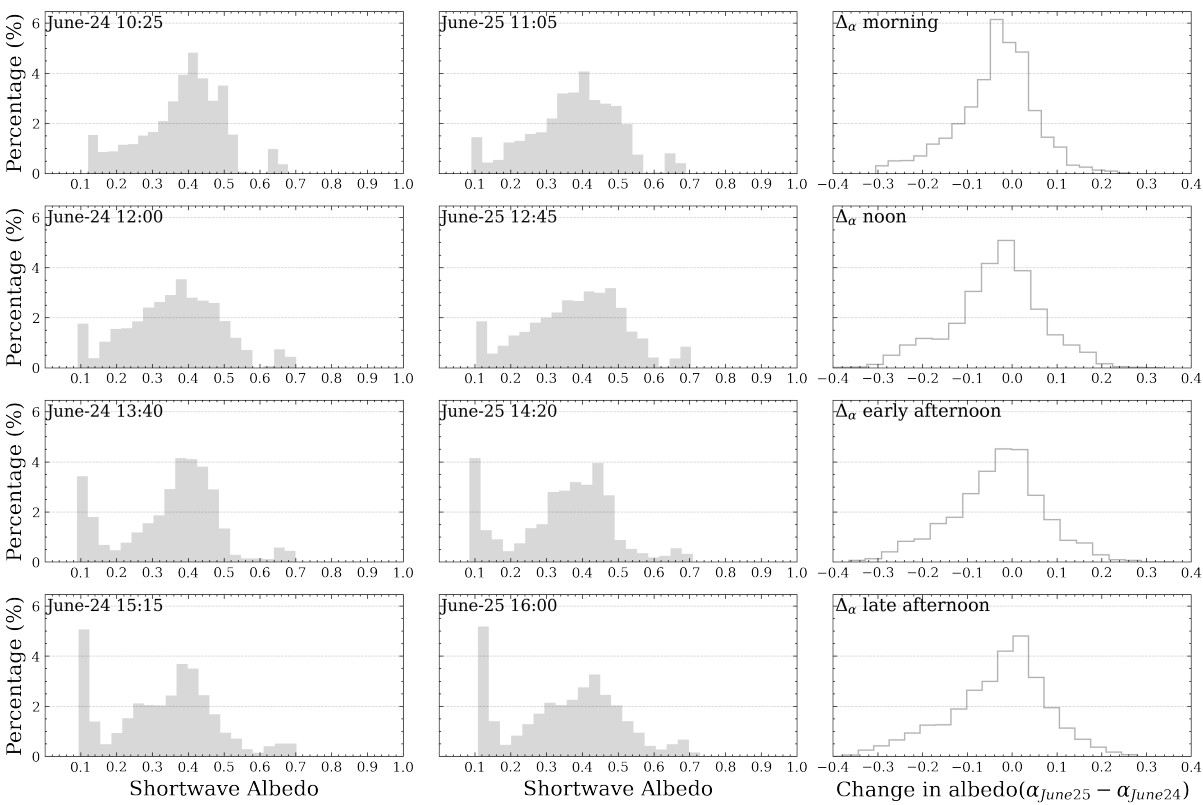

**Figure 11.** Histogram of the distribution of albedo values observed during the eight MODIS overflights on June 24 ~ 25. The albedo histograms for June 24 ($\alpha_{24}$), June 25 ($\alpha_{25}$), and the difference ($\Delta_\alpha = \alpha_{25} - \alpha_{24}$) are shown in the columns from left to right. From top to bottom, the rows represent the four satellite overpasses that occurred throughout the day: in the morning, at midday, in the early afternoon, and in the late afternoon.

## 5   Comparison of RTM/SciML retrievals with the existing methods

In this section, the RTM/SciML-albedo retrievals are compared with the products listed in Table 1. A brief description of the relevant products and sensors are provided in Section 2.5.4.

### 5.1   Comparison with the MCD43D49∼51 BSA products

MODIS MCD43 is a land albedo product; only a small amount of sea ice surface albedo is available near the shore (in the 'shallow ocean' zone denoted by the BRDF/Albedo Quality Product, MCD43A2, (Schaaf and Wang, 2015a)). While prior research has validated the albedo product for glacier, tundra, and snow-covered land surfaces, the small amount of sea ice albedo on the shallow ocean has not been validated previously (Ren et al., 2021; An et al., 2020; Pope et al., 2016; Wang et al., 2012). Using ACLOUD-campaign measurements, the MCD43's reliability in the sea ice zone may be assessed.

On the same days, Figs. 12 and 13 compare the RTM/SciML-derived broadband albedo values utilizing MODIS TOA radiances (i.e. MOD021KM) as input to the MCD43-derived BSAs. The RTM/SciML-derived albedo values represent the average of all available clear-sky pixels observed across four MODIS transits within the day, whereas the MCD43 product is representative of the albedo at local noon (Stroeve et al., 2005). Therefore, values from the MCD43 product are expected to be slightly lower than the RTM/SciML results due to the relatively lower solar zenith angle ($52 \sim 55°$). We need to highlight that the two comparisons are made at different wavelengths (the NIR/SW's upper bounds for RTM/SciML- and MCD43-albedo are 2.8 $\mu$m and 5 $\mu$m, respectively) and with different albedo assumptions (blue-sky and black-sky albedo, for RTM/SciML-albedo and MCD43-albedo, respectively). All albedo maps include superimposed SMART albedometer measurements from the ACLOUD campaign (Jäkel et al., 2018) as reference. Note that the flight segments that overlapped with MCD43D retrievals in Fig. 12 were at an altitude of $h \geq 2500$ m, indicating that the validation data on open water surface have a 33~150% positive bias relative to the surface albedo values; the biases for bare-ice and snow-covered ice surface is -10~-7% and -22~-8%, respectively (see flight heights in Fig. 9(e) and the height-induced bias in Fig. 2).

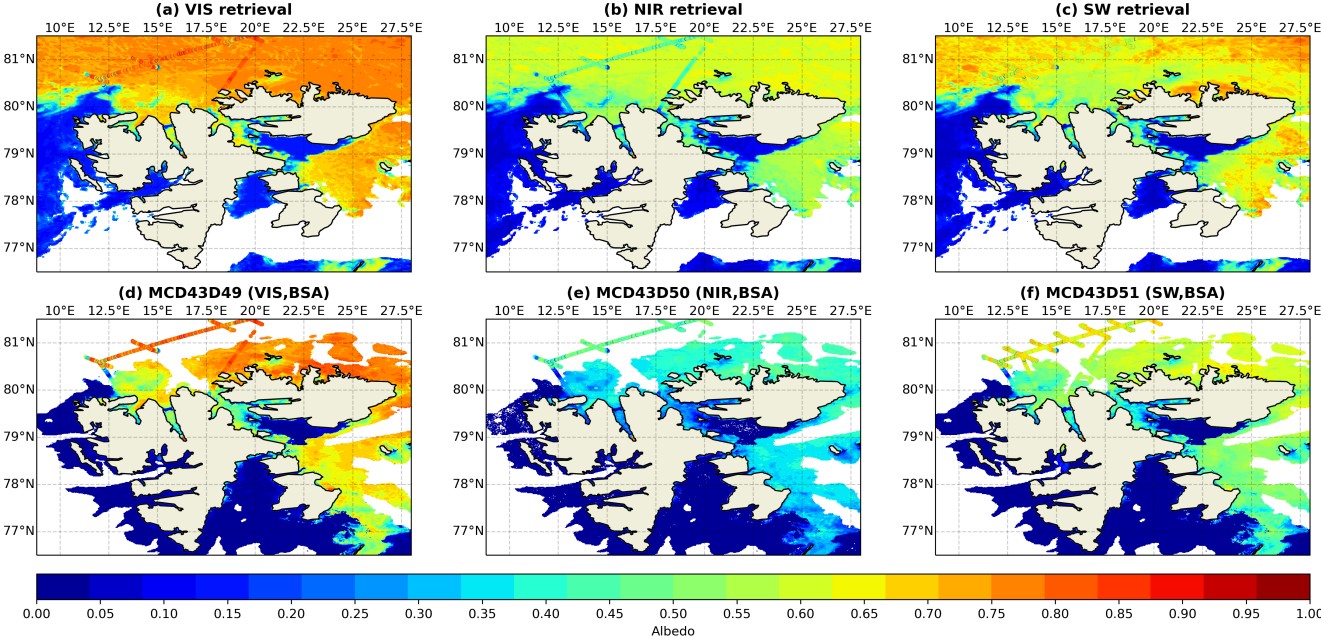

**Figure 12.** Albedo-retrieval maps on June 25, 2017 in the shallow ocean region near Svalbard. The albedometer measurements from the ACLOUD campaign are superimposed over the retrieval maps in each panel. Note that both low- and high-level flight measurements are included. (a)-(c): the visible, near-infrared, and shortwave (VIS, NIR, SW) albedo values derived using the RTM/SciML-albedo product and MODIS radiance data; (d)-(f) the three broadband black-sky albedo values (BSAs) derived by MCD43D49-51. The empty regions in panels (a) to (c) correspond to cloud coverage throughout the day on June 25, 2017, whereas the empty regions in panels (d) to (f) correspond to ocean areas where MCD43 does not provide retrievals.

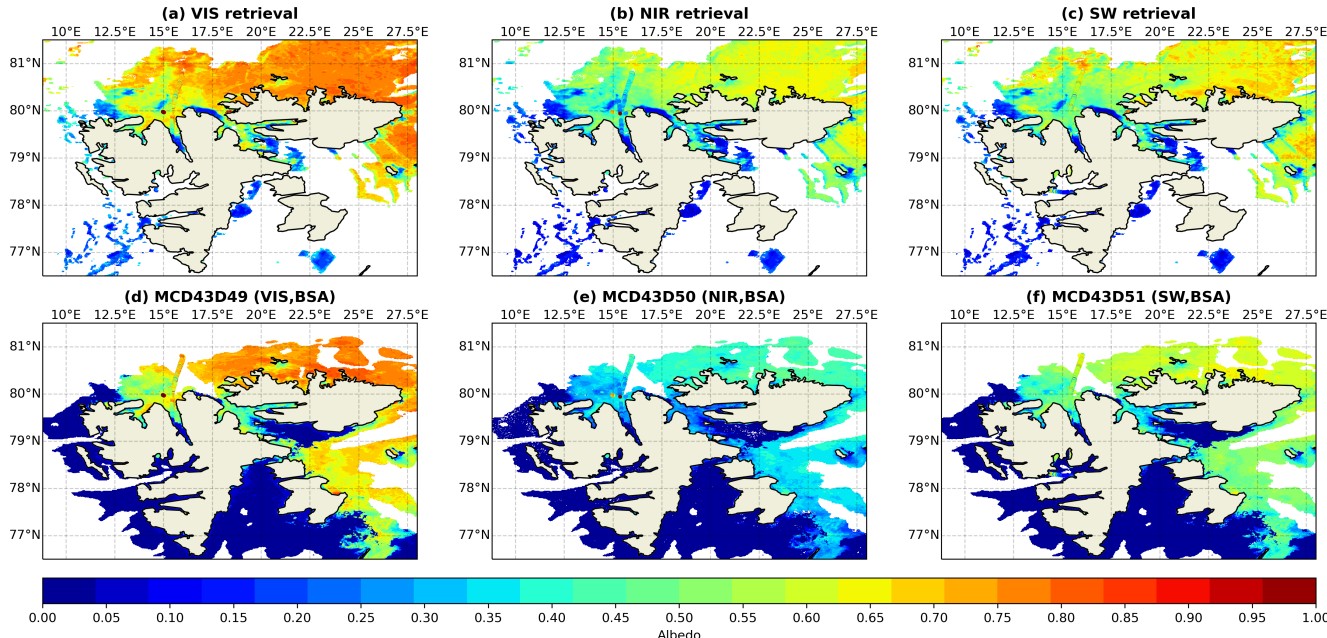

**Figure 13.** Albedo-retrieval maps, similar to Figure 12 but on June 26, 2017.

Because the BRDF is calculated using observations from a 16-day window, the MCD43 result cannot adequately capture daily albedo variations. One example is its failure to detect the opening of a melt pond north of Svalbard on June 25 and 26, located at 15°E, 80°N. According to the eight RGB MODIS transit images (not shown), the melt pond began to open at 12:45 UTC on June 25, 2017, and by 15:05 UTC on June 26, 2017, the ice underlying the pond had entirely melted, leaving some bare sea ice (as categorized by the MLCM) surrounding the open water. While MCD43 could not identify this opening, the daily averaged RTM/SciML-albedo indicates the snow and ice melting that resulted in the formation of a small open water region.

Another point to emphasize is the lower albedo values obtained by MCD43 on snow-covered ice when compared to the RTM/SciML results, as demonstrated by the areas north and south of Nordaustlandet (near 22.5°E, 80.5°N and 25°E, 78.5°N, respectively). While there are no direct measurements to verify these values, we note that the underestimation of snow-covered area has been mentioned in several previous studies. For example, Stroeve et al. (2005) discovered that the MCD43 retrieval for snow-covered Greenland Icesheet (with an albedo greater than 0.7) has a -0.05 bias when compared to ground-based measurements. Similarly, An et al. (2020) observed underestimation when the albedo is greater than 0.4 for ice caps. For the particular case observed on these two days, the main reason is the extensive melting during the warm period (Knudsen et al., 2018).

Although the MCD43D V6.0 products have been adjusted to better capture shorter-term albedo variations through adjusting the BRDF weighting scheme to emphasize the BRDF of the day of interest inside the 16-day sliding window, by examining

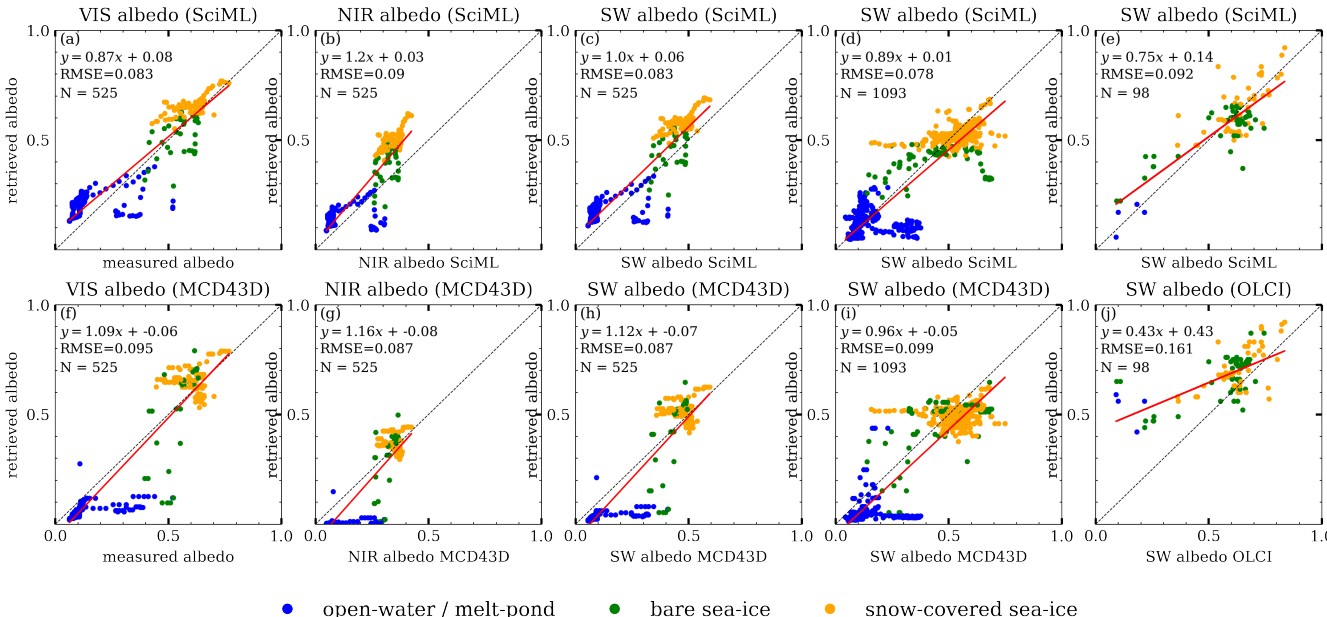

**Figure 14.** Correlation between satellite retrievals and ACLOUD albedo measurements utilizing three retrieval methods. (f)-(h): MCD43D albedo in the visible, near-infrared, and shortwave broadbands compared to albedometer measurements, (i): MCD43D shortwave albedo compared to pyranometer measurements, and (j): Comparison of the OLCI shortwave albedo derived with the MPD algorithm to all shortwave measurements from the ACLOUD campaign. (a)-(e): RTM/SciML retrievals that are regridded to the MCD43D or OLCI pixels. Surface type is represented by color. The regression relation between retrieval and measurements, RMSE and the number of valid pixels are placed at the top left of each panel. Other statistical metrics (R-squared, bias, $f_{EE \leq 15}$ etc.) can be found in Table B3 in Appendix.

the Quality Product (i.e. MCD43A2), we found a much higher 'BRDF albedo uncertainty' marked in the melting-snow areas as compared to the surrounding open-water. According to Lucht and Lewis (2000); Wang et al. (2012), the uncertainty can be
470 related to fluctuations in surface properties, atmospheric correction errors, high solar zenith angle ($>65°$), and cloud detection during snow melt, among others. Figure 14 and Table B3 provide an approximation of the bias imparted by such uncertainty to sea ice albedo: compared to the 'ground truths', the MCD43D retrievals on open water and melt-pond are significantly lower, and compared to the RTM/SciML retrievals at the same pixel, the MCD43D snow retrievals are more dispersed.

## 5.2 Comparison with the MERIS-albedo product

475 Figure 15 compares the retrieval results obtained using the MERIS- and RTM/SciML-albedo retrieval algorithms during a 5-day period in 2007 between DOY 166 and 170. Additionally, the albedo values from CLARA-SAL product (Karlsson, Karl-Göran et al., 2012) and the melt pond fraction (Istomina et al., 2015) are given as reference.

  The challenge of using the MPD algorithm to retrieve albedo is that the algorithm relies on certain empirically-derived criteria (based on ratios of some radiance channels) to determine the exact type and composition of the corresponding pixel in

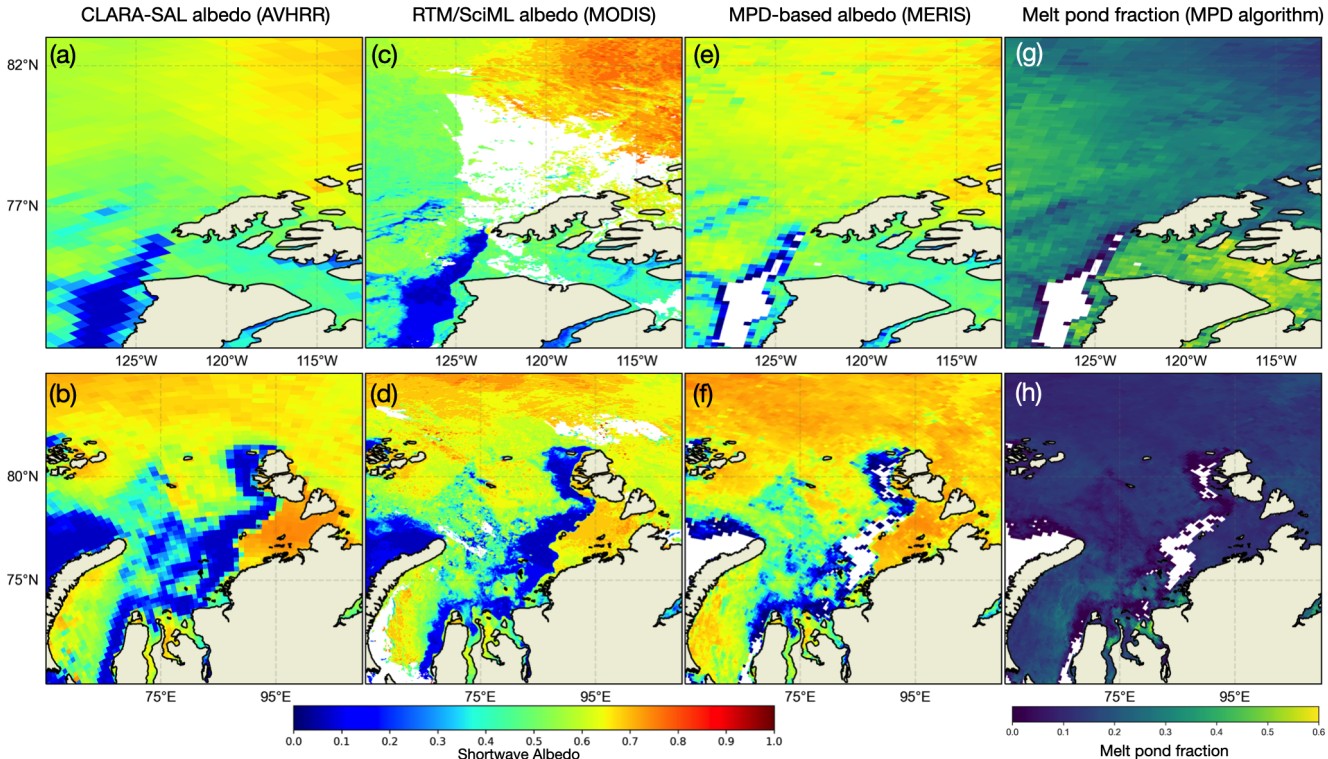

**Figure 15.** Maps of albedo and melt pond fraction averaged during a 5-day period in 2007 between DOY 166 and 170. From left to right: CLARA-SAL albedo, RTM/SciML-based and MPD-based albedo retrievals, as well as the MPD-derived melt pond fraction, respectively (Karlsson, Karl-Göran et al. (2012) this study, and Istomina et al. (2015)). The upper panels depict the Kara Sea, while the lower panels depict the Banks, Prince Patrick, and Melville Islands. At the bottom, colorbars representing the corresponding values are displayed. In panels (c) and (d), empty regions represent cloud pixels that were detected by the MLCM cloud mask (and hence removed), whereas empty regions in panels (e) through (h) represent either cloud pixels or open-water areas that were not processed by the MPD algorithm.

order to assign an appropriate BRDF value for the valid, or filling value for invalid, observations. Since the assigned BRDF is used to derive albedo, the surface type was explicitly designated prior to albedo estimation. Moreover, the spectral reflection coefficients for the melt-pond and thin ice boundaries, as well as the thick ice and snow-cover boundaries, are manually adjusted based on the surface condition. Therefore, there are greater uncertainties in the retrieval during the transitional seasons of spring-summer and summer-autumn, as well as when the surface is highly heterogeneous (low sea ice concentration, discussed

in Istomina et al. (2015)); misclassification or improper manual assignment would result in a considerable uncertainty in the final albedo estimate.

As shown in Fig. 16, the two algorithms produce more consistent results for regions with low melt pond fraction, $f_{mpf}$, i.e. high sea ice concentration, $f_{sic}$; the average albedo values are similar when $f_{mpf} \leq 30$. However, the MERIS-derived albedo values in regions with large $f_{mpf}$ values($f_{mpf} \geq 40\%$) appear to be higher than those produced by the RTM/SciML model.

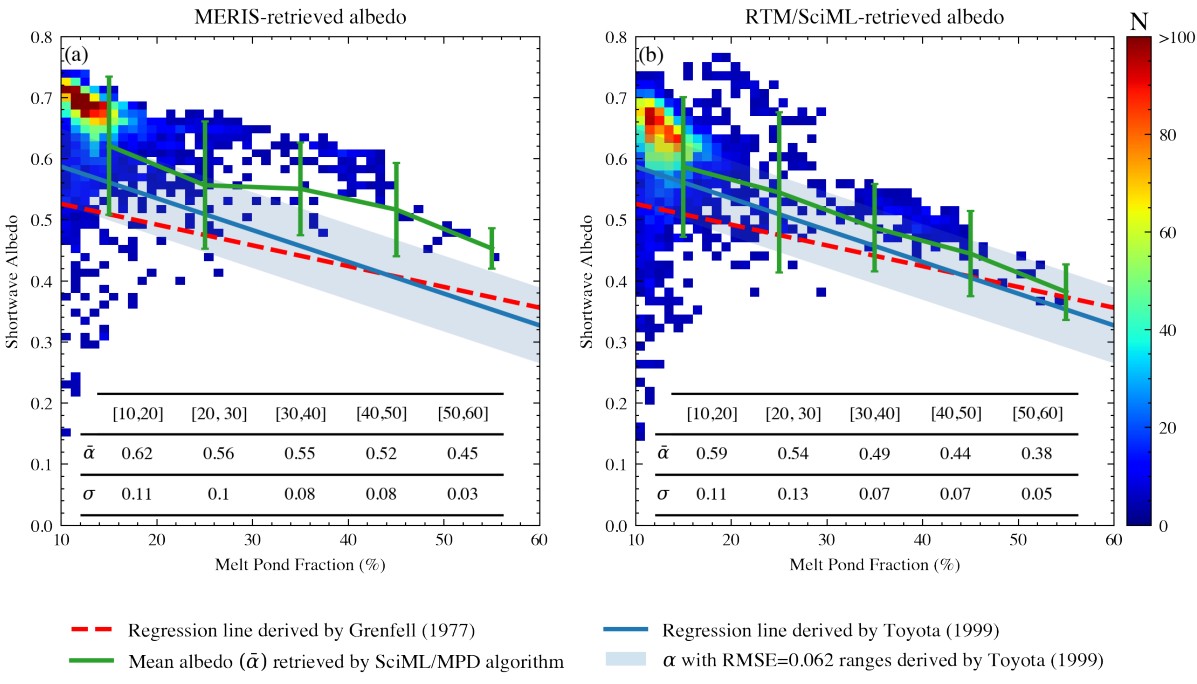

**Figure 16.** Density plot of the retrieved albedo in proportion to the melt pond fraction ($f_{\mathrm{mpf}}$); values are derived from the regions depicted in Fig. 15. The color scheme represents number density. The inset table includes the mean ($\bar{\alpha}$) and standard deviation ($\sigma$) of albedo as retrieved by the SciML/MPD algorithm in different $f_{\mathrm{mpf}}$ intervals. The values are also represented by the green line in (a)-(b). The red lines represent the linear relation between $f_{\mathrm{mpf}}$ and sea ice albedo derived by Grenfell and Maykut (1977). The blue lines and shadings represent the $f_{\mathrm{mpf}} - \alpha$ relation and the RMSE of 0.062 derived by Toyota et al. (1999).

Grenfell and Maykut (1977) derived a linear relation between $f_{\mathrm{mpf}}$ and the surface albedo ($\alpha$), which is representend by the red lines in Fig. 16. Toyota et al. (1999) found a linear relation between ice concentration and albedo. In the absence of leads, the sea ice concentration and melt pond fraction has a $f_{\mathrm{sic}} + f_{\mathrm{mpf}} = 1$ relation. The shaded blue area and blue lines in Fig. 16 denote the $f_{\mathrm{mpf}} - \alpha$ relation derived by Toyota et al.. Similarly, Petrich and Eicken (2009) reported reference albedo values of $> 0.6$ and $< 0.5$, respectively, for sea ice with 10% and 50% areal pond coverage. The sea ice albedo values produced by

the RTM/SciML model correspond more closely to the reference values obtained by Toyota et al. (1999); Grenfell and Maykut (1977); Petrich and Eicken (2009), whereas the values retrieved by the MPD algorithm are overestimated.

Another difference to note is the empty areas in Fig. 15. Chen et al. (2018) and Fan et al. (2021) demonstrated that the MLCM model we used for cloud filtering is more strict than the MODIS cloud filtering algorithm and is capable of identifying very thin cloud (and even fog, see Section 3) on bright surfaces such as snow and ice. The empty areas in (c) and (d) are MLCM-

identified thin cloud pixels, whereas the empty areas in (e) through (h) represent open-water areas that were not processed by the MPD algorithm (Istomina et al., 2015).

## 5.3 Comparison with the OLCI-albedo product

Figure 17 depicts OLCI- and RTM/SciML-albedo retrievals on June 24, 2017 and June 26, 2017. Additionally, broadband albedo values obtained using pyranometers Stapf et al. (2019) are displayed on top of the RGB images as reference.

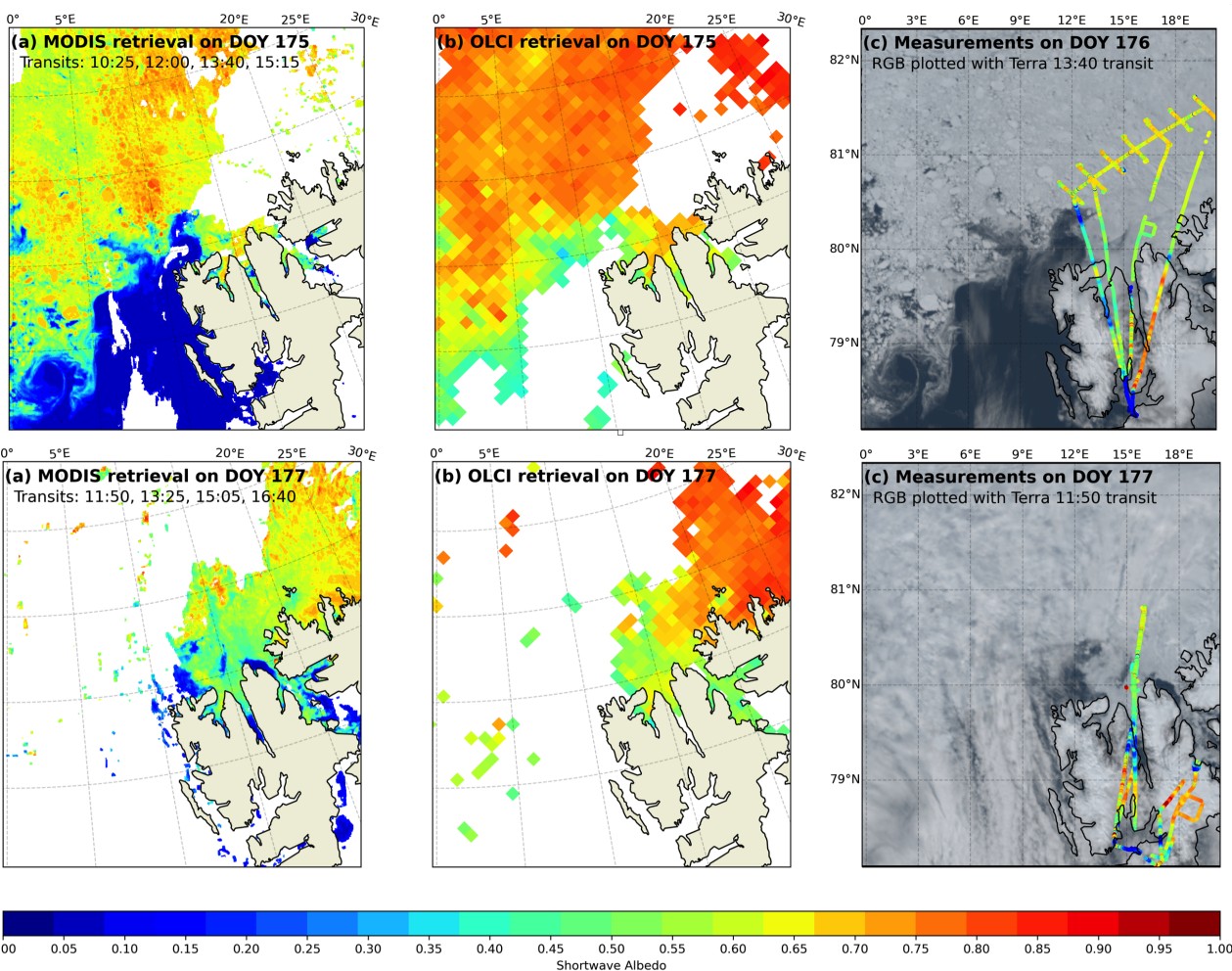

**Figure 17.** Maps of albedo derived using RTM/SciML (MODIS data) and the MPD algorithm (OLCI data, Istomina (2020)). The top panels depict albedo values on June 24, 2017, while the bottom panels depict values on June 26, 2017. From left to right, panels depict retrievals using RTM/SciML, MPD, and pyranometer measurements layered on an RGB image. On the associated images, the MODIS transits used to retrieve albedo and plot the RGB are labeled. Notably, no OLCI retrieval was available on June 25, 2017, and the campaign measurements for reference were taken on June 25 and 26.

In the MPD algorithm, the type of ice used to calculate the ice-BRDF is referred to as 'white ice' (Zege et al., 2015), which forms when melt water drains intermittently from sea ice and has a few centimetres thick white coating that scatters light similarly to a thin snow layer (Grenfell and Warren, 1999; Tschudi et al., 2008).

Due to the logic of the underlying MPD algorithm, the OLCI- and MERIS albedo products place a premium on the melt pond areas in the two polar regions. Because of the algorithm's emphasis on summer melting ice, it has various limitations, including being only applicable for data gathered from May to August, having a limited ability to retrieve albedo on snow-covered ice and ice with brine pockets inclusions, omitting retrieval for open-water areas, and having restrictions in areas with low sea ice concentration and very thin ponded ice.

When OLCI albedo maps are compared to *in-situ* data (Fig. 17 (b)-(c)), it is clear that the derived albedo values for snow-covered ice are excessively high (0.75 to 0.82, versus 0.65 to 0.7). On June 26, 2017 (DOY 177), the RGB image shows a few open water spots around the northern coastline that are not captured by OLCI. This failure is partly attributable to the product's spatial resolution; data from its original resolutions (300 m at full resolution and 1.2 km at reduced resolution) were mapped to a 12.5-km NSIDC grid. However, the main reason is that the MPD employs pixels solely from sea ice grid cells, and that open water pixels have been filtered out using a brightness criterion. The error of OLCI retrieval is quantified by the scatterplots in Fig. 14(e)-(f) and the statistics (R-squared, RMSE, bias, etc.) in Table B3.

By comparison, the RTM/SciML-albedo algorithm is capable of retaining the surface's albedo truethfully regardless of its condition and accurately reflecting the albedo values of these surfaces (i.e. more consistent with the measured albedo values). The current MLCM tool does not discriminate melt ponds from bare ice, and labels both as 'sea ice'. Because the MPD algorithm employs reflectance values from three visible channels (442 nm, 490 nm, and 510 nm), we may use a similar criterion to derive the melt pond fraction from MODIS or perhaps other sensors as well in the future.

## 5.4   Comparison with the direct-estimation algorithms (GLASS and VIIRS SURFALB)

At the time this paper was written, the phase-2 GLASS surface albedo product acquired using the MODIS sensor (Li et al., 2018) had not yet been released; the V40 GLASS (MODIS) product listed on the product page covers just land surface, leaving ocean surface blank. Consequently, the V40 GLASS (AVHRR) is compared to RTM/SciML retrievals in the following section.

The ACLOUD campaign was conducted during the 2017 spring-to-summer transition, which marked the beginning of the melt season, with snow-covered sea ice predominating and a small percentage of melt ponds. Three distinct synoptic weather periods were defined: a Cold Period (May 23∼29, 2017), a Warm Period (May 30 to June 12, 2017), and a Normal Period (June 13∼26, 2017) (Wendisch et al., 2019; Knudsen et al., 2018). Throughout the Cold Period, surface conditions were relatively steady, with albedo values reaching 0.8 on snow-covered ice and approaching 0.6 on bare sea ice. During the Warm Period, the northwesterly wind pushed the Greenland coast ice southward, resulting in the creation of the Northeast Water (NEW) Polynya and the opening of the region north of Nordaustlandet (Wendisch et al., 2019). It is evident from Fig. 18 that the RTM/SciML model accurately represents the three synoptic periods, including the emergence of NEW Polynya during the Warm Period and the polynya's closure during the Normal Period. In contrast, GLASS (AVHRR) is incapable of detecting these transitions.

In Fig. 10 of Qu et al. (2016), the CLARA SAL was used as a reference to compare albedo maps acquired from GLASS (MODIS) with comparisons performed in the same regions as depicted in Fig. 15 of this work. Comparing the two figures demonstrates that the RTM/SciML retrievals agree well with direct-estimation results. Therefore, the limitation of GLASS

(AVHRR) could be attributable to the restricted number of shortwave channels accessible for the AVHRR sensor (three as opposed to seven for MODIS).

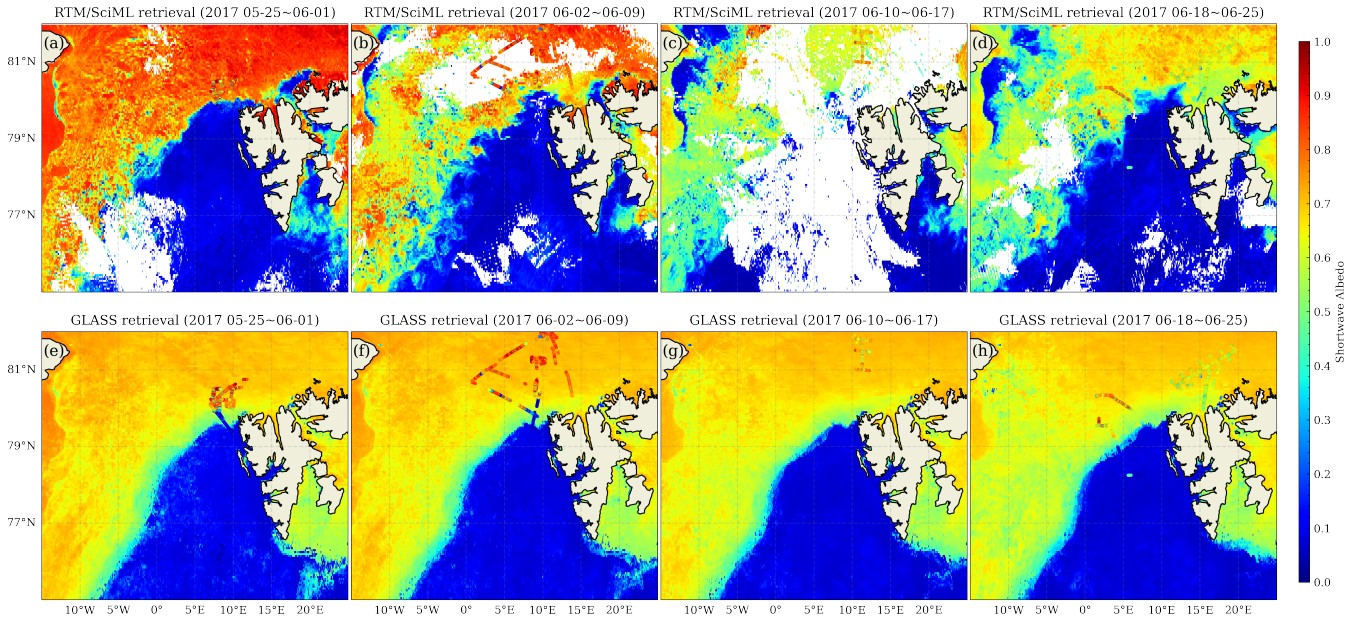

**Figure 18.** Maps of albedo derived using RTM/SciML (MODIS) and the direct-estimation algorithm (AVHRR data, Liang et al. (2021)) averaged during 8-day periods. In the top row, empty regions represent cloud obscurence throughout the 8-day periods. The pyranometer measurements from the ACLOUD campaign are superimposed over the retrieval maps; (a)&(e): flight on 2017-05-31, (b)&(f): flights on 2017-06-02, 06-04 and 06-08, (c)&(g): flight on 2017-06-14, and (d)&(h): flights on 2017-6-18, 06-20, 06-25 and 06-26.

Figure 19 illustrate pan-Arctic albedo images retrieved using RTM/SciML (MODIS and SGLI), the direct-estimation approach (VIIRS), and the MPD method (OLCI) for two representative days with thick ice (partially covered with snow) and
545 melting ice. In order to facilitate analyses, Fig. 20 displays the longitudinally-averaged albedo based on data collected on the same days. Because neither VIIRS nor OLCI provides open-water retrieval and because the cloud-filtering from the four sensors differ, data are collected to calculate average only when all four albedo products (MODIS, SGLI, VIIRS, and OLCI) have values at the same grid cells. Moreover, Fig. B3 illustrates the percentage difference between each retrieval result and the CLARA-SAL product (Karlsson, Karl-Göran et al., 2020). The albedo estimated with MODIS and SGLI sensors using
RTM/SciML models agrees quite well with the CLARA-SAL values. The VIIRS retrievals in the June-29 image are more comparable to these three, but the July-15 image appears to have lower values.

Due to the use of an uncoupled RTM in the training dataset for the direct-estimation method, the depths of snow, sea ice, and melt ponds were retained as fixed values, hence restricting the algorithm's retrieval precision for the more variable sea-ice surfaces. The VIIRS SURFALB product relies on linear relations between TOA reflectance and surface albedo at various
angular bins stored in a look-up table (LUT). The LUT include over 40,000 combinations of geometry angles. Multiplying the RTM-simulated or measured surface BRDFs (around 120,000) by the possible atmospheric configurations and by the

geometry-angle dataset, the resulting LUT is enormously large. Note, however, that only tens of thousands of surface situations were defined by the enormous LUT. Snow in nature possesses complex surface cover (which varies with density, impurity inclusions, thickness, and effective grain size), but the LUT employed in the VIIRS SURFALB prduct simply featured a snow layer with a fixed depth. The same issue exists in terms of sea ice conditions. In addition, since a LUT is essentially a linear regression model, the probable correlations between geometry angles and radiance/reflectance values from different channels are not learned in the training (i.e. look up values from the table) process.

It is difficult to construct a quantitative comparison between the RTM/SciML-albedo and VIIRS SURFALB products due to the lack of validation data that overlaps with the operational time of VIIRS SURFALB. Nonetheless, it is essential to highlight the value of our product's ability to retrieve albedo from any heterogeneous sea-ice surface, given that the impurities, pond depths, snow cover, and ice layers are all included in the training data for RTM/SciML models.

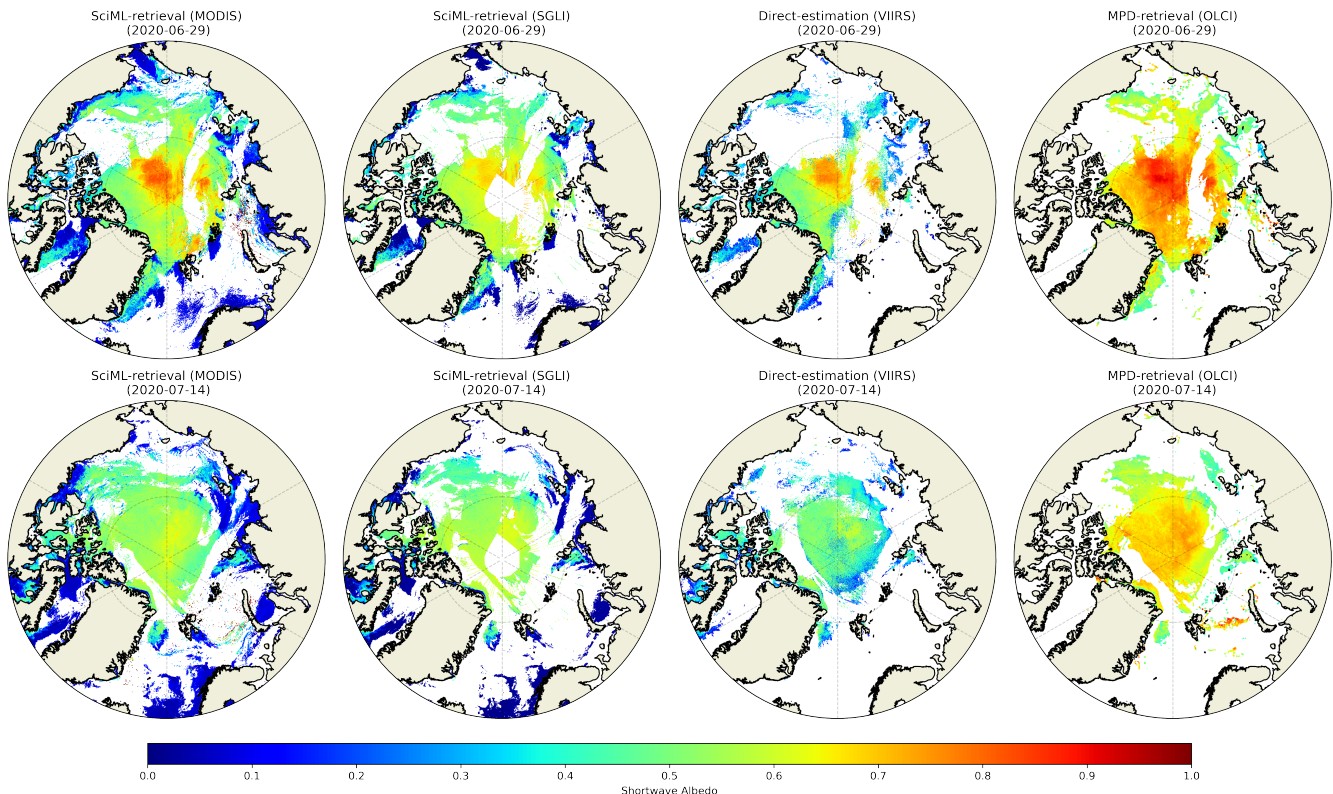

**Figure 19.** Pan-Arctic maps of albedo; from left to right: RTM/SciML (MODIS), RTM/SciML (SGLI), direct-estimation method (VIIRS), and MPD method (OLCI). The top row represents values for 2020-06-29, while the bottom row represents values for 2020-07-14. Empty regions in SciML images represent cloud pixels that were detected by the MLCM cloud mask, whereas empty regions in VIIRS and OLCI images represent either cloud pixels or open-water areas that were not processed by the algorithms.

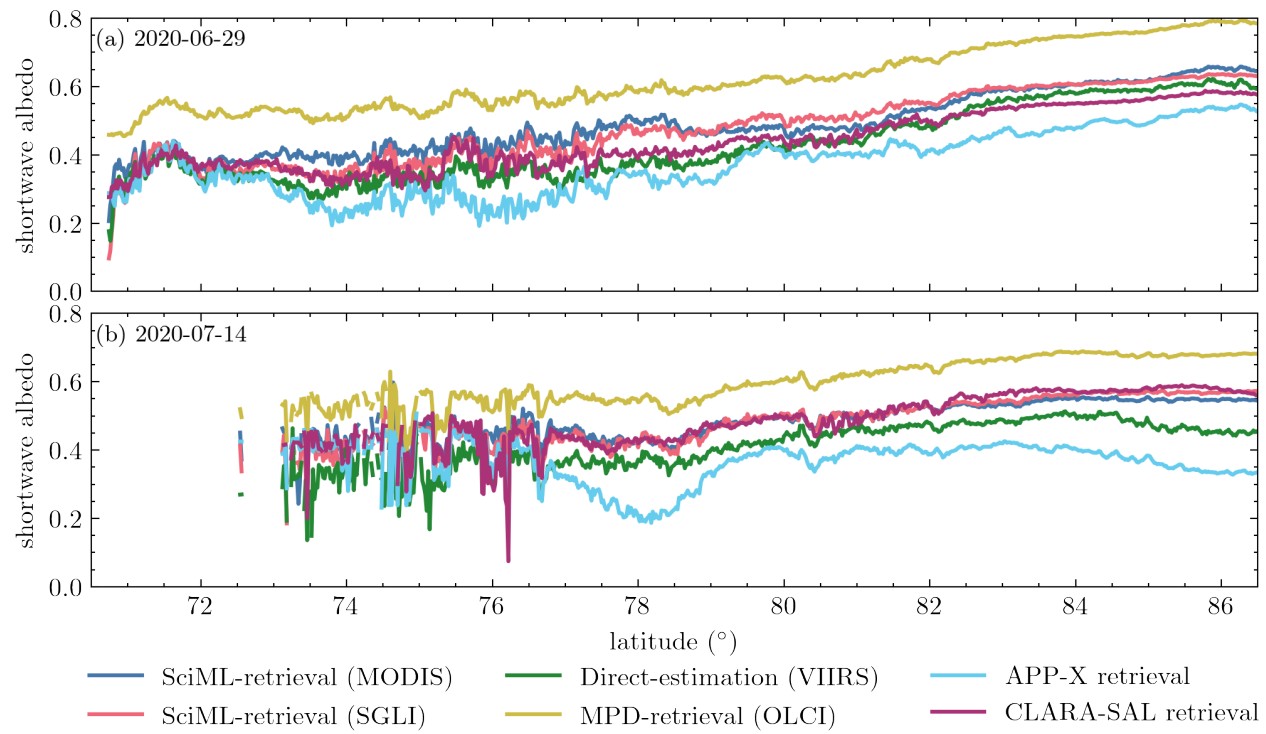

**Figure 20.** Longitudinally-averaged albedo based on data from Fig. 19. Only when all four albedo products (MODIS, SGLI, VIIRS, and OLCI) have values at the same grid cells are data collected for the calculation of the longitudinal average. In addition, the average albedo of APP-x and CLARA-SAL (5-day temporal resolution, Karlsson, Karl-Göran et al. (2020)) are displayed as points of reference. The temporal coverages of CLARA-SAL are (a): 2020-06-25∼2020-06-29 and (b): 2020-07-15∼2020-07-19.

## 6   Conclusion and summary

In this study, we described the development of a novel RTM/SciML sea ice albedo retrieval tool that can be applied to optical sensors that measure suitable radiance data. Comparisons of the retrieval results from SGLI and MODIS sensors with measurements showed good agreement. On a pan-Arctic scale, retrieval results derived from RTM/SciML models are most similar to the CLARA-SAL values, suggesting the reliability of the RTM/SciML framework. For *in situ* validation, comparison with albedo values acquired during low-level aircraft flights, retrieval of albedo values based on more than 2,000 data points during cloudless conditions demonstrated a small RMSE of 0.066.

The RTM/SciML-albedo algorithm was trained on a large synthetic dataset generated using a coupled RTM, and represents the optical properties of the cryosphere surface (bare ice, snow-covered ice, water, and melt pond). The combination of these two characteristics enables it to exploit the advantages of both the AccuRT and the RTM/SciML simulation tools.

When building the SD, we can reap the benefit of AccuRT being a RTM for the coupled system that accurately incorporates atmosphere-snow-sea-ice-water interactions to compute TOA radiances and corresponding surface albedo values. Information

of both the surface BRDF and the IOPs of the atmosphere is implicitly taken into account. We are thus saved from the procedure to perform atmospheric corrections. This retrieval procedure does not rely on predefined spectral reflectance threshold values for individual types of surface; the surface classification and albedo retrieval are separate processes, which eliminates errors caused by incorrect surface condition assumptions.

The RTM/SciML-albedo algorithm still possesses certain limitations. Without prior knowledge of the underlying radiative transfer theory, the ML models addressed in this paper can only approximate the hard-limit physical model (a RTM). Therefore, there may be adversarial instances in which small perturbations in the input data result in drastically divergent retrieval outcomes. Creating specialized network architectures and tailoring loss functions that represent known physical systems is one method for tackling this underspecification challenge (i.e. physics-informed neural networks, PINNs, (Cuomo et al., 2022; Daw et al., 2021; Di Natale et al., 2022)).

Meanwhile, in the current training data, snow or ice does not exhibit layered topographic variation for the sake of simplicity. Additionally, the present RTM/SciML albedo retrieval tool does not account for whitecaps and sun glints on the open ocean surface that would occur at oblique viewing angles. Nevertheless, the limits regarding training data can be adequately addressed by extending the SD in accordance with pertinent RTM simulations.

In summary, the RTM/SciML framework presented and discussed here has huge potential:

1. It may be used to retrieve albedo of relatively flat glaciers, because a flat layer of glacier ice is similar to sea ice without brine pockets (Warren, 2019). The 'wind-roughened air-water interface' can be represented in AccuRT using a one- or two-dimensional Gaussian surface slope distribution (for details see Stamnes et al. (2018)). In comparison to the existing method for retrieving glacier albedo (Ren et al., 2021), which uses the measured BRDF of sea ice to approximate glacier-BRDF (Gatebe and King, 2016) and bi-conical band reflectance observed by a space-borne imaging radiometer to approximate the ice albedo in the shortwave-infrared band, the methodology we propose here may be more appropriate for characterizing the anisotropic reflectance of a rough glacier surface.

2. The framework is generic in nature, allowing for comparisons not only between the MODIS sensors mounted on Terra and Aqua, but also to a large number of existing polar-orbiting sensors and well-planned future sensors, enabling sensor-to-sensor retrieval comparisons. In a recent review, Liang et al. (2019) emphasized the importance of developing retrieval algorithms that are broadly applicable to all satellite sensors: albedo retrievals based on multi-platform satellite sensors can significantly increase the amount of valid and accurate observational data, thereby increasing spatial and temporal coverage regardless of the specific method of retrieval. Meanwhile, developing retrieval algorithms using the same methodology enables a more extensive examination of the uncertainty associated with each sensor (e.g. identifying saturated channels and developing uncertainty assessment methods). The RTM/SciML framework presented in this paper is ideally suited for such a requirement.

## List of Acronyms

AANN: auto-associative neural network

ACLOUD: Arctic Cloud Observations Using Airborne
Measurements During Polar Day

AFLUX: airborne measurements of radiative and turbulent FLUXes of energy and momentum in the Arctic boundary layer

AVHRR: Advanced Very High Resolution Radiometer

BRDF: bi-directional reflectance distribution function

BSA: Black Sky Albedo

$f_{bc}$: black carbon impurity fractions

$f_{cloud}$: cloud fraction

$f_{EE \leq 15}$: percentage of data within 15% expected error

$f_{\text{mpf}}$: melt pond fraction

$f_{\text{sic}}$: sea ice concentration

GLASS: Global Land Surface Satellite

IOPs: inherent optical properties

MAPE: mean absolute percentage error

MERIS: MEdium Resolution Imaging Spectrometer

MISR: Multi-angle Imaging SpectroRadiometer

MLCM: Machine Learning Classification Mask

MLANN: multi-layer artificial neural network

MODIS: Moderate Resolution Imaging Spectroradiometer

MOSAiC: Multidisciplinary drifting Observatory for the Study of Arctic Climate

MPD: melt pond detection

NIR: near-infrared

NTBC: narrow-to-broadband conversion

OLCI: Ocean and Land Colour Instrument

PDF: probability density functions

SciML: scientific machine learning

SD: synthetic dataset

SGLI: Second-generation GLobal Imager

SMART: Spectral Modular Airborne Radiation measurement system

STBC: spectral to broadband conversion

SURFALB: surface albedo (the SURFALB product of VIIRS)

SW: shortwave

TOA: top of the atmosphere

VIS: visible

VIIRS: Visible Infrared Imaging Radiometer Suite

*Data availability.* The validation data, H5 files of MODIS retrieval, MODIS albedo retrieval model, and python script to use the retrieval model are intended to be published on PANGAEA.

*Author contributions.* YZ and WL developed the framework for albedo retrieval; YZ conducted radiative transfer simulations, compiled satellite retrievals and wrote the manuscript; NC developed the surface classification algorithm; and YF, NC, and KS contributed to the discussion of the results. All co-authors assisted with manuscript editing and provided substantial input to the interpretation of the results.

*Competing interests.* The authors declare that they have no conflict of interest.

*Acknowledgements.* We thank Drs. Johannes Stapf and Evelyn Jäkel of Leipzig University for supplying the pyranometer and albedometer observations that are now available on PANGAEA. This work was partially supported by grants from the Japanese Aerospace eXploration Agency (JAXA) and by the United States National Aeronautics and Space Administration (NASA).

## Appendix A: Parameterizations in the Synthetic Dataset (SD) for coupled RTM as input

### A1 Bare sea-ice

The radiative transfer processes in bare sea ice include absorption by pure ice, scattering by air bubbles, as well as scattering and absorption by brine pockets and solid salts (Jin et al., 1994). Unfortunately, direct observational data of air bubble volume fraction and radius ($V_{bu}$, $r_{bu}$) in the field is difficult (Petrich and Eicken, 2009). Lindsay (2001) used sea-ice thickness as the main factor to parameterize areal albedo and achieved an uncertainty within 0.05 and 0.10 (even without cloud screening), showing that sea ice thickness is the dominant parameter determining bare sea-ice albedo.

For the parameterization of sea ice in this study, the rest of the sea ice parameters (including brine pocket volume fraction, air bubble volume fraction, air bubble radius) are linearly fitted as a function of sea ice thickness ($h$) as shown in Table A1. The expectations are that thinner new young ice (NYI) and first year ice (FYI) ($\leq$ 30 cm and $\leq$ 1 m) would contain less bubbles and would in general be more saline than the thicker, multi-year ice (MYI) counterpart ($\geq$ 1.5 m). As sea ice deteriorates, brine water is excluded in the desalination process, which results in a smaller brine pocket volume fraction and radius values ($V_{br}$, $r_{br}$). Empirical equations were derived with the typical values of bubble and brine parameters for sea ice with increased thickness (adopted from Table 1 in Stamnes et al. (2011)).

It is worth noting that the equation of bulk brine concentration with ice thickness in Toyota et al. (2007) (which is based on *in-situ* measurements of sea ice from the Sea of Okhotsk) and our equation ($V_{br}$ in Table A1) both shows brine concentration approaching ~10% asymptotically as sea ice thickens.

**Table A1.** Physical parameters of ice. In generating the sea-ice thickness, a truncated-normal distribution with $\mu = 0.03$, $\sigma = 1.5$ was used to ensure an adequate amount of thin ice in the SD. The brine pocket radius conforms to a Tukey-Lamdba distribution with $\lambda$=0.5.

| Parameter | Sym. | Unit | Value |
|---|---|---|---|
| Sea-ice thickness | $h$ | m | $0 \sim 3$ |
| Brine pocket volume fraction | $V_{br}$ | – | $(-0.067 \cdot \log(h) + 0.1147) \cdot (1 + 0.2 \cdot r_{bu})$ |
| Brine pocket radius | $r_{br}$ | $\mu$m | $300 \sim 700$ |
| Air bubble volume fraction | $V_{bu}$ | – | $0.0214 \cdot h + 0.0068$ |
| Air bubble radius | $r_{bu}$ | $\mu$m | $-18.3 \cdot h^2 + 222.7 \cdot h + 96.5$ |

### A2 Sea ice covered with melt water

The water depth on thinner ice is in general shallower than on thicker ice. This situation is particularly true for the coastal polynya region, where ice intermittently melts and refreezes from melt water. In addition to temperature, topography of the ice underneath also influences melt water depth. A thinner ice layer is unlikely to evolve into hummocky topography (like MYI ice surfaces) to hold deep melt water (Perovich and Polashenski, 2012).

Generally, melt ponds on sea ice do not exceed 1-meter thickness. From RTM simulations, we found that a pond thickness
of 1.5 meter was the critical value that distinguishes the sea ice properties from the overlying melt water, and we therefore use
it as the upper limit of melt water thickness (Table A2).

In order to increase the variation of the SD, in addition to the melt water in the first ocean layer, we also added variation to
the chlorophyll-a (chl-a) concentration as well as Colored dissolved organic matter (CDOM) concentrations in the third ocean
layer (second layer is sea ice) in ranges typical for the areas with sea ice cover (König et al., 2019; Mustapha and Saitoh, 2008).

**Table A2.** Physical parameters of melt water on ice and ocean water. Melt water thickness and CDOM values follow randomly-distributed
uniform distributions in the specified ranges. For the chl-a concentration, a reciprocal continuous distribution (long tail extending to high
values) was used.

| Parameter | Units | Value |
|---|---|---|
| Melt water thickness | m | $0 \sim 1.5$ |
| Chlorophyll concentrations | $mg/m^3$ | $0.5 \sim 10$ |
| CDOM at 443 nm | /m | $0.01 \sim 0.1$ |

**A3  Sea ice covered with snow**

The albedo of snow depends on the grain shape and size, solar zenith angle, impurities in the snow, surface roughness, and the
thickness of the snow layer (Grenfell et al., 1994) as well as sky condition (clear or cloudy).

The snow cover on all sea ice types can accumulate to optically-thick values ($> 10$ cm). Nihashi et al. (2009) found that
with snow present, a 16 cm snow layer on 80 cm thick first-year ice would have the same insulation effect as a much thicker,
192 cm slab of ice. Similarly, in their simulations, Hamre et al. (2004) found that a 2.5-cm-thick snow cover has about the
same transmittance as a 61-cm-thick ice layer. In our simulations, the spectral albedo of both coarse grained (700 $\mu$m) and fine
grained (100 $\mu$m) pure snow (impurity fraction $f_{imp} = 10^{-8}$ ) both saturate when the thickness approaches 20 cm; beyond this
limit, adding more snow does not further increase the albedo. This value is therefore set as the upper boundary of snow cover
depth.

In their snow grain size retrievals, Jäkel et al. (2021a) found an optically equivalent snow grain size ($r_e$) on glacier and
smooth land surfaces centered in the $50 \sim 100$ $\mu$m range, while the range on smooth sea ice surfaces was $100 \sim 150$ $\mu$m. A
broad range of $50 \sim 150$ $\mu$m was adopted in our SD simulations (Table A3).

**A4  Geometries and atmospheric parameters**

The temperature, pressure, and concentrations of the major atmospheric elements were described using the 'subarctic summer'
atmospheric profile (Anderson et al., 1986). The small amount of aerosols typically found in the sea-ice regions on cloud-free
days and the relative impact of aerosols on the surface reflectance of such a bright surface indicate that the aerosol optical depth
(AOD) shown in Table A4 will suffice (Mehta et al., 2016; Winker et al., 2013).

**Table A3.** Physical parameters of snow cover. The snow grain size and snow thickness were generated with a randomly uniform distribution in the specified ranges.

| Parameter | Symbol | Units | Value |
|---|---|---|---|
| Snow grain size | $r_{\mathrm{e}}$ | $\mu$m | $50 \sim 150$ |
| Snow density | $\rho_s$ | kg/m$^3$ | 200 |
| Impurity fractions | $\mathrm{f_{imp}}$ | - | $10^{-7} \sim 10^{-6}$ |
| Snow thickness | $h_{\mathrm{snow}}$ | m | $0.01 \sim 0.2$ |

**Table A4.** Geometries and atmospheric parameters. All parameters conform to random-uniform distributions in the specified ranges.

| Parameters | Value |
|---|---|
| Solar zenith angle | 20~80 degrees |
| Sensor angle | 0.01~50 degrees |
| Azimuth angle | 0.01~180 degrees |
| AOD at 500 nm | $0.01 \sim 0.3$ |
| Relative humidity | 0.5 |
| Fine mode fraction | 0.9 |

## Appendix B:  Appendix Figures and Tables in Discussion

**Table B1.** Summary of the evaluation results using pyranometer data sampled from Section 2.5.2 (N=4000). The SciML models were classified into seven categories: Linear Regression with polynomial terms, multivariate adaptive regression splines (MARS) with polynomial terms, PCA-processed MARS, Tree models, Voting ensemble, Linear Blending ensemble, and MLANN. Within each category, the models that perform the best are shown in bold. The following metrics are used for evaluation: : R2 score, Pearson-r coefficient ($r$), RMSE, MAE, mean absolute percentage error (MAPE), $f_{\text{above}}$, $f_{EE\leq15}$, $f_{\text{below}}$ and Bias.

| Category | Model | R2 | r | RMSE | MAE | MAPE (%) | Regression relation | $f_{\text{above}}$(%) | $f_{\text{below}}$(%) | $f_{EE\leq15}$(%) | Bias |
|---|---|---|---|---|---|---|---|---|---|---|---|
| LR | degree1 (P1) | 0.85 | 0.94 | 0.09 | 0.07 | 24.3 | $\hat{y} = 0.74y + 0.12$ | 9.9 | 4 | 86 | 0.01 |
| | degree2 (P2) | 0.8 | 0.93 | 0.104 | 0.079 | 24.4 | $\hat{y} = 0.97y - 0.03$ | 3.3 | 15 | 82 | -0.013 |
| | degree3 (P3) | 0.79 | 0.92 | 0.105 | 0.073 | 21.1 | $\hat{y} = y - 0.03$ | 4.3 | 9 | 87 | -0.019 |
| | **degree4 (P4)** | **0.83** | **0.93** | **0.096** | **0.063** | **16.7** | $\hat{y} = y - 0.02$ | **7.5** | **4** | **88** | **-0.014** |
| | SGD | 0.85 | 0.94 | 0.09 | 0.07 | 24.5 | $\hat{y} = 0.74y + 0.12$ | 10.3 | 4 | 86 | 0.01 |
| MARS | degree1 (P1) | 0.66 | 0.93 | 0.135 | 0.111 | 21.8 | $\hat{y} = 0.68y + 0.07$ | 2.6 | 24 | 73 | 0.025 |
| | **degree2 (P2)** | **0.81** | **0.93** | **0.1** | **0.074** | **16.9** | $\hat{y} = 0.85y + 0.03$ | **2.7** | **8** | **90** | **-0.004** |
| | degree3 (P3) | 0.82 | 0.93 | 0.098 | 0.072 | 15.7 | $\hat{y} = 0.82y + 0.05$ | 2.7 | 7 | 90 | 0.007 |
| PCA-MARS | degree1 (P1) | 0.76 | 0.89 | 0.114 | 0.089 | 19.2 | $\hat{y} = 0.86y + 0.03$ | 6.3 | 11 | 83 | -0.005 |
| | **degree2 (P2)** | **0.83** | **0.93** | **0.095** | **0.067** | **14.9** | $\hat{y} = 0.86y + 0.03$ | **2.9** | **7** | **90** | **-0.001** |
| | degree3 (P3) | 0.81 | 0.93 | 0.101 | 0.073 | 16 | $\hat{y} = 0.83y + 0.03$ | 2.6 | 9 | 89 | 0.003 |
| Tree Models | Decision Tree (DT) | 0.57 | 0.89 | 0.151 | 0.114 | 28.1 | $\hat{y} = 1.02y + 0.08$ | 23.8 | 4 | 72 | -0.025 |
| | Random Forest (RF) | 0.55 | 0.89 | 0.155 | 0.12 | 34.4 | $\hat{y} = 0.96y + 0.13$ | 29.8 | 1 | 69 | 0.003 |
| | Extra Trees (XT) | 0.79 | 0.92 | 0.105 | 0.076 | 27 | $\hat{y} = 0.87y + 0.13$ | 17 | 2 | 81 | -0.003 |
| | AdaBoost (AdaB) | 0.3 | 0.81 | 0.193 | 0.162 | 36.2 | $\hat{y} = 0.6y + 0.07$ | 5.2 | 49 | 46 | 0.033 |
| | **Grad. Boost. (GB)** | **0.75** | **0.92** | **0.115** | **0.086** | **21.9** | $\hat{y} = 0.97y + 0.08$ | **17.2** | **2** | **81** | **-0.011** |
| | XGBoost (XGB) | 0.65 | 0.88 | 0.137 | 0.103 | 23.3 | $\hat{y} = 0.77y + 0.04$ | 3.3 | 22 | 75 | 0.01 |
| Voting | LR(P1, P2, P3, P4) | 0.85 | 0.93 | 0.09 | 0.06 | 16.6 | $\hat{y} = 0.95y$ | 4.1 | 6 | 90 | -0.012 |
| | MARS(P1, P3, P3) | 0.81 | 0.93 | 0.101 | 0.077 | 18.8 | $\hat{y} = 0.87y + 0.01$ | 2.5 | 8 | 90 | 0.001 |
| | SGD+GB+XGB | 0.86 | 0.94 | 0.087 | 0.056 | 16.4 | $\hat{y} = 0.91y + 0.08$ | 8.6 | 2 | 89 | -0.006 |
| | **LR+MARS+XGB** | **0.87** | **0.93** | **0.084** | **0.058** | **15** | $\hat{y} = 0.86y + 0.06$ | **5.1** | **4** | **91** | **-0.004** |
| Linear Blending | LR(P1, P2, P3, P4) | 0.82 | 0.93 | 0.098 | 0.066 | 18.4 | $\hat{y} = 0.99y + 0.04$ | 10 | 4 | 86 | -0.002 |
| | MARS(P1, P3, P3) | 0.83 | 0.93 | 0.096 | 0.066 | 18.5 | $\hat{y} = 0.96y - 0.01$ | 3.8 | 7 | 89 | -0.002 |
| | SGD+GB+XGB | 0.76 | 0.93 | 0.113 | 0.084 | 20.1 | $\hat{y} = y + 0.06$ | 15.7 | 2 | 82 | -0.001 |
| | **LR+PCA-MARS+SGD** | **0.85** | **0.93** | **0.091** | **0.063** | **16.9** | $\hat{y} = 0.92y + 0.01$ | **3.6** | **6** | **91** | **-0.003** |
| MLANN | (16 x 16 x 16) | 0.81 | 0.92 | 0.100 | 0.065 | 19.3 | $\hat{y} = 0.94y + 0.06$ | 12.7 | 3 | 85 | -0.001 |
| | **(16 x 10 x 8)** | **0.84** | **0.92** | **0.092** | **0.061** | **17.8** | $\hat{y} = 0.94y + 0.03$ | **8.4** | **4** | **88** | **-0.013** |
| | (10 x 10 x 10) | 0.85 | 0.92 | 0.091 | 0.060 | 17.5 | $\hat{y} = 0.93y + 0.03$ | 7.8 | 4 | 88 | -0.002 |
| | (8 x 8 x 8) | 0.84 | 0.92 | 0.092 | 0.060 | 18.2 | $\hat{y} = 0.9y + 0.06$ | 10.3 | 3 | 87 | -0.002 |

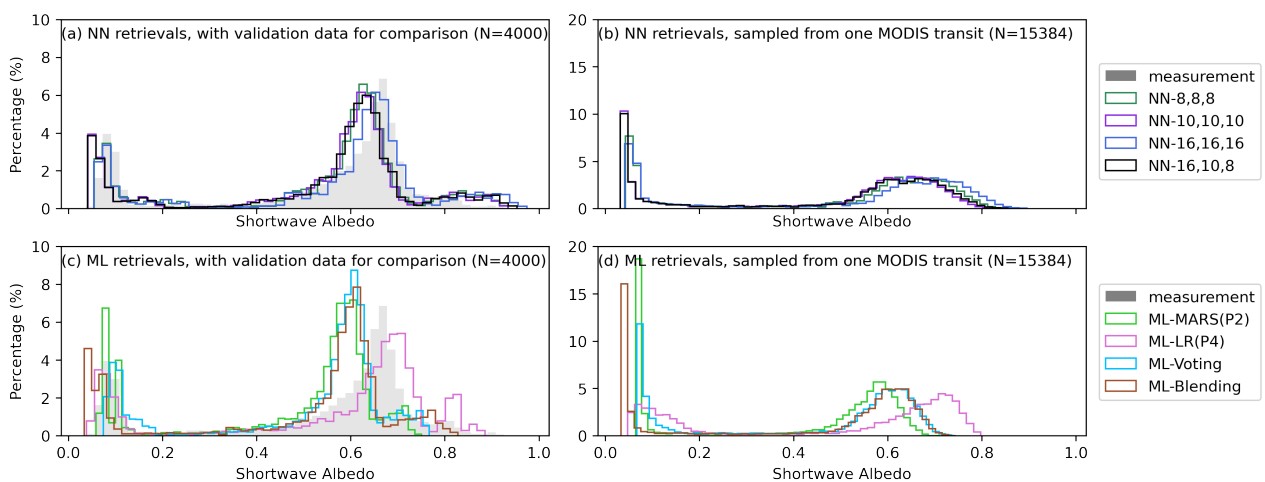

**Figure B1.** Top and bottom are, respectively, comparison of the retrieval results from four MLANN models and four ML models. The validation results of the eight selected models against aircraft measruements are provided in Table B1. Histograms of the measurements are presented as a reference (grey areas) in (a) and (c). (b) and (d) are sampled retrieval results using data from the MODIS transit at 14:00 UTC on 2017-06-25.

**Table B2.** Summary of the evaluation results for a time-constraint with a loose value ($\delta_t \leq 5$-h). The data were divided into four categories: three broadband albedos determined by the albedometer's spectral range, and a broadband shortwave albedo determined by the pyranometer's range. Three subcategories have been identified within each category: snow-covered sea-ice (abbreviated as 'snow' in the table), bare sea-ice ('ice'), and open water ('water'). Pearson-r coefficient ($r$), root mean square error (RMSE), mean absolute error (MAE), mean absolute percentage error (MAPE), $f_{\mathrm{above}}$, $f_{EE\leq15}$, $f_{\mathrm{below}}$, and bias are used to evaluate the retrieval in the subcategories (snow, ice, and water) and the total valid data ('all'). On the right side of the table, the total number of data in each subgroup is included (N).

| equipment (range) | surface | $r$ | RMSE | MAE | MAPE(%) | $f_{\mathrm{above}}$ | $f_{\mathrm{below}}$ | $f_{EE\leq15}$ | Bias | N |
|---|---|---|---|---|---|---|---|---|---|---|
| albedometer VIS | water | 0.89 | 0.053 | 0.046 | 37 | 1.2 | 3 | 96 | -0.005 | 248 |
| | snow | 0.64 | 0.069 | 0.051 | 7.3 | 3.7 | 0 | 96 | -0.004 | 3423 |
| | ice | 0.57 | 0.147 | 0.125 | 34.7 | 37.4 | 8 | 55 | 0.014 | 265 |
| | all | 0.92 | 0.076 | 0.056 | 11 | 5.8 | 1 | 93 | 0.001 | 3936 |
| albedometer NIR | water | 0.83 | 0.032 | 0.021 | 20.3 | 1.2 | 3 | 96 | -0.009 | 248 |
| | snow | 0.62 | 0.143 | 0.133 | 30.2 | 62.1 | 0 | 38 | 0.011 | 3423 |
| | ice | 0.58 | 0.122 | 0.105 | 45.4 | 50.2 | 2 | 48 | 0.003 | 265 |
| | all | 0.87 | 0.137 | 0.124 | 30.6 | 57.5 | 0 | 42 | 0.001 | 3936 |
| albedometer SW | water | 0.88 | 0.04 | 0.034 | 31.1 | 2 | 2 | 96 | -0.008 | 248 |
| | snow | 0.66 | 0.084 | 0.067 | 12.1 | 8.4 | 0 | 91 | -0.002 | 3423 |
| | ice | 0.56 | 0.139 | 0.119 | 41.4 | 46 | 3 | 51 | 0.009 | 265 |
| | all | 0.91 | 0.087 | 0.068 | 15.3 | 10.5 | 0 | 89 | -0.003 | 3936 |
| pyranometer | water | 0.86 | 0.036 | 0.026 | 24.1 | 0.2 | 2 | 97 | -0.008 | 1113 |
| | snow | 0.53 | 0.096 | 0.065 | 12.5 | 8.4 | 1 | 90 | -0.009 | 6257 |
| | ice | 0.5 | 0.182 | 0.152 | 41.8 | 28.3 | 31 | 41 | 0.026 | 594 |
| | all | 0.9 | 0.099 | 0.066 | 16.3 | 8.7 | 4 | 88 | -0.011 | 7964 |

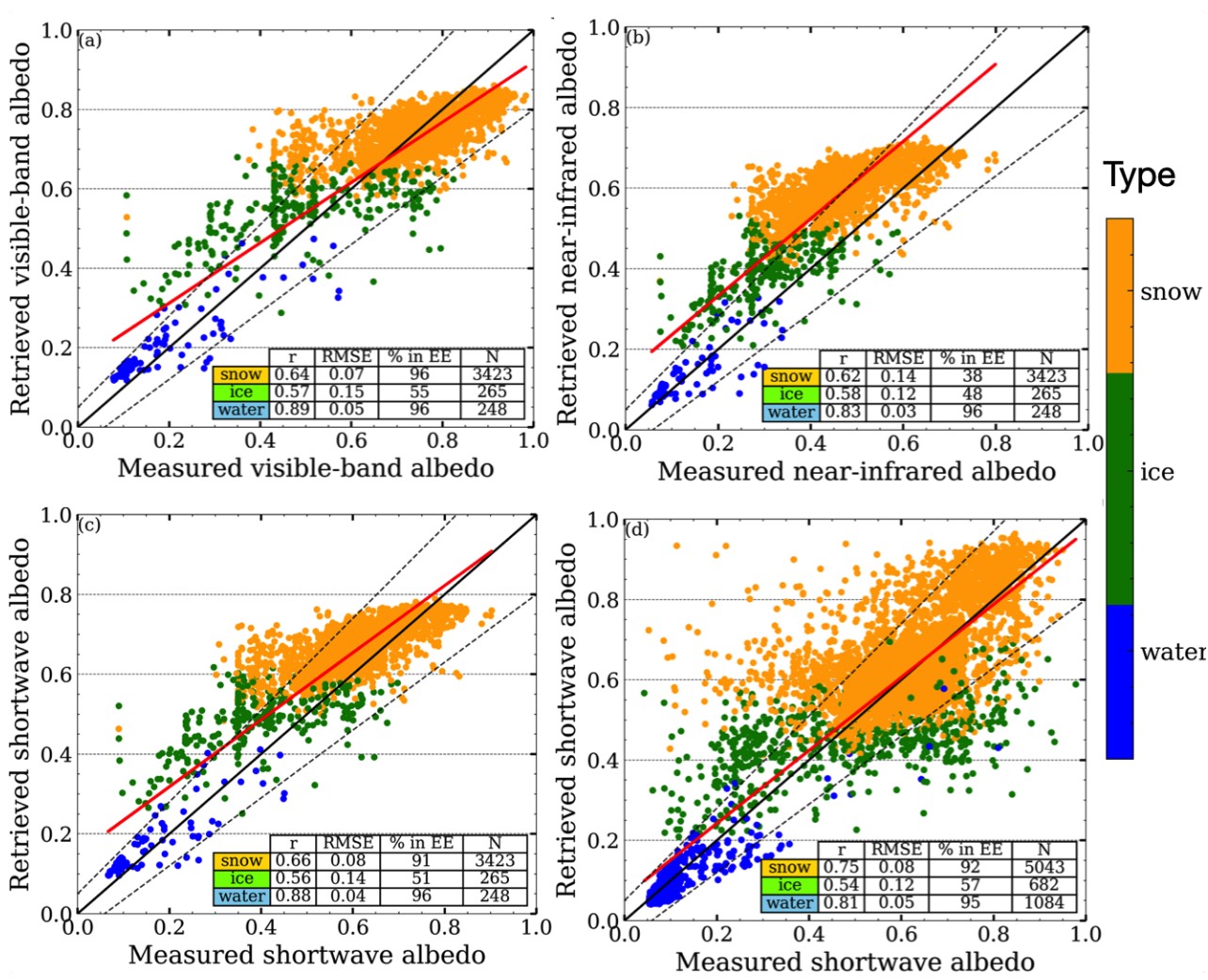

**Figure B2.** Same as Figure 4, but with the surface type represented by color. In addition, the lower right corner displays data for each surface type ($r$, RMSE, $f_{EE \leq 15}$, and $N$).

**Table B3.** Summary of the evaluation results for MCD43D and OLCI. The data were divided into four categories: three broadband albedos determined by the albedometer's spectral range, and a broadband shortwave albedo determined by the pyranometer's range. RTM/SciML retrievals have been regridded to compare the validation errors. R-squared (R2), Pearson-r coefficient ($r$), root mean square error (RMSE), mean absolute error (MAE), $f_{\text{above}}$, $f_{EE\leq15}$, $f_{\text{below}}$, and bias are used to evaluate the retrieval in the subcategories (snow, ice, and water) and the total valid data ('all'). On the left side of the table, the total number of data in each subgroup is included (N).

| | | R2 | r | RMSE | MAE | Regression relations | $f_{\text{above}}(\%)$ | $f_{\text{below}}(\%)$ | $f_{\text{within}}(\%)$ | Bias |
|---|---|---|---|---|---|---|---|---|---|---|
| SciML-albedomedter (N=525) | VIS | 0.86 | 0.96 | 0.083 | 0.073 | $\hat{y} = 0.87y + 0.08$ | 45.3 | 5 | 50 | 0.001 |
| | NIR | 0.44 | 0.92 | 0.09 | 0.075 | $\hat{y} = 1.2y + 0.03$ | 37.1 | 4 | 59 | -0.016 |
| | SW | 0.76 | 0.95 | 0.083 | 0.074 | $\hat{y} = y + 0.06$ | 32.6 | 4 | 64 | -0.01 |
| MCD43-albedomedter (N=525) | VIS | 0.81 | 0.94 | 0.095 | 0.069 | $\hat{y} = 1.09y - 0.06$ | 3.6 | 10 | 86 | -0.024 |
| | NIR | 0.48 | 0.91 | 0.087 | 0.072 | $\hat{y} = 1.16y - 0.08$ | 1.7 | 51 | 47 | -0.016 |
| | SW | 0.73 | 0.93 | 0.087 | 0.067 | $\hat{y} = 1.12y - 0.07$ | 3.4 | 16 | 81 | -0.021 |
| SciML-pyranometer (N=1093) | SW | 0.85 | 0.93 | 0.078 | 0.049 | $\hat{y} = 0.89y + 0.01$ | 7.9 | 7 | 85 | -0.007 |
| MCD43-pyranometer (N=1093) | SW | 0.77 | 0.93 | 0.099 | 0.078 | $\hat{y} = 0.96y - 0.05$ | 3.1 | 30 | 66 | -0.009 |
| SciML-ACLOUD (N=98) | SW | 0.69 | 0.84 | 0.092 | 0.071 | $\hat{y} = 0.75y + 0.14$ | 12.2 | 4 | 84 | -0.005 |
| OLCI-ACLOUD (N=98) | SW | 0.06 | 0.66 | 0.161 | 0.122 | $\hat{y} = 0.43y + 0.43$ | 29.6 | 1 | 69 | 0.016 |

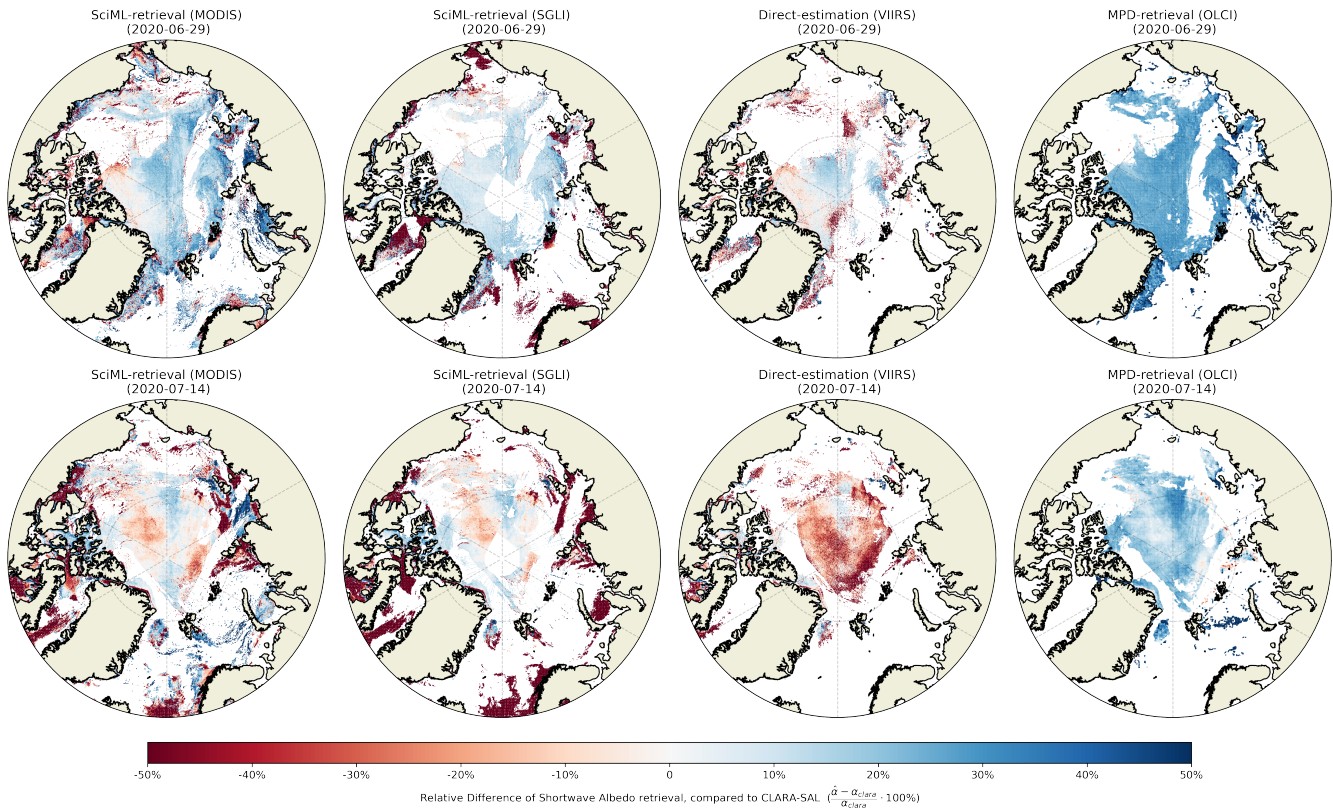

**Figure B3.** Percentage difference $(\frac{\hat{\alpha} - \alpha_{clara}}{\alpha_{clara}} \cdot 100\%)$ of albedo maps, as compared to the CLARA-SAL (Karlsson, Karl-Göran et al., 2020) results; from left to right: RTM/SciML (MODIS), RTM/SciML (SGLI), direct-estimation method (VIIRS), and MPD method (OLCI). The top row represents values for 2020-06-29, while the bottom row represents values for 2020-07-14. Empty regions in SciML images represent cloud pixels that were detected by the MLCM cloud mask, whereas empty regions in VIIRS and OLCI images represent either cloud pixels or open-water areas that were not processed by the algorithms.

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
