# Peer review of "A sensor-agnostic albedo retrieval method for realistic sea ice surfaces - Model and validation"

_The Cryosphere, 2021_

## Referee Comment (RC2)

[referee-annotated manuscript omitted]

---

## Referee Comment (RC4)

**A sensor-agnostic albedo retrieval method for realistic sea ice surfaces - Model and validation**

Yingzhen Zhou1, Wei Li1, Nan Chen1, Yongzhen Fan2, and Knut Stamnes1

1Light and Life Laboratory, Department of Physics, Stevens Institute of Technology, Hoboken, NJ 07307, USA 2Cooperative Institute for Satellite Earth System Studies (CISESS), Earth System Science Interdisciplinary Center (ESSIC), University of Maryland, College Park, MD 20740, USA,

**Correspondence:** Yingzhen Zhou (yzhou64@stevens.edu)

Abstract. The cryosphere's surface (snow, sea ice, and water) regulates global climate through several feedback mechanisms. Broadband albedo is a critical parameter determining the radiative energy balance of the complex atmosphere-cryosphere system, but there is currently no relia (e, o) erational albedo retrieval product capable of assessing the global sea-ice albedo with sufficient spatial-temporal resolution resolution reliates of sea-ice dynamics and for use in global climate models.

- A framework was established for remote sensing of sea ice albedo that integrates sea-ice physics with high computational efficiency, and can be applied to any optical sensor that measures appropriate radiance data. A scientific machine learning (SciML) approach was developed and trained on a large synthetic dataset (SD) constructed using a coupled atmosphere-surface radiative transfer model (RTM). The resulting RTM/SciML framework combines the RTM with a multi-layer artificial neural network SciML model. In comparison to the NASA MODIS MCD43 albedo product, the framework does not depend
- 10 on observations from multiple days, and can be applied to single angular observations obtained under clear-sky conditions. Compared to the existing melt pond fraction-based approach for albedo retrieval, the RTM/SciML framework has the advantage of being applicable to a wide variety of cryosphere surfaces, both heterogeneous and homogeneous. Validation of the RTM/SciML albedo product using MODIS and SG (1 day) against measurements obtained from aircraft campaigns revealed excellent agreement, with mean absolute error 0.0+7 for above 2000 clear-sky pixels.

**15 1 Introduction**

Sea-ice in the Arctic regulates global climate through multiple feedback processes (Robock, 1983; Bitz and Roe, 2004), and the ice-albedo feedback is perceived as one of the most important factors contributing to sea ice thinning in the Arctic (Curry et al., 1995; Gildor et al., 2002). Sea-ice cools Earth by reflecting solar energy back to space and by insulating the ocean water underneath from the atmosphere. An initial warming of sea ice can accelerate the melting process dramatically and thus amplify the surface warming, which makes the ice-covered regions more sensitive to small changes in global temperature and

20

carbon emissions (Pithan and Mauritsen, 2014; Comiso and Hall, 2014).

It is not new that Arctic sea ic new on the decline in the past years (Stroeve et al., 2007; Serreze et al., 2007; Kwok and Rothrock, 2009; Tschudi et al., 2007; Kwok and some areas of the Arctic, the average sea ice thickness near the end of the melt season

25

30

has decreased by two-third over the last six decades (Kwok, 2018). Some researchers even made postulations, forecasting the year when the first ice-free summer would occur in the Arctic (Wang and Overland, 2012).

For decades, optical sensors deployed on geostationary and polar-orbiting satellites have been used to derive the global-scale surface broadband albedo. However, geostationary satellites can only offer data within the latitude range of 60°S to 60°N, omitting the polar regions entirely. With regard to polar-orbiting satellites, the majority of albedo products are land-surface products, while ocean surface albedo data (including sea-ice) are left blank (Qu et al., 2015). Table 1 compares the currently operational products and algorithms capable of retrieving albedo at the cryosphere's surface.

[revised manuscript text omitted]

  flight level (αl, l = 350 m), and high flight level (αh, h = 3000 m). Albedo ratios, rl = αl/αs and rh = αh/αs, were calculated to determine the difference in albedo induced by the atmospheric layer below the flight height.

Figure 1 illustrates the aforementioned spectral and broadband albedos, as well as the ratios. There is greater influence of multiple scattering above the open water surface due to atmospheric components, particularly in the visible (VIS) range; the albedo at h = 3000 m is double that on the surface, whereas the presence of an atmosphere and aerosols results in a decrease in albedo over a bright surface. Among these, the reflection is particularly noticeable in the near-infrared (NIR) band, where air absorption results in a 4% and a 22% decrease in at low and high levels, respectively. Similar simulation results were found by Jäkel et al. (2021a), using the Two-streAm Radiative TransfEr in Snow (TARTES) model (Libois et al., 2013). As can be observed, the difference between low-level and surface albedo is less than 5%, whereas the difference between high-level and low-level albedo is significantly greater. As a result, we did not use aircraft observations taken above 350 m in order to improve the validation of the surface albedo retrieval.

With aforementioned pre-processing steps including geographic collocation and observational altitude restrictions, measurements from two days during the AFLUX campaign were partly from clear-sky, while the remainder were entirely from broken clouds (for a description of the AFLUX data, see Stapf et al. (2021b)), and seven flight transits from ACLOUD were retained. The MOSAiC campaign included fewer than 50 valid data points, and all obtained for broken cloud conditions. To eliminate arrors caused by dense cloud cover, MOSAiC data were omitted from validation.

In all, the selected flight sections are depicted in Fig. 2. Up to four satellite images per day could be employed for retrieval and evaluation in the polar regions. When flight transits were compared to the percentage of cloud coverage  $(f_c)$  for the

---

## Author Response (AR1)

**1 Response to Reviewer 1**

**1.1 Specific Comments**

**Reviewer comment:** Although detailed information of the coupled radiative transfer model AccRT can be found in the literatures, I suggest to add a concise description about it in the manuscript.
**Response:**
The coupled radiative transfer model (AccuRT) was described twice, on line 76∼84 and 156∼179 (including Table 2), which summarizes the two in-text references, Stamnes et al., 2011 and 2018. However, because the material is split across two sections that span three complete pages, we may have missed communicating the concept clearly. Sections 1 and 2 have been reorganized in the revised version.

**Reviewer comment:** The method of how to construct the synthetic dataset (SD) with the coupled RTM is not clear. The detailed information about the inherit optical properties (IOPs) listed in Table 2 are needed, such as the data ranges, probability distribution, and constraints.
**Response:**
Physical parameters of ice, melt water on ice, and ocean water, physical parameters of snow cover, geometries, and atmospheric characteristics has been included in the revised version (Appendix A, pages 37∼39).

**Reviewer comment:** The framework of the RTM/MLANN is not clear. I suggest to add a flowchart for it.
**Response:**
We are grateful for the suggestion. A flowchart has been included (Fig. 1) in the revised version.

**Reviewer comment:** In the manuscript, the MLANN method to used estimate the sea ice albedo. What are the performances of training, validating, and predicting accuracies of this artificial neural network model?
**Response:**
The RMSEs of training, out-of-sample validation and independent validation for Model 1 are 0.006, 0.0063 and 0.092, and those for Model 2 are 0.0062 and 0.0068, 0.099, respectively. These statistics are not included in the revised submission, but the following texts are included: As depicted in Fig. 1, the final models are selected after two levels of validation: the first filter is on

[Figure]

Figure 1: Flowchart of the proposed RTM/SciML framework for albedo retrieval.

the out-of-sample validation, which uses the portion of the SD that was not included in the network training process (20% of SD); and the second filter is on data sampled from campaign measurements (50% of data from the pyranometer measurements, Section 2.5.2). Adopted are the MLANN models with the least RMSEs throughout both levels of validation

**Reviewer comment:** The authors declared that the sensor-agnostic albedo retrieval method has the ability to apply to any optical sensor, however few explanations about this are shown in the manuscript. I suggest the authors to further explain the major theories of this method. In fact, other methods such as the MPD and direct-estimation algorithm, can also be adopted to other sensors easily. Please add a discussion about it.

**Response:**

The missing piece we did not include in the submitted version is how we determined the 'appropriate' channels from any optical sensor. Apart from the coupled RTM model that calculates the synthetic dataset, we use a technique to ensure that the radiance we use in practice (input from satellite channels) is consistent with that in our training data, allowing our machine learning models to be directly applied to satellite data.

The following contents have been added to the revised paper (Section 2.3).

With our knowledge of radiative transfer theory and the differences in the radiative properties of the constituents in the coupled atmosphere-surface system, we first chose the input channels based on the following criteria:

- Avoid wavelengths with significant absorption by water vapor and/or other atmospheric constituents.

- Avoid sensor channels that have been found to be saturated in previous sensitivity investigations.

- Select wavelengths that, based on their albedo spectra, can best identify snow cover, bare ice, open water, and melt-pond surface.

With the assistance of the auto-associative neural network (AANN) technique, channels with a significant reconstruction error are deemed unsuitable for use as input to the retrieval model. More specifically, an AANN is trained using the synthetic data generated by radiative transfer model (RTM), which takes as input the three sun-satellite geometry angles as well as all radiance data that meets the aforementioned requirements and outputs all radiances. The trained AANN is believed to have picked up on the patterns in the RTM-generated dataset. Following that, the AANN is fed the same input features derived from the satellite sensor. We calculate the absolute percentage error of the reconstruction output and prune channels with an error greater than 5%.

This method is intended to avoid 'covariate shift' — a phrase used in machine learning to refer to the difference between independent variables in training and real-world data. Covariate shift is due to either (a) the saturation of certain satellite channels, which results in a much narrower dynamic range of radiance data from the satellite sensor (real world) than that calculated using RTM (training data), or (b) the response function and wide wavelength range results in a non-negligible difference between the radiance derived from the central wavelength and that obtained from the sensor. It has been demonstrated that the AANN technique is effective in detecting mismatches between data acquired for the retrieval task and data utilized for training. A recent paper[1] discusses how the AANN approach was used to identify optimal channels for retrieving ocean color products using a variety of sensors.
* * *
[1] *Fan, Yongzhen, et al. "OC-SMART: A machine learning based data analysis platform for satellite ocean color sensors." Remote Sensing of Environment 253 (2021): 112236.*

| λ of MODIS | λ of SGLI |
|:---:|:---:|
| 469 | 443 |
| 555 | 530 |
| 645 | 673.5* |
| 858.5 | 868.5 |
| 1240 | 1050 |
| 1640 | 1630 |
| 2130 | 2210 |

Table 1: A comparison of the centre wavelengths of the channels explored and tested for the purpose of retrieving the broadband albedo at the cryosphere surface using MODIS and SGLI sensors. The 673.5 nm wavelength channel from SGLI (shown by an asterisk in the table) was proven to be saturated and hence was not used to derive albedo. Details can be found in the revised version.

Similar approaches have been used to identify acceptable channels for albedo retrieval. Table 1 lists the MODIS channels that were utilized to retrieve albedo, as well as the GCOM-C/SGLI channels that were evaluated and eventually employed [2].

**Reviewer comment:** The comparisons with MCD43D, MERIS, and OLCI datasets were not easily for reader to interpret. I suggested to add scatter plots to compare the differences of these datasets.
**Response:**
We appreciate your suggestion. Scatter plots were omitted due to the fact that the retrievals did not cover equivalent areas. For example, our cloud classification mask is stricter than those employed by OLCI and MERIS. The melt-pond detection approach does not provide values for open-water areas. Similarly, the MCD43 product returns values just along the coast. As a result, we anticipated scatter plots would be less useful than albedo maps. However, as you noted, quantitative depiction is helpful. In the revised submission, we re-gridded the data and provided statistics at the geological areas specified in the revised version (e.g. Figure 16 on page 28 and Figure 19 on page 32 in the revised submission).
* * *
[2] Our team initially discovered the saturation issue in the 673.5 nm channel using AANN, submitted it to the GCOM-C/SGLI team, and obtained confirmation of the issue.

**Reviewer comment:** Figure 13, the authors declared that the MERIS albedo product are higher than the albedo estimated by the MLANN method in the areas with large melt pond fraction (greater than 50%). However, this difference is not obvious, and the major differences appeared in the upper right corner. Please provide an explanation for it.

**Response:**

Other than the wavelength difference (MERIS sensor does not have short-wave near-infrared channels), we believe the discrepancy in the area covered by snow is due to algorithmic difference.

In our training data (the synthetic dataset created by the radiative transfer model), we explore the situation of snow-covered sea ice with snow depths ranging from 0.01∼0.2 m to 50∼150 $\mu$m (Table 6). Notably, 0.2 meter is the optically thick snow threshold. We found that the trained models use different channels and relations to retrieve the albedo of snow and ice surface. This topic is discussed in a separate paper that will be submitted shortly, in which we used the Shapley Value to deduce how these models compute albedo based on input channels and geometry angles.

The MPD-based algorithm does not discriminate between snow-covered sea ice and bare ice when retrieving albedo; both scenarios are classified as 'ice', and for both snow and ice surface, the same iterative method and the same channels were used to calculate spectral albedo. The spectral albedo is subsequently integrated to obtain broadband albedo.

In the revised version, details of how the three approaches (SciML/RTM, direct-estimation and MPD) differ in the treatment of snow-covered ice, melt-pond and bare-ice are discussed in more detail in Section 5.

**Reviewer comment:** Figure 13, the measurements of campaigns were not shown in this figure. Why? Please add the validation data for comparison.

**Response:**

The reason is that there are no campaign-measured data available to validate these results. The two locations (top and bottom) and time period (averaged between DOY 166 and DOY 170 in 2007) selected to compare the results from our algorithm and those from MERIS are the same as those used in the subsequent article:

*Qu Y, Liang S, Liu Q, et al. Estimating Arctic sea-ice shortwave albedo from MODIS data[J]. Remote Sensing of Environment, 2016, 186: 32-46.*

As noted in line 36∼41, Qu and Peng retrieved sea-ice albedo using the direct estimation method. Initially, we intended to utilize Qu's results as a

benchmark for comparing the three algorithms. However, because we were unable to obtain the authors' original retrieval data in order to include it in the subplot, we could only show three columns in Figure 13. For your reference, Fig. 2 below shows the comparison of the three products, and the first column are screenshots of the results of Qu's algorithm, taken directly from their paper. For the second and third column, we used the same color-bar to plot the results and manually boxed the same area, but due to difference in printing and in coordinates, the colors/regions don't exactly match with panels (a) and (b).

[Figure]

Figure 2: Maps of albedo and melt pond fraction averaged during a 5-day period in 2007 between DOY 166 and 170. From left to right: Qu's albedo retrievals, MLANN-based and MPD-based albedo retrievals, as well as the MPD-derived melt pond fraction, respectively (Qu2015Mapping, this study, and Istomina2015Melt). The upper panels depict the Banks, Prince Patrick, and Melville Islands, while the lower panels depict the Kara Sea. At the bottom, colorbars representing the corresponding values are displayed. Note that the images of Qu's retrieval results (along with the colorbar) are taken directly from Fig.10 in Qu2015Mapping, as no other data was obtainable. In panels (c) and (d), empty regions represent cloud pixels that were detected by the MLCM model (and hence removed), whereas empty regions in panels (e) through (h) represent either cloud pixels or open-water areas that were not processed by the MPD algorithm.

In the revised manuscript, we provided the CLARA-SAL results (Fig. 15 on page 27, which was also included in Qu's paper as points of reference) as reference to make up for the lack of validation data.

**Reviewer comment:** In the abstract, the mean absolute error (MAE) of 0.047 was used for indicating the accuracy of this method. I suggest to use root mean standard error (RMSE) to represent the estimation accuracies for the visible, near infrared, and shortwave albedo.

**Response:**
The text has been modified (lines 9~12 and 331~334 of the revised submission) to include the statistics.

**1.2 Technical Corrections**

**Reviewer comment:** Figure 7. The color ramp of this figure is not easily to interpret. Please change it.

**Response:**
A figure with more distinguishable colors has been presented in the revised version.

**Reviewer comment:** Line 493, the sentences of "Istomina et al. (2015); Istomina (2020)" can be rewritten as "Istomina et al. (2015; 2020)".

**Response:** Revised.

**Reviewer comment:** Caption of Figure 13. "(Qu et al. (2015), this study, and Istomina et al. (2015))". The reference Qu et al. 2015 is not related with this figure.

**Response:**
Revised.

**2 Response to Reviewer 2**

**2.1 Question & Suggestion**

**Reviewer comment:** The coupled RTM is used to simulate TOA reflectance from various sea ice surface and atmospheric properties. The surface parameters are listed but the values were not mentioned as well as the sampling strategy. I cannot figure out how the authors determine the distribution and relevance among the parameters. Similar concern for the atmospheric parameters and the solar/view angles.

**Response:**

Physical parameters of ice, melt water on ice, and ocean water, physical parameters of snow cover, geometries, and atmospheric characteristics have been included in the revised version (Appendix A, pages 37∼39).

**Reviewer comment:** The machine learning method needs more detailed description about how it was used. How to deal with the invalid retrievals from the relationship? Is there a post-processing? It was mentioned there are two models trained. What are their difference and advantages?

**Response to 'how post/pre-processing were used to avoid invalid retrievals':**

A flowchart (Figure 1) illustrating the process of obtaining a final retrieval model was included in the revised version.

The training dataset only contains snow, ice and melt-pond surface types. As a result, a machine learning classification mask (MLCM) [3] is employed as a post-processing step to filter out invalid pixels.

In addition to 'post-processing', we use auto-associative neural network (AANN) technique as a 'feature selection' tool to avoid the machine learning model obtains invalid retrievals. AANN can effectively avoid 'covariate shift' and ensure that the data range of radiance we use in practice (input from satellite channels) is consistent with that in our training data (synthetic dataset derived from radiative transfer model).

**The description of AANN is included in Section 2.3 in the revised manuscript.**

**Response to 'difference between the two neural networks'**
* * *
[3]Chen, Nan, et al. "New neural network cloud mask algorithm based on radiative transfer simulations." Remote Sensing of Environment 219 (2018): 62-71.

This article discussed two neural network models, the only distinction being the wavelength range of the broadband albedo output. The pyranometer and the albedometer have distinct ranges of wavelengths. The albedometer measures spectral irradiance (up/down) in the range of 0.4∼2.1 $\mu$m, whereas the pyranometer measures broadband irradiance (up/down) in the range of 0.3∼3.6 $\mu$m. As a result, we trained two distinct models to make the most use of the data from the two types of equipment. The revised version includes a 'Data' section that separates the backgrounds of satellite and validation data sources from the discussion of the results. Table 2 is included to show the difference between the instrument types and the matching models.

| | Model 1 | | Model 2 | |
|---|---|---|---|---|
| | $\lambda$ range (nm) | validation data | $\lambda$ range (nm) | validation data |
| Visible | 300-700 | / | 400-700 | albedometer |
| Near Infrared | 700-2500 | / | 700-2100 | albedometer |
| Shortwave | 300-2500 | pyranometer | 400-2100 | albedometer |

Table 2: Difference between the two models mentioned in the text. Figures 3, 6 and Table A2 show retrieval and validation results of the two models.

**Reviewer comment:** The author emphasized many times about the advantage of the proposed method than the previous MPD or direct-estimation methods. However, many descriptions needs to be clarified or discussed more. What are the advantage of the coupled RTM rather than the separate radiative transfer models? Is there any quantitative comparison about this? Is a classification within sea-ice surface needed in previous methods? The method is claimed as independent on sensor or spatial resolution, how that is realized without considering the spectral response function difference? Did the previous method restrict to a specific spatial resolution?
**Response:**
**The following text has been added to the revised version to discuss the advantages of utilizing a coupled RTM over decoupled models.**

Due to the fact that uncoupled RTMs make disparate assumptions about the atmosphere and surface when constructing the BRDF, albedo retrieval with uncoupled RTMs is difficult to scale.

While MPD-based approaches are capable of retrieving both the albedo and melt pond percentage for a surface composed of melt water and white

ice, they are only effective during particular seasons (discussed in [4] [5] [6]). Additionally, because the spectral reflection coefficients for the melt-pond and thin ice boundaries, as well as the thick ice and snow-cover boundaries, are manually adjusted based on the surface condition, there are greater uncertainties in the retrieval during the transitional seasons of spring-summer and summer-autumn, as well as when the surface is highly heterogeneous (low sea ice concentration, discussed in Istomina2015Melt). Two more issues with the method are its omission of open-water conditions and restriction of sea ice type. The MPD algorithm models sea ice's BRDF exclusively for dry and white ice (based on the absorption of yellow pigments), ignoring the effects of air bubbles and brine pockets (discussed in Zege2015Algorithm).

In the direct-estimation method, linear relations between TOA reflectance and surface albedo are derived for different angular bins. Qu2015Mapping [7] and Peng2018VIIRS [8] generated datasets detailing the intervals of geometry angles in their models, which already have over 40,000 combinations. Multiplying the RTM-simulated or measured surface BRDFs (which range between 100,000 and 120,000 for their retrievals) by the possible atmospheric configurations and by the geometry-angle dataset, results in an extremely big value for the look-up table (LUT). However, note that only hundreds-thousand level surface conditions were characterized using the monstrous LUT, and because a LUT is essentially a linear regression model, the LUT does not learn the possible interactions between the input features (geometry angles and radiance/reflectance values from various channels), making the approach less efficient.

It is worth noting that the 'RTM-simulated surface BRDFs' mentioned earlier for estimating the radiative properties of sea-ice and snow surface was
* * *
[4]Istomina, L., et al."Melt pond fraction and spectral sea ice albedo retrieval from MERIS data–Part 2: Case studies and trends of sea ice albedo and melt ponds in the Arctic for years 2002–2011." The Cryosphere 9.4 (2015): 1567-1578.

[5]Istomina, L., et al. "Melt pond fraction and spectral sea ice albedo retrieval from MERIS data–Part 1: Validation against in situ, aerial, and ship cruise data." The Cryosphere 9.4 (2015): 1551-1566.

[6]Zege, E., et al. "Algorithm to retrieve the melt pond fraction and the spectral albedo of Arctic summer ice from satellite optical data." Remote Sensing of Environment 163 (2015): 153-164.

[7]Qu, Ying, et al. "Mapping surface broadband albedo from satellite observations: A review of literatures on algorithms and products." Remote Sensing 7.1 (2015): 990-1020.

[8]Peng, Jingjing, et al. "The VIIRS sea-ice albedo product generation and preliminary validation." Remote Sensing 10.11 (2018): 1826.

derived from the IOP model, which is employed by the coupled RTM that we are utilizing [9]. Because Qu's algorithm decouples the atmosphere from the ocean layer, it is unable to accurately simulate the 'snow-covered sea ice' situation. In a coupled RTM, snow is simulated as a layer of snow on the surface above the interface (the upper slab of the coupled system), and sea ice is simulated as a layer of ice with brine pockets and air bubble inclusions on deep ocean water (the lower slab of the coupled system).

Although Qu used the same IOP model to calculate the optical properties of the two media, the 'snow surface' scenario refers to snow that has been placed on land. Additionally, snow possesses complex surface cover (that varies with density, impurity inclusions, thickness, and effective grain size), but the LUT only included a snow layer of a fixed depth. The same issue exists with regard to sea ice conditions. Due to the fact that the direct-estimation algorithm is very dependent on the quality of the LUT, Qu's model's snow and ice retrieval is rather crude in comparison to the more refined approach used in this study.

One last methodological difference is, when using the direct estimation method and building a BRDF table, the reflectance anisotropy (namely, the strong forward peak of snow and ice), which occurs when the solar zenith angle equals the sensor zenith angle, is manually corrected offline (see Qu2016Estimating). However, in our coupled RTM, the forward peak can be adjusted inside the radiative transfer calculations (described in [10]).

**Quantitative comparison between a coupled and decoupled model?**
**Response:**
Unfortunately, there currently is no literature comparing the performance between a 'coupled' and a 'decoupled' RTM. This topic is also out of the scope of the current paper.

**Is a classification within sea-ice surface needed in previous methods?**
To clarify, the classification in our framework is a 'post-processing' step; it is conducted independently and has no effect on the albedo retrieval results.
* * *
[9]Stamnes, Knut, et al. "Modeling of radiation transport in coupled atmosphere-snow-ice-ocean systems." Journal of Quantitative Spectroscopy and Radiative Transfer 112.4 (2011): 714-726. was cited in the relevant sections

[10]Jiang, Shigan, et al. "Enhanced solar irradiance across the atmosphere-sea ice interface: a quantitative numerical study." Applied optics 44.13 (2005): 2613-2625.

For both the direct-estimation method and the MPD-based method, other than cloud-screening, they did not have a post–processing classification algorithm to identify the surface type of each pixel after the surface albedo is retrieved. However, the logic of MPD algorithm requires the surface to be classified and obtains spectral albedo afterwards. In 2.3.2 of Zege's paper, it was mentioned that three channels are used to separate and discard open water pixels, two channels are used to separate the white surface (snow and ice are considered as the same category) from melt pond. The added contents (from the previous response) should help to clarify this issue.

**Clarification on the spatial resolution**

Other than the MCD43 product, all the other three methods discussed can all retrieve albedo based on single-angular observations. MCD43 requires data from multiple days and therefore the spatial and temporal resolution are lower.

**2.2 Minor Comments**

**Reviewer comment** in the context of 'but there is currently no reliable, operational albedo retrieval product capable of assessing the global sea-ice albedo with sufficient spatial-temporal resolution for studies of sea-ice dynamics and for use in global climate models:', **How about the GLASS sea ice albedo and VIIRS sea ice albedo?**
and the relevant comment in the context of 'indicates that the two data sources (measurements and retrieval) are similar, but does not provide statistical evidence for the albedo product's reliability'. **The issue is, the insufficient or unreliable validation data does not means the algorithm/product is not reliable.**
**Response:**
We appreciate that you have brought the GLASS albedo to our attention; it was added to Table 1 and we also included a section to discuss the difference between the two retrieval products (Section 5.4).
In the revised version, Sections 1 and 2 have been reorganized to emphasize on the algorithmic difference rather than the use of validation data.

**Reviewer comment** in the context of 'can be applied to *any* optical sensor that measures appropriate radiance data'. **The word *any* needs to be moderated.**

and the relevant comment in the context of 'The accurate RTM ensures that the 'forward problem' is solved *correctly*' **Reviewer comment:** I would suggest moderate the word. Any model has its limitations. We cannot get the truth by simulation.

**Response:**

Noted and agree. The sentences have been modified in the revised version.

**Reviewer comment** MCD43: That is very limited coverage for a cross-comparison

**Response:**

We agree that the spatial coverage from MCD43 is not ideal. However, MCD43 actually is used by GLASS to 'validate' the visible and near-IR albedo retrieval, as mentioned on page 329 of the following manuscript.

*Liang, Shunlin, et al. "The global land surface satellite (GLASS) product suite." Bulletin of the American Meteorological Society 102.2 (2021): E323-E337.*

**Reviewer comment** in the context of "SciML models can be used to solve the 'inverse problem' ": **There might be ill-posed problems.**

**Response:**

We try to avoid the 'over-fitting' induced by ill-posed problems in two ways: (1) Generate a large synthetic dataset to provide a large size of training data. (2) Halt the training when the metrics (mean absolute error) of out-of-sample data does not improve.

**Reviewer comment:** Looks like a section 1 content. **on the 2.1 section, Existing albedo retrieval procedures and their constraints.**

**Response:**

In the revised version, Sections 1 and 2 have been reorganized

**Reviewer comment:** What are the value range of these properties? Table 2 only includes the IOPs of the surface condition. How about the atmospheric conditions used? Suggest more specific/detailed description about the atmospheric parameters. Are they independently sampled? What are the angle ranges? Are the angles all independently randomly sampled?

**Response:**

Addressed and replied in the previous section.

**Reviewer comment** in the context of 'compatible with any optical sensor capable of measuring TOA radiance in suitable wavelength channels, and the albedo map's spatial resolution matches that of the sensor footprint'**: This sentence seems irrelevant with this paragraph. Moreover, do you mean that the RTM model does not have a scale effect and could be used for all spatial resolutions? Or your training relationship could be used for all sensors? How about the spectral response function difference?**

**Response:**

In the revised version, this sentence is removed.

We hope the added flowchart in the revised version helps to explain why the framework is 'sensor agnostic'; for other sensors, retrieval can be obtained with the same framework as shown in in Fig. 1.

Adequate treatment of spectral response function is important, as discussed and shown in Figure 1 of the following paper: *Chen, Nan, et al. "Fast yet accurate computation of radiances in shortwave infrared satellite remote sensing channels." Optics express 25.16 (2017): A649-A664.*

**Reviewer comment:** Not sure why the authors want to emphasize a classification-avoided advantage here. The referred method do not need any classification step before the albedo retrieval. Make sure the citation is objective and you really understand the previous studies.

**Response:**

As addressed in the previous section, the MPD-based approach implicitly classifies pixels.

**Reviewer comment:** (1) Does the rank of the parameters influence the performance in the model?

(2) Which section is about comparison of the different MLANN models? What are the difference between the models?

(3) The statistics of bias, RMSE, and unbiased RMSE are desired.

**Response:**

(1) The values of the hyper-parameters influence model performance, but not to a large extent if the training is successful (i.e., gradient explosion or overfitting are avoided). Take the number of neurons in each hidden layer as an example. In our exploration, we found that a neural network model with two hidden layers and more neurons in each layer ($256 \times 512$) has comparable results to a model with three hidden layers but much fewer neurons ($16 \times 10 \times 8$). When analyzed with the Shapley value, the two models have learned

the same relations from the training data.

(2) Table 2 is included in the revised version to explain why there were two MLANN models.

(3) Table B2 in the Appendix shows the statistics of bias, RMSE, etc.

**Reviewer comment:** What is the differences between (c) and (d), albedometer vs. pyranometer?

**Response:** Addressed in the previous section.

**Reviewer comment:** It would be more clear to list a table to show the statistics, compared to other products.

**Response:**

Statistics have been added in the revised version (Table B3).

**3 Response to Reviewer 3**

**3.1 Specific Comments**

**Reviewer comment:** Table 1: Please add advantages and disadvantages of each product in the table.
**Response:**
In the revised version, the 'Description' column is removed and objective information (spatial/temporal resolution, temporal coverage, retrieval method) are included instead. The advantage/disadvantage of each product is included in main text in Section 5 and the algorithmic differences are included in Section 1.

**Reviewer comment:** P4, L 104: Please Just give some short explanation, as we don't see the paper ready to submit
 **Response:**
In the revised manuscript, this sentence was removed.

**Reviewer comment:** Please make data section and explain satellite used for the retrieval, validation dataset, comparison dataset before methodology.
**Response:**
We appreciate this suggestion. The revised version includes a 'Data' section (Section 2.5.4) that separates the backgrounds of satellite and validation data sources from the discussion of the results.

**Reviewer comment:** Section 2.2: I would be merited to have a flowchart to understand better.
**Response:**
We are grateful for the suggestion. A flowchart has been included (Fig. 1) in the revised version.

**Reviewer comment:** Section 2.4: The details of structure of MLANN must be addressed. For example, the number of layers, activation functions, weight initialization, input variables (should be synthetic dataset, SD), target variables, how to train and validate, accuracies.
**Response:**
The relevant text has been added to the revised version (Section 2.6, lines 275∼291).

**Reviewer comment:** P10, L251: the cloud screening method used MODIS bands? If it's right, how can it be used for SGLI?

**Response:**

The cloud screening and surface classification model is also sensor-agnostic. Table 3 is added in the revised version to clarify.

| | SGLI channels | | | MODIS channels | |
|---|---|---|---|---|---|
| $\lambda$ | albedo | cloud mask | $\lambda$ | albedo | cloud mask |
| 380 | | x | | | |
| 443 | x | x | 469 | x | x |
| 530 | x | x | 555 | x | x |
| 673.5 | | x | 645 | x | x |
| 868.5 | x | x | 858.5 | x | x |
| 1050 | x | x | 1240 | x | x |
| 1630 | x | x | 1640 | x | x |
| 2210 | x | x | 2130 | x | x |

Table 3: Central wavelengths used by SGLI and MODIS to retrieve albedo and obtain cloud and surface classification mask.

**Reviewer comment:** Figure 2: This figure should go data section.
**Response:**
Moved.

**Reviewer comment:** Figure 3: Can you explain what is the difference between c and d?
**Response:**
Table 2 is included in the 'Data' section of the revised version to show the difference between the equipments and the matching models.

**Reviewer comment:** Section 3.4: I don't understand the link between surface metamorphism and two days (Morning-noon-early afternoon, late afternoon) albedo changes. Figure 8 is not mentioned in section 3.4. If they have some links please elaborate more.
**Response:**
We appreciate your pointing out our omission of Figure 8 from the discussed text. In the revised version, the figure is referenced in the discussion context, and the pertinent texts have been revised to clarify (lines $428 \sim 432$).

**Reviewer comment:** Section 3.5: The retrieved albedo using SGLI is also comprehensively validated like a MODIS and analyzed with solar zenith angle, surface metamorphism. The retrieved albedo using SGLI should be validated and compared in parallel.

**Response:**

The GCOM-C/SGLI was launched in 2019 and data from the AFLUX campaign (Figure 5∼6) was the only validation data we could found. The purpose of including the results from SGLI is to show that the SciML/RTM framework (Fig. 1) is applicable to not just one sensor. Two pan-Arctic retrievals are included in Figure 19 (page 32) in the revised manuscript, and we hope the discussions in Section 5.4 can address the evaluations on this product. Comprehensive validation of SGLI is beyond the scope of the current paper and is discussed in a separate work by comparing the retrieval results with MODIS in the Sea of Okhotsk Region.

**Reviewer comment:** Section 4.1 and 4.2: In 4.1, albedo retrieval map against MCD is daily but in 4.2, 5-day mean albedo map against MERIS. Can you elaborate why they are different?

**Response:**

As noted in line 36∼41 (of the previous submission), Qu and Peng retrieved sea-ice albedo using the direct estimation method. Initially, we intended to utilize Qu's results as a benchmark for comparing the three algorithms. However, because we were unable to obtain the authors' original retrieval data in order to include it in the subplot, we could only show three columns in Figure 13 by the time we submitted the first version. For your reference, Fig. 2 below shows the comparison of the three products, and the first column are screenshots of the results of Qu's algorithm, taken directly from their paper. For the second and third column, we used the same color-bar to plot the results and manually boxed the same area, but due to difference in printing and in coordinates, the colors/regions don't exactly match with panels (a) and (b).

**Reviewer comment:** The retrieved albedo maps are only shown near Svalbard islands but Pan-Arctic retrieved albedo map should be shown and have to be compared with other comparison dataset.

**Response:**

We appreciate this comment. Pan-Arctic retrieval maps and the statistical comparisons are included in the revised version (Figure 19, 20 and B2).

**3.2 Minor Comments**

**Reviewer comment:** All captions in the table should be above table.

**Response:**

Done.

**Reviewer comment:** L 99, 100: Please mention SGLI MCD 43 full name
**Response:**
Revised. Second-generation Global Imager (SGLI), and Moderate Resolution Imaging Spectroradiometer (MODIS) MCD43D49, MCD43D50 and MCD43D51.

**Reviewer comment:** P4, L101-105: should go to the discussion section
**Response:**
Done.

**Reviewer comment:** 3 validaiton: The authors mentioned MOSAiC. Have you used the data from MOSAiC for the validation?
**Response:**
As described on line 304 (of the previous submission), "The MOSAiC campaign included fewer than 50 valid data points, and all obtained for broken cloud conditions. To eliminate errors caused by dense cloud cover, MOSAiC data were omitted from validation." This point is emphasized in lines 182 $\sim$ 185 of the revised manuscript.

**Reviewer comment:** L 497-499: should go comparison dataset
**Response:**
Done.

**4 Response to Reviewer 4**

**4.1 Overall comment**

**Reviewer comment:** AccuRT/RTM looks novel but needs sharing as open source as well as documentation to be subject to rigorous peer review. For example compare this situation with the RAMI experiments (Widlowski et al., 2007) or the MYSTIC cloud simulator (Mayer et al., 2010). Similarly, MLANN looks like a significant advance but again needs sharing as an open source resource to have any impact on the community. Very limited examples are not really a proper "validation" when the uncertainties are unknown of the "truth" data-sets. The authors do not present convincing evidence that MLNN will work on a time series of MODIS (let alone other instruments) to show the evolution of sea ice albedo during the Arctic spring/summer. They ignore the work of the NOAA group on VIIRS and the NASA group at UMD on the VIIRS-SNPP and MODIS time series and the UCL group on MISR instantaneous albedo retrievals all of which have long time series datasets publicly available which this paper does not. This technique and the paper is of high interest to the community but needs less hyperbole (on line 3 the authors claim there are no reliable albedo products, this reviewer would strongly dispute this) and more quantitative intercomparison with the aforementioned datasets before it can be considered for publication. Otherwise, this paper will represent cherry-picking results without any serious self-critical analysis.

**Response**

We appreciate the comments provided. Our main objective in this paper is to describe the methods and algorithms developed. Our radiative transfer codes have been extensively tested over many years and documented in several publications. We hope our paper will be well received by the community and we are open to make the tools available to interested users upon request. We have changed the word 'reliable' to 'unvalidated' to address the concern expressed by the reviewer.

**Reviewer comment:** 1. There is an incorrect assertion in the abstract: "there is currently no reliable, operational albedo retrieval product capable of assessing the global sea-ice albedo with sufficient spatial-temporal resolution for studies of sea-ice dynamics and for use in global climate models.

**Response:**

This sentence is removed in the revised version.

**Reviewer comment:** 2. NOAA have had an operational spectral and short-wave albedo product multiple times per day derived from NOAA-20 VIIRS since September 2018.

**Response:**

Note that in the previous submission, the NOAA VIIRS albedo product is discussed in Table 1 and line 36∼41, which is referred to as "Peng's direct-estimation (VIIRS)". This product has not been validated against actual sea ice. Instead, the only sea ice related reference in the product page is a paper discussing how they used data from Greenland Icesheet as 'sea ice validation'. The direct-estimation algorithm was also used in Qu's paper and as mentioned in line 36∼41: "Qu's validation used fewer than 50 matched retrieval-measurement data points during 90-day expedition, which does not provide statistical evidence for the albedo product's reliability."

In the revised version, sections 1 and 2 have been reorganized to stress on the algorithmic difference and any statements which might be regarded as hyperbole have been removed. In addition, Section 5.4 is added to compare with the GLASS and VIIRS SURFALB L3 albedo products.

**Reviewer comment:** 3. There are a bewildering number of acronyms that are not defined in the order that they are introduced. The paper needs to include a list of acronyms that the reader can consult. 4. One example is "comprehensive SD" on line 179 which is not defined previously. What is "SD"?

**Response:**

In the previous submission, the full-spelling of "synthetic dataset (SD)" was mentioned twice; the first appearance is on line 7 in Abstract and the second time on line 85, the fourth letter. In the current version, a list of acronyms is added at the end.

**Reviewer comment:** 5. The authors should provide evidence for the negligible differences of NIR and SW albedos for the differences given the upper wavelengths of 2.1µm, 2.5µm, and 3µm (lines 278-279)

**Response:**

The point of 'small error due to wavelength range difference' is line 272, which refers to the difference between a neural network model that estimates broadband albedo defined in the range of 0.3∼2.8 $\mu$m and the pyranometer measurements which yield broadband irradiance in the range of 0.2∼3.6 $\mu$m. The 'negligible difference' between these two is a fact. For line 278-279, these refer to a different model that can be used to compare with the albedometer

measurements. In the revised version, a 'Data' section is separated and Table 2 which explains the wavelength ranges of the two models we trained is included to avoid confusion.

**Reviewer comment:** 6. Absolute albedo is not very helpful when the range in albedos is so large. It is better to show the coefficient of variation (stdv/mean) to see how the albedo varies in uncertainties. (Line 437)
**Response:**
Statistics are included in Figure 4 (Pearson-r, RMSE, number of pixels with estimation error smaller than 15%), Figure 6 (number of pixels with estimation error smaller than 15%, mean absolute error), and Table B2 (Pearson-r, RMSE, mean absolute error, mean absolute percentage error, bias, number of pixels with estimation error smaller than 15%, mean absolute error).

**Reviewer comment:** 7. The so-called validation shown here is usually referred to as stage 1 (CEOS-WGCV-LPV, see https://lpvs.gsfc.nasa.gov/) as there are very limited dates and there is no uncertainty specified for the aircraft measurements.
**Response:**
In the previous submission, the uncertainties of the instruments (pyranometer and albedometer) were mentioned on line 318-320 with reference. 'The uncertainty of the "SMART Albedometer" was reported to be 7%, whereas the pyranometer's uncertainty is less than 3% (Grobner2014new, Ehrlich2019comprehensive).' This sentence is also included in the revised version, on lines 193~194 (top of Page 9).

In this paper we present and discuss a newly developed albedo-retrieval framework that is different from the direct-estimation method and the melt-pond-detection (MPD)-based approach. When the two methodologies were first brought up, the MPD-based approach was only validated with MELTEX measurements with less than 300 data points [11], whereas the albedo retrieved with SciML/RTM framework is validated with ∼9000 data points against pyranometer and ∼4000 data points against albedometer measurements. As for the direct-estimation approach, the VIIRS sea-ice albedo product was not validated against actual sea-ice albedo at all, and the 'methodology paper' [12] published two years prior to the VIIRS product did not have statistically-
* * *
[11]Istomina, L., et al. "Melt pond fraction and spectral sea ice albedo retrieval from MERIS data-Part 1: Validation against in situ, aerial, and ship cruise data." The Cryosphere 9.4 (2015): 1551-1566.

[12]Qu Y, Liang S, Liu Q, et al. Estimating Arctic sea-ice shortwave albedo from MODIS

significant validation data (fewer than 50 data points) to prove the stage-1 validation.

We believe that at the current phase, the validation as discussed in this paper is sufficient to make the scientific community aware of the SciML/RTM methodology. We are currently working with the GCOM-C team to deploy the albedo-retrieval model as a product for interested users to access the SGLI retrievals. The 'h5' files of the MODIS retrieval model will be published on PANGAEA once this paper is finalized.

**Reviewer comment:** 8. Why was MLANN not adapted for uses with SGLI, VIIRS, and OLCI?

**Response:**
The MLANN *was* adapted for use with SGLI. Figures 5 and 6 show validation results against AFLUX-campaign measurements. The same retrieval from MODIS is also shown in Figs. 5-6 to demonstrate that the SciML/RTM methodology is applicable to optical sensors. In Section 5.4, pan-Arctic figures from SGLI, MODIS using the SciML/RTM framework can further strengthen this point.

Adaptation to VIIRS and OLCI is beyond the scope of this 'methodology' paper, but could be considered in the future.

**Reviewer comment:** 9. Also, what about comparisons with the OLCI product derived using the Kokhanovsky et al. 2020 (Line 687) SNAP processor?

**Response**
Kokhanovsky et al. 2020 presented an algorithm for snow parameter retrievals, which is a 'snow-on-land' algorithm. Kokanonsky's algorithm was validated not against sea-ice measurements, but with measurements from Greenland Icesheet and compared with MODIS MOD10A1 (also a 'snow-on-land' albedo product).

The SciML/RTM presented in this work is a 'sea ice-albedo' algorithm. The two scenarios are not directly comparable. 'Sea ice' refers to a sheet of ice floating on ocean water, which might be covered by snow or melt-ponds. The comparable cases are the MPD-based algorithm and the direct-estimation method, which were designed for application to sea ice surfaces.

**Reviewer comment:** 10. Where are the open-source repositories of AccuRT and the RTM/SciML as well as MLANN?
* * *
data[J]. Remote Sensing of Environment, 2016, 186: 32-46.

**Response:**

AccuRT is not an open-source software, but interested users may contact Knut Stamnes for a copy of the software. The 'h5' files of the MODIS-retrieval model will be uploaded to PANGAEA with python code showing how to load the 'MOD021KM' file and 'MOD03' file to provide input to obtain surface albedo. The SGLI-retrieval product will be made available by the end of 2022 on the JAXA page and users may directly download the retrieval results rather than manually run the retrieval model.

**4.2 Annotations**

**Reviewer comment:** Note [page 1]: Line 3: NOAA have had an operational DAILY spectral and shortwave validated albedo product derived from VIIRS since September 2018. There is also a paper which describes the sea ice product specifically which you reference below from Peng et al. (2018) but which you ignore in your paper. Where is the evidence that this is not reliable and operational? Is your proposed product operational? This sentence should be modified.

**Response:**

As addressed in the response in the previous section, Peng's algorithm is not neglected. In the revised version, this sentence is removed from the abstract and Section 5.4 is included to discuss the comparison results as well as algorithmic differences.

**Reviewer comment:** Note [page 1]: Line 9: But neither does the MISR (Kharbouche & Muller, 2018) nor does the GLASS product both of which are produced from instantaneous measurements.

**Response:**

The focus of the 'comparison' part in this paper is mainly about comparing the SciML/RTM framework with the currently operating algorithm or product used for sea ice albedo retrieval. From the algorithm perspective, the direct-estimation method (GLASS and VIIRS) and the melt-pond-detection (MPD) algorithm (MERIS and OLCI) both were designed for sea ice albedo retrieval and are the most up-to-date approaches.

MISR was not included mainly because the algorithm of MISR albedo product uses a spectral-to-broadband albedo conversion equation to retrieve broadband albedo directly from surface reflectance. The factors for conversion was debeloped by Dr. Shunlin Liang more than 20 years ago and uses

only four spectral bands. The developers came up with the direct-estimation approach in recent years, which takes into account all possible surface types, uses more spectral bands, and is considered a better and more precise approach that substitutes the simple form of conversion equation. The limited number of spectral bands available from MISR means it is rather difficult to apply the direct-estimation approach to this sensor. From the 2018 paper and the May 7 2020 slide that the authors of MISR used at EGU (entitled 'Mapping Antarctic sea ice albedo properties from MISR fused with MODIS'), retrieval is only made in the periods of 2000∼2016.

As for GLASS, from the product documentation[13], the algorithm of GLASS is Qu's algorithm (direct-estimation), which is listed and discussed.

We appreciate that the two products are specifically brought up; they are included in Table 1 in the revised version.

**Reviewer comment:**Note [page 1]: Define acronym
**Response:**
In the revised version, Second-generation Global Imager (SGLI) and Moderate Resolution Imaging Spectroradiometer (MODIS) are spelled in full.

**Reviewer comment:** Note [page 1]: Line 14: This is not a very helpful measure of error if you don't provide the range and mean?
**Response:**
The 'mean absolute error' included in the abstract is a summary of the results from Fig. 5(e), and the data range as well as other statistics are discussed: (a) in the text, (b) in Figure 3, and in Table A2.
In the revised version, this sentence is replaced with the following text to include more information:

Excellent agreement was found between the RTM/SciML albedo-retrieval results and measurements collected from airplane campaigns. Assessment against pyranometer data ($N = 4144$) yields RMSE $= 0.094$ for the shortwave albedo retrieval, while evaluation against albedometer data ($N = 1225$) yields RMSE $= 0.069, 0.143, 0.085$ for the broadband albedo in the visible, near infrared, and shortwave spectral ranges, respectively.

**Reviewer comment:**Note [page 1]: Line 23: Extent? Thickness? Concentration? Which attribute is in decline?
**Response:**
* * *
[13]Liang, Shunlin, et al. "The global land surface satellite (GLASS) product suite." Bulletin of the American Meteorological Society 102.2 (2021): E323-E337.

This sentence was removed in the revised manuscript.

**Reviewer comment:** Note [page 2]: Spatial resolution? Note [page 2]: What is the resolution? Note [page 2]: Spatial resolution? Note [page 2]: Omits MISR products from Kharbouche & Muller (2018) Also, needs spatial and temporal resolution and time range adding as well as URLs of where the product is described and available. Note [page 2]: Table 1 is very poor. Needs consistency in spatial resolution, needs a column for time range for which they are available. Needs an additional column for validation level (see CEOS comment later)

**Response:**

We appreciate this comment, especially the CEOS comment for guidance. This spatial resolution, temporal resolution and the operation period are included in the revised version; the validation level is described in the Data section to provide more context (Section 2.5.4).

**Reviewer comment:** Strikeout [page 2]: (2018)) ground truth instead of ground truths

**Response:**

Revised.

**Reviewer comment:** Note [page 2]: L38: This is because there are no reliable long-term measurements of sea ice albedo publicly available. and a relevant comment, **Reviewer comment:** Note [page 2]: Line 41: But that is true of all the so-called validation exercises including your aircraft data. This I sonly for a few dates, can be up to 5 hours different in time with the satellite overpass and dos not have any uncertainties associated with the aircraft measurements.

**Response:** We appreciate the two comments. In the revised version, Sections 1 and 2 have been reorganized to emphasize on the algorithmic difference rather than the use of validation data. These sentences were removed.

**Reviewer comment:** Strikeout [page 3]: compared to with, repetitive word.

**Response:**

Revised.

**Reviewer comment:** Note [page 5]: Line 118: Define acronym: NTBC, IOP, SGLI, MLCM

**Response:**

The term 'narrow-to-broadband conversion (NTBC)' was defined on line 54. The term 'inherent optical propserties (IOPs)' is defined on line 157.

Full spelling of SGLI is included in the revised version.

The term 'machine learning classification mask (MLCM)' is defined on line 257.

We appreciate these comments, and an 'Acronyms' section is included at the end listing all the abbreviations in alphabetical order to avoid confusion.

**Reviewer comment:** Note [page 8]: Line 185: all the parameters need to be elaborated in a table as this is an open journal. Also, is AccuRT open source? And what about the retrieval method?

**Response:**

Physical parameters of ice, melt water on ice, and ocean water, physical parameters of snow cover, geometries, and atmospheric characteristics (Appendix A), as well as a flowchart (Figure 1) has been included in the revised version.

**Reviewer comment:** Note [page 8]: Line 195: Are these available? Where are they described?

**Response:**

The validation data we used (MODIS-channel radiance, angles, as well as the measured broadband albedo averaged to MODIS grid and the time-difference between MODIS transit and measurements) will be uploaded to PANGAEA once this paper is finalized. Relevant text of the products that will be uploaded (validation data, MODIS albedo retrieval model, and python script to use the retrieval model) has been added to the 'Data Availability' section of the revised version to set the correct expectation.

The original data source which was used to derive the validation data is described in Section 2.5.2.

**Reviewer comment:** Note [page 8]: Line 203: Reference needed for the independent surface classification model

**Response:**

Added, Chen et al. (2018). Details of this model are included in Section 2.4.

**Reviewer comment:** Note [page 8]: Line 208: need reference and/or URL for this unknown sensor.

**Response:**

Added the JAXA page in which GCOM-C/SGLI is described.

**Reviewer comment:** Note [page 8]: Line 209: It is disappointing that this sensor was not examined as it could then be compared against the operational VIIRS product from Peng.

**Response:**

Included in Section 5.4.

**Reviewer comment:** Note [page 9]: Line 215: What does the L stand for?

**Response:**

We appreciate this comment. The full spelling which gives 'L' was deleted when we were revising the paper. The abbreviation is corrected in the revised version:

Level-1 and Atmosphere Archive and Distribution System (LAADS) Distributed Active Archive Center (DAAC).

**Reviewer comment:** Note [page 11]: Line 281: What is this footprint? How is the difference in resolution dealt with? Aggregation?

**Response:**

The exact value of footprint was not provided by the scientists who recorded these data. Based on the speed of the aircraft and the lat-lon information of the recorded data, we found that 150~180 measurements are matched to a 1-km distance. Therefore, as explained in the following sentence, *the estimated albedo is collocated with the MODIS grid and the average value of about 170 measurements from each flight is mapped to a single MODIS pixel.*

**Reviewer comment:** Note [page 11]: Line 295: Where does this significant decrease come from? H2O absorption?

**Response:**

Yes.

**Reviewer comment:** Note [page 12]: Line 311: The visible results do show the lowest value of r and slope. The authors should comment on why these produce the worst results.

**Response:**

We currently do not have a good reason why the visible and near-inrared results show higher error than the shortwave broadband.

**Reviewer comment:** Note [page 13]: Line 326: how fast did the sea ice move over the time period between the MODIS observation and the aircraft observation? It is likely that the poorer disagreement is due to the fact that the same piece of sea ice is not observed by the aircraft. and the relevant comment, **Reviewer comment:** Note [page 14]: Figure 3: caption: What is the time range shown here between these 2 sets of measurements?

**Response:**

Ice drift is an error source that was considered in the study, and when

only 1.5-hour of time difference is allowed, the error due to ice drift, melting/refreezing is minimized (shown in Figure 8). The time difference of the data presented in Figure 3 (label from the previous submission) is in the range of 2∼5.

**Reviewer comment:** Note [page 16]: Line 378: This is difficult to believe as most sea ice moves at >10 km/day at this time of year.

**Response:**
From the RGB images, the sea ice discussed in this subsection indeed did not show apparent ice drifting. We included eight figures in a zip file that shows the retrievals and RGB. The filenames indicate the date of year and time in UTC of the MODIS images and retrievals. The eight files have been compiled to a gif animation and is included in the current submission.

**Reviewer comment:** Note [page 21]: Line 441: Remind the reader what MPD is and define in a list of acronyms.

**Response:**
Added.

**Reviewer comment:** Note [page 22]: Figure 10: What does EE mean? Define in the caption.

**Response:**
We appreciate this comment. The full spelling of expected error (EE) was included in the captions of Figures 3 and 5 as well as line 430, but was missed in the caption of Figure 10. It is added in the revised version.

**Reviewer comment:** Note [page 23]: Lines 460-461: Is this upper range of wavelength for n2b significant?

**Response:**
We appreciate this comment. The upper bound is reasoned on line 274, 'the contribution to the albedo for wavelengths beyond 2.5 $\mu$m is negligible'.

Therefore, the difference between the upper bound of MCD43 (5$\mu$m) and that of our retrieval (2.8$\mu$m) is not significant.

**Reviewer comment:** Note [page 27]: Figure 14 caption: Why is the OLCI retrieval so much coarser in spatial resolution? Note [page 28]: Line 529: Why on earth was this done?

**Response:**
This is the choice of the authors who developed the MERIS and OLCI retrieval algorithms; only 12.5-km resolution data is provided to the public.

We sent requests for the pre-gridded retrieval files of these days but did not hear back from the authors.

**Reviewer comment:** Note [page 29]: Line 55: this is hyperbole. Where is this demonstrated? I only see MODIS & SGLI results.

**Response:**

We appreciate this comment. The word 'any' is removed; the application of this framework to other optical sensors stays in theory until retrieval products of all sensors have been developed.

**Reviewer comment:** Note [page 29]: L567: Why is this important? What impact does this have?

**Response:**

A Look-up-table is essentially a linear regression model, which does not learn the possible interactions between the input features (geometry angles and radiance/reflectance values from various channels). We found that the trained models use different channels and relations to retrieve the albedo of snow and ice surface. This topic is discussed in a separate paper that will be submitted shortly, in which we used the Shapley Value to deduce how these models compute albedo based on input channels and geometry angles.

**Reviewer comment:** Note [page 29]: Line 574: What is a whole image? A 5-minute MODIS Level-1B data granule?

**Response:**

Yes. To avoid confusion on 'over an entire image from a satellite sensor', the text is altered to the following:

Once a RTM/SciML model has been properly trained, it takes only a few seconds to make retrievals on the Level-1B data granule.

**Reviewer comment:** Note [page 29]: Line 585: But so are MISR (which uses MODIS cloud masks) and VIIRS & MODIS (e.g. GLASS) direct estimation algorithms?

**Response:**

This sentence was rephrased to the following. "... albedo retrievals based on multi-platform satellite sensors can significantly increase the amount of valid and accurate observational data, thereby increasing spatial and temporal coverage regardless of the specific method of retrieval."

**Reviewer comment:** Note [page 30]: Line 588: EGU journals should only permit open access datasets with a publication DOI. In addition, all software

should be open access. This is what differentiates EGU from other comparable journals. This should not be an exception.
**Response:**
We appreciate this comment. Links to PANGAEA will be included in the final revised version.

**Reviewer comment:** Note [page 30]: Table A1 caption: Where does these percentages come from?
**Response:**
We appreciate this comment. The citation for MLCM algorithm which produces cloud filtering and surface classification and the lat-lon range of campaign operation is added in the caption in the revised version (latitudes in the range of 77.8$\sim$82.4°N , and longitude in the range of -0.25$\sim$20.5°E).

**Reviewer comment:** Note [page 33]: Line 597: Exact URLs should be provided. Note [page 33]: Line 600: Grant numbers should be listed.
**Response:** We are sorting out the codes and files to upload to PANGAEA. The text will be added/modified in the final version.

**Appendix**

| Parameter | Sym. | Unit | Value |
|---|---|---|---|
| Sea-ice thickness | $h$ | m | $0 \sim 3$ |
| Brine pocket volume fraction | $V_{\mathrm{br}}$ | – | $(-0.067 \cdot \log(h) + 0.1147) \cdot (1 + 0.2 \cdot r_{\mathrm{bu}})$ |
| Brine pocket radius | $r_{\mathrm{br}}$ | $\mu$m | $300 \sim 700$ |
| Air bubble volume fraction | $V_{\mathrm{bu}}$ | – | $0.0214 \cdot h + 0.0068$ |
| Air bubble radius | $r_{\mathrm{bu}}$ | $\mu$m | $-18.3 \cdot h^2 + 222.7 \cdot h + 96.5$ |

Table 4: Physical parameters of ice. In generating the sea-ice thickness, a truncated-normal distribution with $\mu = 0.03$, $\sigma = 1.5$ was used to ensure an adequate amount of thin ice in the SD. The brine pocket radius conforms to a Tukey-Lamdba distribution with $\lambda = 0.5$.

| Parameter | Units | Value |
|---|---|---|
| Melt water thickness | m | $0 \sim 1.5$ |
| Chlorophyll concentrations | mg/m$^3$ | $0.5 \sim 10$ |
| CDOM at 443 nm | /m | $0.01 \sim 0.1$ |

Table 5: Physical parameters of melt water on ice and ocean water. Melt water thickness and CDOM values follow randomly-distributed uniform distributions in the specified ranges. For the chl-a concentration, a reciprocal continuous distribution (long tail extending to high values) was used.

| Parameter | Symbol | Units | Value |
|---|---|---|---|
| Snow grain size | $r_{\mathrm{e}}$ | $\mu$m | $50 \sim 150$ |
| Snow density | $\rho_s$ | kg/m$^3$ | 200 |
| Impurity fractions | $f_{\mathrm{imp}}$ | - | $10^{-7} \sim 10^{-6}$ |
| Snow thickness | $h_{\mathrm{snow}}$ | m | $0.01 \sim 0.2$ |

Table 6: Physical parameters of snow cover. The snow grain size and snow thickness were generated with a randomly uniform distribution in the specified ranges.

| Parameters | Value |
|---|---|
| Solar zenith angle | 20~80 degrees |
| Sensor angle | 0.01~50 degrees |
| Azimuth angle | 0.01~180 degrees |
| AOD at 500 nm | $0.01 \sim 0.3$ |
| Relative humidity | 0.5 |
| Fine mode fraction | 0.9 |

Table 7: Geometries and atmospheric parameters. All parameters conform to random-uniform distributions in the specified ranges.

---

## Author Response (AR4)

**1 First round of revision**

**1.1 Response to Reviewer 1**

**1.1.1 Specific Comments**

**Reviewer comment:** Although detailed information of the coupled radiative transfer model AccRT can be found in the literatures, I suggest to add a concise description about it in the manuscript.
**Response:**
The coupled radiative transfer model (AccuRT) was described twice, on line 76~84 and 156~179 (including Table 2), which summarizes the two in-text references, Stamnes et al., 2011 and 2018. However, because the material is split across two sections that span three complete pages, we may have missed communicating the concept clearly. Sections 1 and 2 have been reorganized in the revised version.

**Reviewer comment:** The method of how to construct the synthetic dataset (SD) with the coupled RTM is not clear. The detailed information about the inherit optical properties (IOPs) listed in Table 2 are needed, such as the data ranges, probability distribution, and constraints.
**Response:**
Physical parameters of ice, melt water on ice, and ocean water, physical parameters of snow cover, geometries, and atmospheric characteristics has been included in the revised version (Appendix A).

**Reviewer comment:** The framework of the RTM/MLANN is not clear. I suggest to add a flowchart for it.
**Response:**
We are grateful for the suggestion. A flowchart has been included (Fig. 1) in the revised version.

**Reviewer comment:** In the manuscript, the MLANN method to used estimate the sea ice albedo. What are the performances of training, validating, and predicting accuracies of this artificial neural network model?
**Response:**
We have included the following statistics to the revised version of this paper: the RMSEs in training, out-of-sample validation and validation with ACLOUD data are: 0.006, 0.063, 0.099, respectively.

[Figure]

Figure 1: Flowchart of the proposed RTM/SciML framework for albedo retrieval.

**Reviewer comment:** The authors declared that the sensor-agnostic albedo retrieval method has the ability to apply to any optical sensor, however few explanations about this are shown in the manuscript. I suggest the authors to further explain the major theories of this method. In fact, other methods such as the MPD and direct-estimation algorithm, can also be adopted to other sensors easily. Please add a discussion about it.

**Response:**

The missing piece we did not include in the submitted version is how we determined the 'appropriate' channels from any optical sensor. Apart from the coupled RTM model that calculates the synthetic dataset, we use a technique to ensure that the radiance we use in practice (input from satellite channels) is consistent with that in our training data, allowing our machine learning models to be directly applied to satellite data.

The following contents have been added to the revised paper (Section 2.3).

With our knowledge of radiative transfer theory and the differences in the radiative properties of the constituents in the coupled atmosphere-surface system, we first chose the input channels based on the following criteria:

- Avoid wavelengths with significant absorption by water vapor and/or other atmospheric constituents.

- Avoid sensor channels that have been found to be saturated in previous sensitivity investigations.

- Select wavelengths that, based on their albedo spectra, can best identify snow cover, bare ice, open water, and melt-pond surface.

With the assistance of the auto-associative neural network (AANN) technique, channels with a significant reconstruction error are deemed unsuitable for use as input to the retrieval model. More specifically, an AANN is trained using the synthetic data generated by radiative transfer model (RTM), which takes as input the three sun-satellite geometry angles as well as all radiance data that meets the aforementioned requirements and outputs all radiances. The trained AANN is believed to have picked up on the patterns in the RTM-generated dataset. Following that, the AANN is fed the same input features derived from the satellite sensor. We calculate the absolute percentage error of the reconstruction output and prune channels with an error greater than 5%.

This method is intended to avoid 'covariate shift' — a phrase used in machine learning to refer to the difference between independent variables in training and real-world data. Covariate shift is due to either (a) the saturation of certain satellite channels, which results in a much narrower dynamic range of radiance data from the satellite sensor (real world) than that calculated using RTM (training data), or (b) the response function and wide wavelength range results in a non-negligible difference between the radiance derived from the central wavelength and that obtained from the sensor. It has been demonstrated that the AANN technique is effective in detecting mismatches between data acquired for the retrieval task and data utilized for training. A recent paper[1] discusses how the AANN approach was used to identify optimal channels for retrieving ocean color products using a variety of sensors.

Similar approaches have been used to identify acceptable channels for albedo retrieval. Table 1 lists the MODIS channels that were utilized to retrieve albedo, as well as the GCOM-C/SGLI channels that were evaluated and eventually employed [2].
* * *
[1] *Fan, Yongzhen, et al. "OC-SMART: A machine learning based data analysis platform for satellite ocean color sensors." Remote Sensing of Environment 253 (2021): 112236.*

[2] Our team initially discovered the saturation issue in the 673.5 nm channel using AANN, submitted it to the GCOM-C/SGLI team, and obtained confirmation of the issue.

| $\lambda$ of MODIS | $\lambda$ of SGLI |
|:---:|:---:|
| 469 | 443 |
| 555 | 530 |
| 645 | 673.5* |
| 858.5 | 868.5 |
| 1240 | 1050 |
| 1640 | 1630 |
| 2130 | 2210 |

Table 1: A comparison of the centre wavelengths of the channels explored and tested for the purpose of retrieving the broadband albedo at the cryosphere surface using MODIS and SGLI sensors. The 673.5 nm wavelength channel from SGLI (shown by an asterisk in the table) was proven to be saturated and hence was not used to derive albedo. Details can be found in the revised version.

**Reviewer comment:** The comparisons with MCD43D, MERIS, and OLCI datasets were not easily for reader to interpret. I suggested to add scatter plots to compare the differences of these datasets.

**Response:**

We appreciate your suggestion. Scatter plots were omitted due to the fact that the retrievals did not cover equivalent areas. For example, our cloud classification mask is stricter than those employed by OLCI and MERIS. The melt-pond detection approach does not provide values for open-water areas. Similarly, the MCD43 product returns values just along the coast. As a result, we anticipated scatter plots would be less useful than albedo maps. However, as you noted, quantitative depiction is helpful. Figures 14, 16, 19 and 20 have been added in the revised version.

**Reviewer comment:** Figure 13, the authors declared that the MERIS albedo product are higher than the albedo estimated by the MLANN method in the areas with large melt pond fraction (greater than 50%). However, this difference is not obvious, and the major differences appeared in the upper right corner. Please provide an explanation for it.

**Response:**

Other than the wavelength difference (MERIS sensor does not have short-wave near-infrared channels), we believe the discrepancy in the area covered by snow is due to algorithmic difference.

In our training data (the synthetic dataset created by the radiative transfer model), we explore the situation of snow-covered sea ice with snow depths ranging from 0.01∼0.2 m to 50∼150 $\mu$m (Table **??**). Notably, 0.2 meter is the optically thick snow threshold. We found that the trained models use different channels and relations to retrieve the albedo of snow and ice surface. This topic is discussed in a separate paper that will be submitted shortly, in which we used the Shapley Value to deduce how these models compute albedo based on input channels and geometry angles.

The MPD-based algorithm does not discriminate between snow-covered sea ice and bare ice when retrieving albedo; both scenarios are classified as 'ice', and for both snow and ice surface, the same iterative method and the same channels were used to calculate spectral albedo. The spectral albedo is subsequently integrated to obtain broadband albedo.

In the revised version, details of how the three approaches (SciML/RTM, direct-estimation and MPD) differ in the treatment of snow-covered ice, melt-pond and bare-ice were discussed in more detail in Section 1.

**Reviewer comment:** Figure 13, the measurements of campaigns were not shown in this figure. Why? Please add the validation data for comparison.
**Response:**
The reason is that there are no campaign-measured data available to validate these results. The two locations (top and bottom) and time period (averaged between DOY 166 and DOY 170 in 2007) selected to compare the results from our algorithm and those from MERIS are the same as those used in the subsequent article:

*Qu Y, Liang S, Liu Q, et al. Estimating Arctic sea-ice shortwave albedo from MODIS data[J]. Remote Sensing of Environment, 2016, 186: 32-46.*

As noted in line 36∼41, Qu and Peng retrieved sea-ice albedo using the direct estimation method. Initially, we intended to utilize Qu's results as a benchmark for comparing the three algorithms. However, because we were unable to obtain the authors' original retrieval data in order to include it in the subplot, we could only show three columns in Figure 13. For your reference, Fig. 2 below shows the comparison of the three products, and the first column are screenshots of the results of Qu's algorithm, taken directly from their paper. For the second and third column, we used the same color-bar to plot the results and manually boxed the same area, but due to difference in printing and in coordinates, the colors/regions don't exactly match with panels (a) and (b).

[Figure]

Figure 2: Maps of albedo and melt pond fraction averaged during a 5-day period in 2007 between DOY 166 and 170. From left to right: Qu's albedo retrievals, MLANN-based and MPD-based albedo retrievals, as well as the MPD-derived melt pond fraction, respectively (Qu2015Mapping, this study, and Istomina2015Melt). The upper panels depict the Banks, Prince Patrick, and Melville Islands, while the lower panels depict the Kara Sea. At the bottom, colorbars representing the corresponding values are displayed. Note that the images of Qu's retrieval results (along with the colorbar) are taken directly from Fig.10 in Qu2015Mapping, as no other data was obtainable. In panels (c) and (d), empty regions represent cloud pixels that were detected by the MLCM model (and hence removed), whereas empty regions in panels (e) through (h) represent either cloud pixels or open-water areas that were not processed by the MPD algorithm.

**Reviewer comment:** In the abstract, the mean absolute error (MAE) of 0.047 was used for indicating the accuracy of this method. I suggest to use root mean standard error (RMSE) to represent the estimation accuracies for the visible, near infrared, and shortwave albedo.

**Response:**

The text was modified as follow to include the statistics shown in Table A2 in abstract.

Assessment against pyranometer data (N = 4144) yields RMSE = 0.094 for the shortwave albedo retrieval, while evaluation against albedometer data (N = 1225) yields RMSE = 0.069, 0.143, 0.085 for the broadband albedo in the visible, near infrared, and shortwave spectral ranges, respectively.

**1.1.2 Technical Corrections**

**Reviewer comment:** Figure 7. The color ramp of this figure is not easily to interpret. Please change it.

**Response:**

A figure (Fig. 10) with more distinguishable colors is presented in the next version.

**Reviewer comment:** Line 493, the sentences of "Istomina et al. (2015); Istomina (2020)" can be rewritten as "Istomina et al. (2015; 2020)".

**Response:** Revised.

**Reviewer comment:** Caption of Figure 13. "(Qu et al. (2015), this study, and Istomina et al. (2015))". The reference Qu et al. 2015 is not related with this figure.

**Response:**

Revised.

**1.2 Response to Reviewer 2**

**1.2.1 Question & Suggestion**

**Reviewer comment:** The coupled RTM is used to simulate TOA reflectance from various sea ice surface and atmospheric properties. The surface parameters are listed but the values were not mentioned as well as the sampling strategy. I cannot figure out how the authors determine the distribution and relevance among the parameters. Similar concern for the atmospheric parameters and the solar/view angles.

**Response:**

Physical parameters of ice, melt water on ice, and ocean water, physical parameters of snow cover, geometries, and atmospheric characteristics have been included in the revised version (Appendix A).

**Reviewer comment:** The machine learning method needs more detailed description about how it was used. How to deal with the invalid retrievals from the relationship? Is there a post-processing? It was mentioned there are two models trained. What are their difference and advantages?

**Response to 'how post/pre-processing were used to avoid invalid retrievals':**

A flowchart (Figure 1) illustrating the process of obtaining a final retrieval model was included in the revised version (Fig. 1).

The training dataset only contains snow, ice and melt-pond surface types. As a result, a machine learning classification mask (MLCM) [3] is employed as a post-processing step to filter out invalid pixels.

In addition to 'post-processing', we use auto-associative neural network (AANN) technique as a 'feature selection' tool to avoid the machine learning model obtains invalid retrievals. AANN can effectively avoid 'covariate shift' and ensure that the data range of radiance we use in practice (input from satellite channels) is consistent with that in our training data (synthetic dataset derived from radiative transfer model).

**The following paragraphs are included in the revised version to elaborate on the AANN (Section 2.3).**

With the assistance of the auto-associative neural network (AANN) technique, channels with a significant reconstruction error are deemed unsuitable
* * *
[3]Chen, Nan, et al. "New neural network cloud mask algorithm based on radiative transfer simulations." Remote Sensing of Environment 219 (2018): 62-71.

for use as input to the retrieval model. More specifically, an AANN is trained using the synthetic data generated by radiative transfer model (RTM), which takes as input the three sun-satellite geometry angles as well as all radiance data that meets the aforementioned requirements and outputs all radiances. The trained AANN is believed to have picked up on the patterns in the RTM-generated dataset. Following that, the AANN is fed the same input features derived from satellite sensor. We calculate the absolute percentage error of the reconstruction output and prune channels with an error greater than 5%.

This method is intended to avoid 'covariate shift' — a phrase used in machine learning to refer to the difference between independent variables in training and real-world data. Covariate shift is due to either (a) the saturation of certain satellite channels, which results in a much narrower dynamic range of radiance data from the satellite sensor (real world) than that calculated using RTM (training data), or (b) the response function and wide wavelength range results in a non-negligible difference between the radiance derived from the central wavelength and that obtained from the sensor. It has been demonstrated that the AANN technique is effective in detecting mismatches between data acquired for the retrieval task and data utilized for training. A recent paper[4] discusses how the AANN approach was used to identify optimal channels for retrieving ocean color products using a variety of sensors.

**Response to 'difference between the two neural networks'**

This article discussed two neural network models, the only distinction being the wavelength range of the broadband albedo output. The pyranometer and the albedometer have distinct ranges of wavelengths. The albedometer measures spectral irradiance (up/down) in the range of 0.4∼2.1 $\mu$m, whereas the pyranometer measures broadband irradiance (up/down) in the range of 0.3∼3.6 $\mu$m. As a result, we trained two distinct models to make the most use of the data from the two types of equipment. The revised version includes a 'Data' section that separates the backgrounds of satellite and validation data sources from the discussion of the results. Table 2 is included to show the difference between the equipment types and the matching models.

**Reviewer comment:** The author emphasized many times about the ad-
* * *
[4] *Fan, Yongzhen, et al. "OC-SMART: A machine learning based data analysis platform for satellite ocean color sensors." Remote Sensing of Environment 253 (2021): 112236.*

|  | Model 1 | | Model 2 | |
| --- | --- | --- | --- | --- |
|  | $\lambda$ range (nm) | validation data | $\lambda$ range (nm) | validation data |
| Visible | 300-700 | / | 400-700 | albedometer |
| Near Infrared | 700-2500 | / | 700-2100 | albedometer |
| Shortwave | 300-2500 | pyranometer | 400-2100 | albedometer |

Table 2: Difference between the two models mentioned in the text. Figures 3, 6 and Table A2 show retrieval and validation results of the two models.

vantage of the proposed method than the previous MPD or direct-estimation methods. However, many descriptions needs to be clarified or discussed more. What are the advantage of the coupled RTM rather than the separate radiative transfer models? Is there any quantitative comparison about this? Is a classification within sea-ice surface needed in previous methods? The method is claimed as independent on sensor or spatial resolution, how that is realized without considering the spectral response function difference? Did the previous method restrict to a specific spatial resolution?

**Response:**

**The following text has been added to the revised version to discuss the advantages of utilizing a coupled RTM over decoupled models.**

Due to the fact that uncoupled RTMs make disparate assumptions about the atmosphere and surface when constructing the BRDF, albedo retrieval with uncoupled RTMs is difficult to scale.

While MPD-based approaches are capable of retrieving both the albedo and melt pond percentage for a surface composed of melt water and white ice, they are only effective during particular seasons (discussed in [5] [6] [7]). Additionally, because the spectral reflection coefficients for the melt-pond and thin ice boundaries, as well as the thick ice and snow-cover boundaries, are manually adjusted based on the surface condition, there are greater uncer-
* * *
[5]Istomina, L., et al. "Melt pond fraction and spectral sea ice albedo retrieval from MERIS data–Part 2: Case studies and trends of sea ice albedo and melt ponds in the Arctic for years 2002–2011." The Cryosphere 9.4 (2015): 1567-1578.

[6]Istomina, L., et al. "Melt pond fraction and spectral sea ice albedo retrieval from MERIS data–Part 1: Validation against in situ, aerial, and ship cruise data." The Cryosphere 9.4 (2015): 1551-1566.

[7]Zege, E., et al. "Algorithm to retrieve the melt pond fraction and the spectral albedo of Arctic summer ice from satellite optical data." Remote Sensing of Environment 163 (2015): 153-164.

tainties in the retrieval during the transitional seasons of spring-summer and summer-autumn, as well as when the surface is highly heterogeneous (low sea ice concentration, discussed in Istomina2015Melt). Two more issues with the method are its omission of open-water conditions and restriction of sea ice type. The MPD algorithm models sea ice's BRDF exclusively for dry and white ice (based on the absorption of yellow pigments), ignoring the effects of air bubbles and brine pockets (discussed in Zege2015Algorithm).

In the direct-estimation method, linear relations between TOA reflectance and surface albedo are derived for different angular bins. Qu2015Mapping [8] and Peng2018VIIRS [9] generated datasets detailing the intervals of geometry angles in their models, which already have over 40,000 combinations. Multiplying the RTM-simulated or measured surface BRDFs (which range between 100,000 and 120,000 for their retrievals) by the possible atmospheric configurations and by the geometry-angle dataset, results in an extremely big value for the look-up table (LUT). However, note that only hundreds-thousand level surface conditions were characterized using the monstrous LUT, and because a LUT is essentially a linear regression model, the LUT does not learn the possible interactions between the input features (geometry angles and radiance/reflectance values from various channels), making the approach less efficient.

It is worth noting that the 'RTM-simulated surface BRDFs' mentioned earlier for estimating the radiative properties of sea-ice and snow surface was derived from the IOP model, which is employed by the coupled-RTM that we are utilizing [10]. Because Qu's algorithm decouples the atmosphere from the ocean layer, it is unable to accurately simulate the 'snow-covered sea ice' situation. In a coupled-RTM, snow is simulated as a layer of snow floating on the surface above the interface (the upper slab of the coupled system), and sea ice is simulated as a layer of ice with brine pockets and air bubble inclusions floating on deep ocean water (the lower slab of the coupled system).

Although Qu used the same IOP model to calculate the optical properties

[8]Qu, Ying, et al. "Mapping surface broadband albedo from satellite observations: A review of literatures on algorithms and products." Remote Sensing 7.1 (2015): 990-1020.

[9]Peng, Jingjing, et al. "The VIIRS sea-ice albedo product generation and preliminary validation." Remote Sensing 10.11 (2018): 1826.

[10]Stamnes, Knut, et al. "Modeling of radiation transport in coupled atmosphere-snow-ice-ocean systems." Journal of Quantitative Spectroscopy and Radiative Transfer 112.4 (2011): 714-726. was cited in the relevant sections

of the two media, the 'snow surface' scenario refers to snow that has been placed on land. Additionally, snow possesses complex surface cover (that varies with density, impurity inclusions, thickness, and effective grain size), but the LUT only included a snow layer of a fixed depth. The same issue exists with regard to sea ice conditions. Due to the fact that the direct-estimation algorithm is very dependent on the quality of the LUT, Qu's model's snow and ice retrieval is rather crude in comparison to the more refined approach used in this study.

One last methodological difference is, when using the direct estimation method and building a BRDF table, the reflectance anisotropy (namely, the strong forward peak of snow and ice), which occurs when the solar zenith angle equals the sensor zenith angle, is manually corrected offline (see Qu2016Estimating). However, in our coupled-RTM, the forward peak can be adjusted inside the radiative transfer calculations (described in [11]).

**Quantitative comparison between a coupled and decoupled model?**
**Response:**
Unfortunately, there currently is no literature comparing the performance between a 'coupled' and a 'decoupled' RTM. This topic is also out of the scope of the current paper.

**Is a classification within sea-ice surface needed in previous methods?**
To clarify, the classification in our framework is a 'post-processing' step; it is conducted independently and has no effect on the albedo retrieval results.

For both the direct-estimation method and the MPD-based method, other than cloud-screening, they did not have a post–processing classification algorithm to identify the surface type of each pixel after the surface albedo is retrieved. However, the logic of MPD algorithm requires the surface to be classified and obtains spectral albedo afterwards. In 2.3.2 of Zege's paper, it was mentioned that three channels are used to separate and discard open water pixels, two channels are used to separate the white surface (snow and ice are considered as the same category) from melt pond. The added contents (from the previous response) should help to clarify this issue.

**Clarification on the spatial resolution**
* * *
[11] Jiang, Shigan, et al. "Enhanced solar irradiance across the atmosphere-sea ice interface: a quantitative numerical study." Applied optics 44.13 (2005): 2613-2625.

Other than the MCD43 product, all the other three methods discussed can all retrieve albedo based on single-angular observations. MCD43 requires data from multiple days and therefore the spatial and temporal resolution are lower.

**1.2.2   Minor Comments**

**Reviewer comment** in the context of 'but there is currently no reliable, operational albedo retrieval product capable of assessing the global sea-ice albedo with sufficient spatial-temporal resolution for studies of sea-ice dynamics and for use in global climate models:', **How about the GLASS sea ice albedo and VIIRS sea ice albedo?**
and the relevant comment in the context of 'indicates that the two data sources (measurements and retrieval) are similar, but does not provide statistical evidence for the albedo product's reliability'. **The issue is, the insufficient or unreliable validation data does not means the algorithm/product is not reliable.**
**Response:**
We appreciate that you have brought the GLASS albedo to our attention; it was added to Table 1.
In the revised version, Sections 1 and 2 have been reorganized to emphasize on the algorithmic difference rather than the use of validation data.

**Reviewer comment** in the context of 'can be applied to *any* optical sensor that measures appropriate radiance data'. **The word *any* needs to be moderated.**
and the relevant comment in the context of 'The accurate RTM ensures that the 'forward problem' is solved *correctly*' **Reviewer comment:** I would suggest moderate the word. Any model has its limitations. We cannot get the truth by simulation.
**Response:**
Noted and agree. The sentences have been modified in the revised version.

**Reviewer comment**  MCD43: That is very limited coverage for a cross-comparison
**Response:**
We agree that the spatial coverage from MCD43 is not ideal. However, MCD43 actually is used by GLASS to 'validate' the visible and near-IR albedo retrieval, as mentioned on page 329 of the following manuscript.

*Liang, Shunlin, et al. "The global land surface satellite (GLASS) product suite." Bulletin of the American Meteorological Society 102.2 (2021): E323-E337.*

**Reviewer comment** in the context of "SciML models can be used to solve the 'inverse problem' "**: There might be ill-posed problems.**
**Response:**
We try to avoid the 'over-fitting' induced by ill-posed problems in two ways:
(1) Generate a large synthetic dataset to provide a large size of training data.
(2) Halt the training when the metrics (mean absolute error) of out-of-sample data does not improve.

**Reviewer comment:** Looks like a section 1 content. **on the 2.1 section, Existing albedo retrieval procedures and their constraints.**
**Response:**
In the revised version, Sections 1 and 2 have been reorganized

**Reviewer comment:** What are the value range of these properties?
Table 2 only includes the IOPs of the surface condition. How about the atmospheric conditions used?
Suggest more specific/detailed description about the atmospheric parameters.
Are they independently sampled?
What are the angle ranges? Are the angles all independently randomly sampled?
**Response:**
Addressed and replied in the previous section.

**Reviewer comment** in the context of 'compatible with any optical sensor capable of measuring TOA radiance in suitable wavelength channels, and the albedo map's spatial resolution matches that of the sensor footprint'**: This sentence seems irrelevant with this paragraph. Moreover, do you mean that the RTM model does not have a scale effect and could be used for all spatial resolutions? Or your training relationship could be used for all sensors? How about the spectral response function difference?**
**Response:**
In the revised version, this sentence is removed.
We hope the added flowchart in the revised version helps to explain why the framework is 'sensor agnostic'; for other sensors, retrieval can be obtained

with the same framework as shown in in Fig. 1.

Adequate treatment of spectral response function is important, as discussed and shown in Figure 1 of the following paper: *Chen, Nan, et al. "Fast yet accurate computation of radiances in shortwave infrared satellite remote sensing channels." Optics express 25.16 (2017): A649-A664.*

**Reviewer comment:** Not sure why the authors want to emphasize a classification-avoided advantage here. The referred method do not need any classification step before the albedo retrieval. Make sure the citation is objective and you really understand the previous studies.

**Response:**

As addressed in the previous section, the MPD-based approach implicitly classifies pixels.

**Reviewer comment:** (1) Does the rank of the parameters influence the performance in the model?

(2) Which section is about comparison of the different MLANN models? What are the difference between the models?

(3) The statistics of bias, RMSE, and unbiased RMSE are desired.

**Response:**

(1) The values of the hyper-parameters influence model performance, but not to a large extent if the training is successful (i.e., gradient explosion or overfitting are avoided). Take the number of neurons in each hidden layer as an example. In our exploration, we found that a neural network model with two hidden layers and more neurons in each layer ($256 \times 512$) has comparable results to a model with three hidden layers but much fewer neurons ($16 \times 10 \times 8$ ). When analyzed with the Shapley value, the two models have learned the same relations from the training data.

(2) Table 2 is included in the revised version to explain why there were two MLANN models.

(3) Table B1 and Table B2 in the Appendix shows the statistics of bias, RMSE, etc.

**Reviewer comment:** What is the differences between (c) and (d), albedometer vs. pyranometer?

**Response:** Addressed in the previous section.

**Reviewer comment:** It would be more clear to list a table to show the statistics, compared to other products.

**Response:**

Statistics were added in the revised version.

**1.3 Response to Reviewer 3**

**1.3.1 Specific Comments**

**Reviewer comment:** Table 1: Please add advantages and disadvantages of each product in the table.
**Response:**

In the revised version, the 'Description' column is removed and objective information (spatial/temporal resolution, temporal coverage, retrieval method) are included instead. The advantage/disadvantage of each product is included in main text.

**Reviewer comment:** P4, L 104: Please Just give some short explanation, as we don't see the paper ready to submit
 **Response:**
The following is added to the revised version: Using Shapley value to interpret different SciML models trained using the synthetic dataset, it was found that models with good performance have learned the spectral difference between snow, ice and water pixels.

**Reviewer comment:** Please make data section and explain satellite used for the retrieval, validation dataset, comparison dataset before methodology.
**Response:**
We appreciate this suggestion. The revised version includes a 'Data' section that separates the backgrounds of satellite and validation data sources from the discussion of the results.

**Reviewer comment:** Section 2.2: I would be merited to have a flowchart to understand better.
**Response:**
We are grateful for the suggestion. A flowchart has been included (Fig. 1) in the revised version.

**Reviewer comment:** Section 2.4: The details of structure of MLANN must be addressed. For example, the number of layers, activation functions, weight initialization, input variables (should be synthetic dataset, SD), target variables, how to train and validate, accuracies.
**Response:**
The following text has been added to the revised version.
The adaptive moment estimation (Adam) was chosen to update weights and

biases in an MLANN, which is trained in 200 epochs with a batch size of 64. A MLANN's hyperparameters include the learning rate and the activation function. To determine the optimal learning rate, Bayesian optimization was employed, and the Rectified Linear Units (ReLU) were used as the activation function in the hidden layers. Batch normalization is performed to enhance the MLANN's generalization capabilities and make the network less sensitive to random initialization of the weights and biases. To avoid overfitting, dropout layers were included as a regularization for networks with more than two hidden layers. In our evaluation, dropout layers with a rate of 0.2 were optimal, implying that one in every five inputs is randomly eliminated from each update cycle. A hidden-layer structure of $(16 \times 10 \times 8)$ was found to perform effectively with input data from both SGLI and MODIS sensors.

**Reviewer comment:** P10, L251: the cloud screening method used MODIS bands? If it's right, how can it be used for SGLI?
**Response:**
The cloud screening and surface classification model is also sensor-agnostic. Table 3 is added in the revised version to clarify.

| | SGLI channels | | | MODIS channels | |
|---|---|---|---|---|---|
| $\lambda$ | albedo | cloud mask | $\lambda$ | albedo | cloud mask |
| 380 | | x | | | |
| 443 | x | x | 469 | x | x |
| 530 | x | x | 555 | x | x |
| 673.5 | | x | 645 | x | x |
| 868.5 | x | x | 858.5 | x | x |
| 1050 | x | x | 1240 | x | x |
| 1630 | x | x | 1640 | x | x |
| 2210 | x | x | 2130 | x | x |

Table 3: Central wavelengths used by SGLI and MODIS to retrieve albedo and obtain cloud and surface classification mask.

**Reviewer comment:** Figure 2: This figure should go data section.
**Response:**
Moved.

**Reviewer comment:** Figure 3: Can you explain what is the difference between c and d?
**Response:**

Table 2 is included in the 'Data' section of the revised version to show the difference between the equipments and the matching models.

**Reviewer comment:** Section 3.4: I don't understand the link between surface metamorphism and two days (Morning-noon-early afternoon, late afternoon) albedo changes. Figure 8 is not mentioned in section 3.4. If they have some links please elaborate more.

**Response:**

We appreciate your pointing out our omission of Figure 8 from the discussed text. In the revised version, the figure is referenced in the discussion context. In addition, the text below was added to Section 3.4 to provide more context.

Similar to how the Eulerian flow field is specified, the 'surface metamorphism' of sea ice can be studied by analyzing the albedo change at fixed locations. By subtracting the albedo on the first day from that at the same location on the following day, the albedo change over the last 24 hours due to metamorphism can be determined; a positive $\Delta_\alpha$ at a fixed pixel indicates that the melt-pond (or open-water) has refrozen (or frozen), while a negative $\Delta_\alpha$ indicates ice (or snow) has melted. Notably, the subtractions are carried out at similar solar zenith angles (morning to morning, noon to noon, etc.), which eliminates the effect of solar zenith angle on albedo change.

**Reviewer comment:** Section 3.5: The retrieved albedo using SGLI is also comprehensively validated like a MODIS and analyzed with solar zenith angle, surface metamorphism. The retrieved albedo using SGLI should be validated and compared in parallel.

**Response:**

The GCOM-C/SGLI was launched in 2019 and data from the AFLUX campaign (Figure 9) was the only validation data we could found. The purpose of including the results from SGLI is to show that the SciML/RTM framework (Fig. 1) is applicable to not just one sensor. Comprehensive validation of SGLI is beyond the scope of the current paper and is discussed in a separate work by comparing the retrieval results with MODIS in the Sea of Okhotsk Region.

**Reviewer comment:** Section 4.1 and 4.2: In 4.1, albedo retrieval map against MCD is daily but in 4.2, 5-day mean albedo map against MERIS. Can you elaborate why they are different?

**Response:**

As noted in line 36~41, Qu and Peng retrieved sea-ice albedo using the direct estimation method. Initially, we intended to utilize Qu's results as a

benchmark for comparing the three algorithms. However, because we were unable to obtain the authors' original retrieval data in order to include it in the subplot, we could only show three columns in Figure 13 by the time we submitted the first version. For your reference, Fig. 2 below shows the comparison of the three products, and the first column are screenshots of the results of Qu's algorithm, taken directly from their paper. For the second and third column, we used the same color-bar to plot the results and manually boxed the same area, but due to difference in printing and in coordinates, the colors/regions don't exactly match with panels (a) and (b).

**Reviewer comment:** The retrieved albedo maps are only shown near Svalbard islands but Pan-Arctic retrieved albedo map should be shown and have to be compared with other comparison dataset.
**Response:**
We appreciate this comment. A Pan-Arctic retrieval map was added in the revised version (Fig. 19).

**1.3.2   Minor Comments**

**Reviewer comment:** All captions in the table should be above table.
**Response:**
Done.

**Reviewer comment:** L 99, 100: Please mention SGLI MCD 43 full name
**Response:**
Revised. Second-generation Global Imager (SGLI), and Moderate Resolution Imaging Spectroradiometer (MODIS) MCD43D49, MCD43D50 and MCD43D51.

**Reviewer comment:** P4, L101-105: should go to the discussion section
**Response:**
Done.

**Reviewer comment:** 3 validaiton: The authors mentioned MOSAiC. Have you used the data from MOSAiC for the validation?
**Response:**
As described on line 183, "The MOSAiC campaign included fewer than 50 valid data points, and all obtained for broken cloud conditions. To eliminate errors caused by dense cloud cover, MOSAiC data were omitted from validation."

**Reviewer comment:** L 497-499: should go comparison dataset
**Response:**
Done.

**1.4 Response to Reviewer 4**

**1.4.1 Overall comment**

**Reviewer comment:** AccuRT/RTM looks novel but needs sharing as open source as well as documentation to be subject to rigorous peer review. For example compare this situation with the RAMI experiments (Widlowski et al., 2007) or the MYSTIC cloud simulator (Mayer et al., 2010). Similarly, MLANN looks like a significant advance but again needs sharing as an open source resource to have any impact on the community. Very limited examples are not really a proper "validation" when the uncertainties are unknown of the "truth" data-sets. The authors do not present convincing evidence that MLNN will work on a time series of MODIS (let alone other instruments) to show the evolution of sea ice albedo during the Arctic spring/summer. They ignore the work of the NOAA group on VIIRS and the NASA group at UMD on the VIIRS-SNPP and MODIS time series and the UCL group on MISR instantaneous albedo retrievals all of which have long time series datasets publicly available which this paper does not. This technique and the paper is of high interest to the community but needs less hyperbole (on line 3 the authors claim there are no reliable albedo products, this reviewer would strongly dispute this) and more quantitative intercomparison with the aforementioned datasets before it can be considered for publication. Otherwise, this paper will represent cherry-picking results without any serious self-critical analysis.

**Response**

We appreciate the comments provided. Our main objective in this paper is to describe the methods and algorithms developed. Our radiative transfer codes have been extensively tested over many years and documented in several publications. We hope our paper will be well received by the community and we are open to make the tools available to interested users upon request. We have changed the word 'reliable' to 'unvalidated' to address the concern expressed by the reviewer.

**Reviewer comment:** 1. There is an incorrect assertion in the abstract: "there is currently no reliable, operational albedo retrieval product capable of assessing the global sea-ice albedo with sufficient spatial-temporal resolution for studies of sea-ice dynamics and for use in global climate models.

**Response:**

This sentence is removed in the revised version.

**Reviewer comment:** 2. NOAA have had an operational spectral and short-wave albedo product multiple times per day derived from NOAA-20 VIIRS since September 2018.

**Response:**

Note that the NOAA VIIRS albedo product is discussed in Table 1 and line 36∼41, which is referred to as "Peng's direct-estimation (VIIRS)"). This product has not been validated against for actual sea-ice. Instead, the only sea ice related reference in the product page is a paper discussing how they used data from Greenland Icesheet as 'sea ice validation'. The direct-estimation algorithm was also used in Qu's paper and as mentioned in line 36∼41: "Qu's validation used fewer than 50 matched retrieval-measurement data points during 90-day expedition, which does not provide statistical evidence for the albedo product's reliability."

In the revised version, sections 1 and 2 have been reorganized to stress on the algorithmic difference and any statements which might be regarded as hyperbole will be removed.

**Reviewer comment:** 3. There are a bewildering number of acronyms that are not defined in the order that they are introduced. The paper needs to include a list of acronyms that the reader can consult. 4. One example is "comprehensive SD" on line 179 which is not defined previously. What is "SD"?

**Response:**

The full-spelling of "synthetic dataset (SD)" was mentioned twice; the first appearance is on line 7 in Abstract and the second time on line 85, the fourth letter. A list of acronyms is added in the revised version.

**Reviewer comment:** 5. The authors should provide evidence for the negligible differences of NIR and SW albedos for the differences given the upper wavelengths of 2.1µm, 2.5µm, and 3µm (lines 278-279)

**Response:**

The point of 'small error due to wavelength range difference' is line 272, which refers to the difference between a neural network model that estimates broadband albedo defined in the range of 0.3∼2.8 $\mu$m and the pyranometer measurements which yield broadband irradiance in the range of 0.2∼3.6 $\mu$m. The 'negligible difference' between these two is a fact. For line 278-279, these refer to a different model that can be used to compare with the albedometer measurements. In the revised version, a 'Data' section is separated and Table 2 which explains the wavelength ranges of the two models we trained is

included to avoid confusion.

**Response:**
Statistics are included in Figure 4 (Pearson-r, RMSE, number of pixels with estimation error smaller than 15%), Figure 6 (number of pixels with estimation error smaller than 15%, mean absolute error), and Table B2 (Pearson-r, RMSE, mean absolute error, mean absolute percentage error, bias, number of pixels with estimation error smaller than 15%, mean absolute error).

**Reviewer comment:** 7. The so-called validation shown here is usually referred to as stage 1 (CEOS-WGCV-LPV, see https://lpvs.gsfc.nasa.gov/) as there are very limited dates and there is no uncertainty specified for the aircraft measurements.
**Response:**
The uncertainties of the equipments (pyranometer and albedometer) are mentioned on line 318-320 with reference. 'The uncertainty of the "SMART Albedometer" was reported to be 7%, whereas the pyranometer's uncertainty is less than 3% (Grobner2014new, Ehrlich2019comprehensive).'

In this paper we present and discuss a newly developed albedo-retrieval framework that is different from the direct-estimation method and the melt-pond-detection (MPD)-based approach. When the two methodologies were first brought up, the MPD-based approach was only validated with MELTEX measurements with less than 300 data points [12], whereas the albedo retrieved with SciML/RTM framework is validated with ∼9000 data points against pyranometer and ∼4000 data points against albedometer measurements. As for the direct-estimation approach, the VIIRS sea-ice albedo product was not validated against actual sea-ice albedo at all, and the 'methodology paper' [13] published two years prior to the VIIRS product did not have statistically-significant validation data (fewer than 50 data points) to prove the stage-1 validation.

We believe that at the current phase, the validation as discussed in this
* * *
[12]Istomina, L., et al. "Melt pond fraction and spectral sea ice albedo retrieval from MERIS data-Part 1: Validation against in situ, aerial, and ship cruise data." The Cryosphere 9.4 (2015): 1551-1566.

[13]Qu Y, Liang S, Liu Q, et al. Estimating Arctic sea-ice shortwave albedo from MODIS data[J]. Remote Sensing of Environment, 2016, 186: 32-46.

paper is sufficient to make the scientific community aware of the SciML/RTM methodology. We are currently working with the GCOM-C team to deploy the albedo-retrieval model as a product for interested users to access the SGLI retrievals. The 'h5' files of the MODIS retrieval model will be published on PANGAEA once this paper is finalized.

**Reviewer comment:** 8. Why was MLANN not adapted for uses with SGLI, VIIRS, and OLCI?

**Response:**

The MLANN *was* adapted for use with SGLI. Figures 9 and 10 show validation results against AFLUX-campaign measurements. The same retrieval from MODIS is also shown in Figs. 9-10 to demonstrate that the SciML/RTM methodology is applicable to optical sensors.

Adaptation to VIIRS and OLCI is beyond the scope of this 'methodology' paper, but could be considered in the future.

**Reviewer comment:** 9. Also, what about comparisons with the OLCI product derived using the Kokhanovsky et al. 2020 (Line 687) SNAP processor?

**Response**

Kokhanovsky et al. 2020 presented an algorithm for snow parameter retrievals, which is a 'snow-on-land' algorithm. Kokanonsky's algorithm was validated not against sea-ice measurements, but with measurements from Greenland Icesheet and compared with MODIS MOD10A1 (also a 'snow-on-land' albedo product).

The SciML/RTM presented in this work is a 'sea ice-albedo' algorithm. The two scenarios are not directly comparable. 'Sea ice' refers to a sheet of ice floating on ocean water, which might be covered by snow or melt-ponds. The comparable cases are the MPD-based algorithm and the direct-estimation method, which were designed to work with the sea ice surface.

**Reviewer comment:** 10. Where are the open-source repositories of AccuRT and the RTM/SciML as well as MLANN?

**Response:**

AccuRT is not an open-source software, but interested users may contact Knut Stamnes for a copy of the software. The 'h5' files of the MODIS-retrieval model will be uploaded to PANGAEA with python code showing how to load the 'MOD021KM' file and 'MOD03' file to provide input to obtain surface albedo. The SGLI-retrieval product will be made available in

mid-2022 on the JAXA page and users may directly download the retrieval results rather than manually run the retrieval model.

**1.4.2 Annotations**

**Reviewer comment:** Note [page 1]: Line 3: NOAA have had an operational DAILY spectral and shortwave validated albedo product derived from VIIRS since September 2018. There is also a paper which describes the sea ice product specifically which you reference below from Peng et al. (2018) but which you ignore in your paper. Where is the evidence that this is not reliable and operational? Is your proposed product operational? This sentence should be modified.

**Response:**
As addressed in the response in the previous section, Peng's algorithm is not neglected. In the revised version, this sentence is removed from the abstract.

**Reviewer comment:** Note [page 1]: Line 9: But neither does the MISR (Kharbouche & Muller, 2018) nor does the GLASS product both of which are produced from instantaneous measurements.

**Response:**
The focus of the 'comparison' part in this paper is mainly about comparing the SciML/RTM framework with the currently operating algorithm or product used for albedo retrieval. From the algorithm perspective, the direct-estimation method (used by MODIS and VIIRS) and the melt-pond-detection (MPD) algorithm (MERIS and OLCI) both were designed for sea ice albedo retrieval and are the most up-to-date approaches. Therefore, they are specifically mentioned in the abstract.

MISR was not included mainly because the algorithm of MISR albedo product uses a spectral-to-broadband albedo conversion equation to retrieve broadband albedo directly from surface reflectance. The factors for conversion was debeloped by Dr. Shunlin Liang more than 20 years ago and uses only four spectral bands. The developers came up with the direct-estimation approach in recent years, which takes into account all possible surface types, uses more spectral bands, and is considered a better and more precise approach that substitutes the simple form of conversion equation. The limited number of spectral bands available from MISR means it is rather difficult to apply the direct-estimation approach to this sensor. From the 2018 paper and the May 7 2020 slide that the authors of MISR used at EGU (entitled 'Mapping Antarctic sea ice albedo properties from MISR fused with MODIS'),

retrieval is only made in the periods of 2000∼2016.

As for GLASS, from the product documentation[14], the algorithm of GLASS is Qu's algorithm (direct-estimation), which is listed and discussed.

We appreciate that the two products are specifically brought up; they will be included in Table 1 in the revised version.

**Reviewer comment:**Note [page 1]: Define acronym
**Response:**
In the revised version, Second-generation Global Imager (SGLI) and Moderate Resolution Imaging Spectroradiometer (MODIS) are spelled in full.

**Reviewer comment:** Note [page 1]: Line 14: This is not a very helpful measure of error if you don't provide the range and mean?
**Response:**
The 'mean absolute error' included in the abstract is a summary of the results from Fig. 5(e), and the data range as well as other statistics are discussed: (a) in the text, (b) in Figure 3, and in Table A2.
In the revised version, this sentence is replaced with the following text to include more information:

In comparison to the ACLOUD campaign's albedometer measurements, the 3936 pixels of albedo retrieved under clear skies have RMSE values of 0.076, 0.137, and 0.087 in the visible, near-infrared, and short-wave bands, respectively. The RMSE is 0.099 when 7964 clear-sky pixels are compared to pyranometer observations from two aircraft during the ACLOUD campaign. The best agreement was reached on June 25th, 2017, when the campaign region experienced the least cloud cover.

**Reviewer comment:**Note [page 1]: Line 23: Extent? Thickness? Concentration? Which attribute is in decline?
**Response:**
The papers cited in the context included proofs of both decline in extent (Stroeve's 2007 paper) and decline in thickness (the other three sources). This sentence is revised to the following to be more exact:

It is not new that Arctic sea ice has been on the decline in the past years, in terms of both extent and thickness.

**Reviewer comment:** Note [page 2]: Spatial resolution? Note [page 2]: What is the resolution? Note [page 2]: Spatial resolution? Note [page 2]:
* * *
[14]Liang, Shunlin, et al. "The global land surface satellite (GLASS) product suite." Bulletin of the American Meteorological Society 102.2 (2021): E323-E337.

Omits MISR products from Kharbouche & Muller (2018) Also, needs spatial and temporal resolution and time range adding as well as URLs of where the product is described and available. Note [page 2]: Table 1 is very poor. Needs consistency in spatial resolution, needs a column for time range for which they are available. Needs an additional column for validation level (see CEOS comment later)

**Response:**
We appreciate this comment, especially the CEOS comment for guidance. This table is revised as suggested.

**Reviewer comment:** Strikeout [page 2]: (2018)) ground truth instead of ground truths

**Response:**
Revised.

**Reviewer comment:** Note [page 2]: L38: This is because there are no reliable long-term measurements of sea ice albedo publicly available. and a relevant comment, **Reviewer comment:** Note [page 2]: Line 41: But that is true of all the so-called validation exercises including your aircraft data. This I sonly for a few dates, can be up to 5 hours different in time with the satellite overpass and dos not have any uncertainties associated with the aircraft measurements.

**Response:** We appreciate the two comments. In the revised version, Sections 1 and 2 have been reorganized to emphasize on the algorithmic difference rather than the use of validation data. These sentences will be removed.

**Reviewer comment:** Strikeout [page 3]: compared to with, repetitive word.
**Response:**
Revised.

**Reviewer comment:** Note [page 5]: Line 118: Define acronym: NTBC, IOP, SGLI, MLCM

**Response:**
The term 'narrow-to-broadband conversion (NTBC)' was defined on line 54.
The term 'inherent optical propserties (IOPs)' is defined on line 157.
Full spelling of SGLI is included in the revised version.
The term 'machine learning classification mask (MLCM)' is defined on line 257.

We appreciate these comments, and an 'Acronyms' section will be included at the end listing all the abbreviations in alphabetical order to avoid

confusion.

**Reviewer comment:** Note [page 8]: Line 185: all the parameters need to be elaborated in a table as this is an open journal. Also, is AccuRT open source? And what about the retrieval method?

**Response:**

Physical parameters of ice, melt water on ice, and ocean water, physical parameters of snow cover, geometries, and atmospheric characteristics, as well as a flowchart has been included in the revised version. As a reference, this section is included at the close of this response for your review (Tables ??∼??), Figure 1) prior to our submission of the revised version.

**Reviewer comment:** Note [page 8]: Line 195: Are these available? Where are they described?

**Response:**

The validation data we used (MODIS-channel radiance, angles, as well as the measured broadband albedo averaged to MODIS grid and the time-difference between MODIS transit and measurements) will be uploaded to PANGAEA once this paper is finalized. Relevant text of the products that will be uploaded (validation data, MODIS albedo retrieval model, and python script to use the retrieval model) has been added to the 'Data Availability' section of the revised version to set the correct expectation.

The original data source which was used to derive the validation data is described on line 266∼270.

**Reviewer comment:** Note [page 8]: Line 203: Reference needed for the independent surface classification model

**Response:**

Added, Chen et al. (2018). Details of this model are included in Section 2.5 (line 250∼258).

**Reviewer comment:** Note [page 8]: Line 208: need reference and/or URL for this unknown sensor.

**Response:**

Added the JAXA page in which GCOM-C/SGLI is described.

**Reviewer comment:** Note [page 8]: Line 209: It is disappointing that this sensor was not examined as it could then be compared against the operational VIIRS product from Peng.

**Response:**

We will be working on applying the same framework to VIIRS and proceed with this comparison.

**Reviewer comment:** Note [page 9]: Line 215: What does the L stand for?
**Response:**
We appreciate this comment. The full spelling which gives 'L' was deleted when we were revising the paper. The abbreviation is corrected in the revised version:

Level-1 and Atmosphere Archive and Distribution System (LAADS) Distributed Active Archive Center (DAAC).

**Reviewer comment:** Note [page 11]: Line 281: What is this footprint? How is the difference in resolution dealt with? Aggregation?
**Response:**
The exact value of footprint was not provided by the scientists who recorded these data. Based on the speed of the aircraft and the lat-lon information of the recorded data, we found that 150~180 measurements are matched to a 1-km distance. Therefore, as explained in the following sentence, *the estimated albedo is collocated with the MODIS grid and the average value of about 170 measurements from each flight is mapped to a single MODIS pixel.*

**Reviewer comment:** Note [page 11]: Line 295: Where does this significant decrease come from? H2O absorption?
**Response:**
Yes.

**Reviewer comment:** Note [page 12]: Line 311: The visible results do show the lowest value of r and slope. The authors should comment on why these produce the worst results.
**Response:**
We currently do not have a good reason why the visible and near-inrared results show higher error than the shortwave broadband.

**Reviewer comment:** Note [page 13]: Line 326: how fast did the sea ice move over the time period between the MODIS observation and the aircraft observation? It is likely that the poorer disagreement is due to the fact that the same piece of sea ice is not observed by the aircraft. and the relevant comment, **Reviewer comment:** Note [page 14]: Figure 3: caption: What is the time range shown here between these 2 sets of measurements?
**Response:**
As discussed in line 323~327, ice drift is an error source that was considered

in the study, and when only 1.5-hour of time difference is allowed, the error due to ice drift, melting/refreezing is minimized (shown in Figure 5). Line 325 shows that the time difference of the data presented in Figure 3 is in the range of 2∼5

**Reviewer comment:** Note [page 16]: Line 378: This is difficult to believe as most sea ice moves at >10 km/day at this time of year.

**Response:**

From the RGB images, the sea ice discussed in this subsection indeed did not show apparent ice drifting. We included eight figures in a zip file that shows the retrievals and RGB. The filenames indicate the date of year and time in UTC of the MODIS images and retrievals.

**Reviewer comment:** Note [page 21]: Line 441: Remind the reader what MPD is and define in a list of acronyms.

**Response:**

Added.

**Reviewer comment:** Note [page 22]: Figure 10: What does EE mean? Define in the caption.

**Response:**

We appreciate this comment. The full spelling of expected error (EE) was included in the captions of Figures 3 and 5 as well as line 430, but was missed in the caption of Figure 10. It is added in the revised version.

**Reviewer comment:** Note [page 23]: Lines 460-461: Is this upper range of wavelength for n2b significant?

**Response:**

We appreciate this comment. The upper bound is reasoned on line 274, 'the contribution to the albedo for wavelengths beyond 2.5 $\mu$m is negligible'.

Therefore, the difference between the upper bound of MCD43 ($5\mu$m) and that of our retrieval ($2.8\mu$m) is not significant.

**Reviewer comment:** Note [page 27]: Figure 14 caption: Why is the OLCI retrieval so much coarser in spatial resolution? Note [page 28]: Line 529: Why on earth was this done?

**Response:**

This is the choice of the authors who developed the MERIS and OLCI retrieval algorithms; only 12.5-km resolution data is provided to the public. We sent requests for the pre-gridded retrieval files of these days but did not hear back from the authors.

**Reviewer comment:** Note [page 29]: Line 55: this is hyperbole. Where is this demonstrated? I only see MODIS & SGLI results.

**Response:**

We appreciate this comment. The word 'any' is removed; the application of this framework to other optical sensors stays in theory until retrieval products of all sensors have been developed.

**Reviewer comment:** Note [page 29]: L567: Why is this important? What impact does this have?

**Response:**

A Look-up-table is essentially a linear regression model, which does not learn the possible interactions between the input features (geometry angles and radiance/reflectance values from various channels). We found that the trained models use different channels and relations to retrieve the albedo of snow and ice surface. This topic is discussed in a separate paper that will be submitted shortly, in which we used the Shapley Value to deduce how these models compute albedo based on input channels and geometry angles.

**Reviewer comment:** Note [page 29]: Line 574: What is a whole image? A 5-minute MODIS Level-1B data granule?

**Response:**

Yes. To avoid confusion on 'over an entire image from a satellite sensor', the text is altered to the following:

Once a RTM/SciML model has been properly trained, it takes only a few seconds to make retrievals on the Level-1B data granule.

**Reviewer comment:** Note [page 29]: Line 585: But so are MISR (which uses MODIS cloud masks) and VIIRS & MODIS (e.g. GLASS) direct estimation algorithms?

**Response:**

This sentence was rephrased to the following. "... albedo retrievals based on multi-platform satellite sensors can significantly increase the amount of valid and accurate observational data, thereby increasing spatial and temporal coverage regardless of the specific method of retrieval."

**Reviewer comment:** Note [page 30]: Line 588: EGU journals should only permit open access datasets with a publication DOI. In addition, all software should be open access. This is what differentiates EGU from other comparable journals. This should not be an exception.

**Response:**

We appreciate this comment. Links to PANGAEA will be included in the final revised version.

**Reviewer comment:** Note [page 30]: Table A1 caption: Where does these percentages come from?
**Response:**
We appreciate this comment. The citation for MLCM algorithm which produces cloud filtering and surface classification and the lat-lon range of campaign operation is added in the caption in the revised version (latitudes in the range of 77.8∼82.4°N , and longitude in the range of -0.25∼20.5°E).

**Reviewer comment:** Note [page 33]: Line 597: Exact URLs should be provided. Note [page 33]: Line 600: Grant numbers should be listed.
**Response:** The text is added/modified in the revision.

**2 Second round of revision**

**2.1 Response to Reviewer 1**

**Reviewer comment:** Line 83-86. As discussed above, a neural network cannot a priori "solve the 'inverse problem' ", as is misleadingly stated here.
**Response:**
The wording in the original sentence 'Following the physically consistent SD, scientific machine learning (SciML) models can be used to solve the 'inverse problem',...' has been rephrased:
'Following the physically consistent SD, scientific machine learning (SciML) models can be used to approximate solutions to the 'inverse problem'...,'

**Reviewer comment:** Line 135. Can you characterise the uncertainties used for pruning the channels to separate error coming from the network performance and the error originating from unsuitability of the channels? If yes, how? If not, why is this approach still valid?
**Response:** In our experiments, using AANN models with a network structure of $13 \times 7 \times 13$ or $12 \times 6 \times 12$ (heuristically, the number of neurons in the middle layer being half of the other two layers), we were able to correctly identify channels requiring additional calibration. However, this approach is quite heuristic and it is difficult to isolate errors from the two sources.
When unsuitable channels are omitted from our model, the overall band-averaged deviation is significantly reduced, and the reduction can be as great

as one order of magnitude: the 'restoration error' for an AANN to replicate the spectral radiance (input) can be reduced to less than 1 percent.

The validity of the approach is provided in the given reference [15]:

*After being trained by the same training dataset, the aaNN works as a duplicator. If the input data are within the range of the training dataset and the shape of the spectral $L_{rc}$ is very close to that of the training dataset, then the aaNN output will duplicate the input spectral $L_{rc}$ with a very high precision. But if some of the input data are out of the range or the shape of the spectral $L_{rc}$ is not included in the training dataset, then the output from the aaNN deviates significantly from the input spectral $L_{rc}$, i.e. the band-averaged percentage difference is larger than 5%. Therefore, by comparing the output from aaNN with the input spectral $L_{rc}$, we can identify pixels that are out of scope of the training dataset. This capability of the aaNN is due to the bottleneck layer in the neural network structure. The aaNN also has 3 hidden layers. The number of neurons in the first and third layers is set to equal the number of inputs. The second layer is the bottleneck layer with a much smaller number of neurons, and designed such that a well trained aaNN is able to duplicate only input data available in the training dataset.*

As a side note, in addition to selecting channels, the authors of OC-SMART include the aaNN as a pre-processing step in the ocean-color retrieval workflow: when satellite data are entered, the aaNN checks for pixels outside the scope of the training dataset.

**Reviewer comment:** Line 298. As discussed above, give information about all the networks performances.

**Response:** We greatly value your remark on local minima and this recommendation. The results of three other neural network (NN) models that we trained have been added to Table B1. In the meantime, Figure B1 was added to compare the histograms of the four NN and four machine learning (ML) models. Note that the validation loss (on the 20% of data excluded from model training) of all ML models was at least one magnitude greater than that of the four NN model. Figure B1 (c) and (d) illustrates how this 'under-performance' impacts final retrievals.

From Figure B1 (a) and (b), it can be seen that the four neural network models are highly comparable, at least in terms of the sampled data used
* * *
[15]OC-SMART: A machine learning based data analysis platform for satellite ocean color sensors

to generate the histograms. In addition to Fig.B1-a, as proposed, we gathered additional independent data to illustrate the overall value ranges of the four NNs. The two panels suggest, citing your remark, that "any ambiguous solutions still give similar results or maybe there even exists an analytic inversion'. However, we recognize that this small sample is by no means exhaustive, so there may be adversarial instances that we missed. Therefore, the concluding remarks on page 34 were also revised to reflect this limitation.

**Reviewer comment:** Figure 4 and 6. Does this validation contain the data used for selecting the best models? (It should not!). Generally also the comparison with other products should not be made over areas that have been included in the training or selection process of the neural network models, as this would wrongly skew the results in favor of the machine learning approach.
**Response:** Approximately fifty percent of the data presented in Fig. 4(d) was used to determine the final model mentioned in subsequent sections.
Although this approach is not ideal, it cannot be avoided: (a) It is not possible to include the retrieval results of all NN models and compare them with the results of other approaches (melt-pond detection, MCD43, and direct-estimation method), and (b) the NN models exhibited identical loss on the independent testing dataset (20% of the dataset generated by the radiative transfer model).
Therefore, we had to rely on *in-situ* measured data to select a 'winning model', as no other data sources were available for this purpose.

**Reviewer comment:** Line 591. The discussion needs to include the fact that the performance of the forward model is a hard limit on the performance of the retrieval, as sea ice in particular is known to be difficult to model accurately.
**Response:** We appreciate the recommendation. The relevant information is added in the Conclusion section.

**2.2   Response to Editor**

**Editor comment:** Unfortunately I find your response to the reviewer comment about separating training and validation data insufficient (the second last comment). Please could you include clear statements in that regard in the revised manuscript as well. Were the 50% of the data not used in later validation? Please clearly describe in the manuscript what data were used for what purpose, and why not if they could not be separated.

**Response:**
Section 2.6 was reorganized to emphasize on how the synthetic dataset (SD) and independent validation dataset (sample from measurement) are used to obtain the final model. In particular, the new edits in Lines 274-275, Lines 293-295 and footnote 4 in the track-changes file details what data were used for what purpose. They are completely separate processes, which we specified in the flowchart as 'out-of-sample validation' and 'validation, passed?' steps. We hope the edition this time clarifies this issue.